# Learning Sparse Graphs for Functional Regression using Graph-induced Operator-valued Kernels

**Akash Saha**                                                                    *akashsaha@iitb.ac.in*
*IEOR, IIT Bombay*
*Mumbai, India*

**P. Balamurugan**                                          *balamurugan.palaniappan@iitb.ac.in*
*IEOR, IIT Bombay*
*Mumbai, India*

**Reviewed on OpenReview:** *https://openreview.net/forum?id=f9l4eiPKpV*

## Abstract

A functional regression problem aims to learn a map $\mathfrak{F} : \mathcal{Z} \mapsto \mathcal{Y}$, where $\mathcal{Z}$ is an appropriate input space and $\mathcal{Y}$ is a space of output functions. When $\mathcal{Z}$ is also a space of functions, the learning problem is known as function-to-function regression. In this work, we consider the problem of learning a map of the form $F : \mathcal{Z}^p \mapsto \mathcal{Y}$, a many-to-one function-to-function regression problem, where the aim is to learn a suitable $F$ which maps $p$ input functions to an output function. In order to solve this regression problem with $p$ input functions and a corresponding output function, we propose a graph-induced operator-valued kernel (OVK) obtained by imposing a graphical structure describing the inter-relationships among the $p$ input functions. When the underlying graphical structure is unknown, we propose to learn an appropriate Laplacian matrix characterizing the graphical structure, which would also aid in learning the map $F$. We formulate a learning problem using the proposed graph-induced OVK, and devise an alternating minimization framework to solve the learning problem. To learn $F$ along with meaningful and important interactions in the graphical structure, a minimax concave penalty (MCP) is used as a sparsity-inducing regularization on the Laplacian matrix. We further extend the alternating minimization framework to learn $F$, where each of the $p$ constituent input functions as well as the output function are multi-dimensional. To scale the proposed algorithm to large datasets, we design an efficient sample-based approximation algorithm. Further, we provide bounds on generalization error for the map obtained by solving the proposed learning problem. An extensive empirical evaluation on both synthetic and real data demonstrates the utility of the proposed learning framework. Our experiments show that simultaneous learning of $F$ along with sparse graphical structure helps in discovering significant relationships among the input functions, and motivates interpretability of such relationships driving the regression problem.

## 1 Introduction

Learning to predict functional output from a suitable input is characterized as a functional regression problem, which aims at learning a function-valued function $F : \mathcal{Z} \to \mathcal{Y}$, where $\mathcal{Z}$ is an appropriate input space and $\mathcal{Y}$ is an output space of functions. In many scenarios, multiple inputs decide the value of an output, which gives rise to functional regression problems of the form $F : \mathcal{Z}^p \to \mathcal{Y}$, where $p$ is the number of inputs considered. Even more interesting is the case where interactions among the $p$ inputs can be used in a precise manner to predict $y \in \mathcal{Y}$. In particular, we consider $\mathcal{Z}$ to be a space of functions, hence learning a map $F : \mathcal{Z}^p \to \mathcal{Y}$ is called a many-to-one function-to-function regression problem. Without loss of generality, we refer to this many-to-one function-to-function regression problem as the functional regression problem considered throughout this paper. Applications of this type of problems can be found in weather forecast-

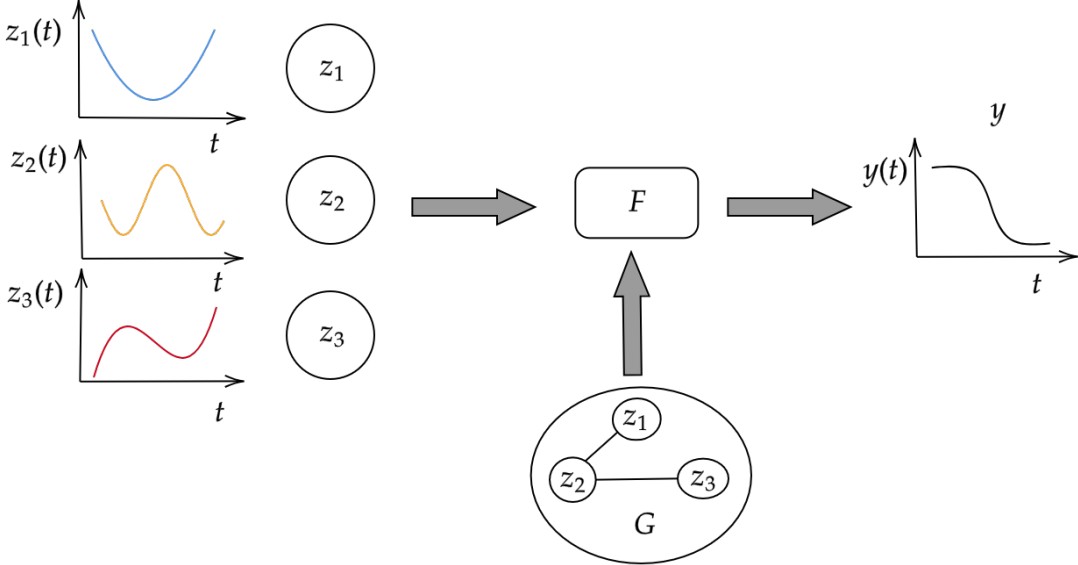

Figure 1: Illustrative example of a functional regression problem, where $z_1, z_2, z_3$ represent atmospheric pressure at 3 stations in a region, graph $G$ depicts the inter-relationships among $z_1, z_2, z_3$ and $y$ represents the average temperature of the region. $F$ maps $z_1, z_2, z_3$ to $y$ incorporating $G$.

ing where different weather parameters in stations measured at multiple timepoints across a month can be characterized as functional inputs used to determine the average rainfall as a time-varying function in that month. Similarly, emissions from a factory in a day can be predicted as a function of time, based on the functional data obtained from readings of different components involved in the manufacturing process at different timepoints in that day. In sports analytics, the movement data of different players throughout the game can let us know the influence of a particular strategy in ball possession/movement as a functional output over the duration of the game. Thus in all these applications, we notice situations where a set of input functions interact to produce an output function. Even though digital data is discrete, systems where the inherent data produced is smooth and continuous by nature, can be modeled as functions over a suitable domain (Ramsay, 1982) to leverage the variations based on that domain.

Consider a simple functional regression problem illustrated in Figure 1, where input functions $z_1, z_2, z_3 \in \mathcal{Z}$ denote the atmospheric pressure measurements of 3 nearby weather stations and the output function $y \in \mathcal{Y}$ denotes the average temperature of the region, throughout a particular day. For predicting $y$, considering the input functions $z_1, z_2$ and $z_3$ without any relation among them may be restrictive as inherent relations between the input functions may dictate the generation of $y$. In order to capture interactions among $z_1, z_2, z_3$, we introduce a graph structure $G$ between $z_1, z_2$ and $z_3$ in Figure 1, where the nodes of $G$ represent $z_i$'s and the edges depict potential relations among them. The graph structure $G$ will be useful in representing the influences and inter-relations among $z_i$'s, which can be useful in the prediction of $y \in \mathcal{Y}$ using $F$. We propose a framework for combining the impact of $z_1, z_2, \ldots, z_p$ with the additional information of $G$ to predict $y$. In determining the output function $y$, the graphical structure $G$ on the input functions may be known from domain knowledge and can possibly be directly incorporated to learn $F$. A more interesting case is when $G$ is unknown and needs to be learned along with $F$. Learning the graph structure $G$ would help to discover interactions among $z_i$'s which facilitate predicting $y$. When the number of input functions $z_1, z_2, \ldots, z_p$ grow larger, the associated graph $G$ might also become dense with many edges and incorporating such dense $G$ might lead to computational difficulties and would also lead to spurious connections/edges which lack interpretability. Thus, learning a sparse graphical structure $G$ on input functions becomes instrumental in understanding the significant relationships that drive the functional regression problem to predict the output function.

In this work, we consider kernel methods to learn the map $F : \mathcal{Z}^p \to \mathcal{Y}$, either using a priori knowledge of $G$ or by learning $G$ along with $F$. Kernel methods have been a popular class of methods that use kernel functions to associate inputs to a higher dimensional feature space and find applications in classification, clustering and regression (Shawe-Taylor et al., 2004). For a simple scalar-valued regression problem, the aim is to learn $f : X \to \mathbb{R}$, where $X$ is an appropriate input space of vectors. A scalar-valued kernel $k : X \times X \to \mathbb{R}$ associates two inputs in $X$ to a real number which provides a measure of similarity between those inputs. Scalar-valued kernels which are positive semi-definite are associated to a (unique) space $\mathcal{H}$ of candidate functions $f$ mapping input space $X$ to $\mathbb{R}$, $\mathcal{H}$ being referred to as a reproducing kernel Hilbert space (RKHS). The function $f$ to be learned resides in the aforementioned space $\mathcal{H}$ of functions which enables us to formulate a regularized loss minimization problem over $\mathcal{H}$ (Shawe-Taylor et al., 2004), whose solution can be used to predict the desired output for input samples. On the other hand, for a functional regression problem, an extension of scalar-valued kernel to operator-valued kernel (OVK) of the form $K : \mathcal{Z} \times \mathcal{Z} \to \mathcal{L}(\mathcal{Y})$ associates two input functions to a bounded linear operator on the output space of functions instead of a real number (Kadri et al., 2016; Saha & Palaniappan, 2020). An operator-valued kernel $K$ which is positive semi-definite is associated to a (unique) space $\mathcal{H}_K$ of candidate functions $F$ mapping input space $\mathcal{Z}$ to $\mathcal{Y}$, $\mathcal{H}_K$ being called a function-valued reproducing kernel Hilbert space (Kadri et al., 2016). Similar to scalar-valued kernel setting, in the operator-valued kernel setting too, a regularized loss minimization learning problem can be formulated in the function-valued RKHS $\mathcal{H}_K$. The reproducing property of OVKs (Definition A.1) helps in reformulating the learning problem in the output space $\mathcal{Y}$, enabling algorithms to be developed for solving the resultant learning problem.

In this paper, we consider another natural extension of this kernel-based framework which leads to a case where $p$ input functions and their interactions among themselves (captured by a graph $G$) can be used to predict the output function. To learn $F : \mathcal{Z}^p \to \mathcal{Y}$, we incorporate $p$ input functions along with the underlying graphical structure to create a graph-induced operator-valued kernel that induces a corresponding function-valued RKHS to facilitate learning of $F$. In order to use graph-induced OVKs in the task of functional regression based on $p$ input functions with unknown graphical structure, we propose to learn $F$ simultaneously along with the graphical structure. Predicting output function based on a graph-induced OVK constructed using graphical structure $G$ over $p$ input functions, to our knowledge, is a novel problem and has not been explored much in literature. In this context, we outline our major contributions below.

**Contributions:** We aim to address the following objectives in this work.

- We propose a graph-induced OVK for solving a functional regression problem with $p$ input functions $z_1, z_2, \ldots, z_p$ and their interactions represented using a graphical structure (known/unknown) to predict a corresponding output function $y$. This is enabled by Proposition A.1 considering the Laplacian matrix ($L$) of the underlying graph $G$.

- For practical scenarios where the underlying graphical structure is unknown, we provide a construction of graph-induced OVK and propose to jointly learn the functional regression map $F$, along with the graphical structure $G$ represented by $L$ and $D$, where $D$ is a diagonal matrix with non-negative entries signifying the impact of individual input functions on $y$. A regularized loss minimization problem is formulated using $L, D$ and $F$ in the function-valued RKHS associated with the graph-induced OVK.

- We propose an alternating minimization framework for solving the designed regularized loss minimization problem to learn $L, D$ and the map $F$. The functional regression map $F$ is learned for a fixed $L$ and $D$ by adapting Operator based Minimum Residual (OpMINRES) algorithm (Saha & Palaniappan, 2020) to solve a linear operator system associated with the graph-induced OVK. For a fixed $F$, matrices $L$ and $D$ are learned using projected gradient descent. To learn sparse $L$, we introduce minimax concave penalty (MCP) (Ying et al., 2020) as a sparsity-inducing regularization on the Laplacian matrix $L$.

- We further extend the proposed alternating minimization framework to solve a multi-dimensional functional regression problem, where each input function $z_i \in \mathcal{Z}$, $i \in \{1, 2, \ldots, p\}$ and $y \in \mathcal{Y}$ are multi-dimensional.

- In order to scale the proposed alternating minimization framework to handle large datasets, we design an efficient sample-based approximation algorithm which enables to solve the learning problem over only a carefully chosen subset of training samples.

- We establish bounds on generalization error for the map $F$ obtained by solving the proposed learning problem. Our generalization analysis also incorporates the learning of graph-induced OVK.

- An extensive empirical evaluation on both synthetic and real data has been carried out and the comparison results demonstrate the efficacy of the proposed learning framework. Further our experiments show that simultaneous learning of sparse graphical structure along with the function-valued regression map $F$ establishes interpretable relationships driving the functional regression problem.

## 2    Paper Organization

In Section 3, related works from the areas of functional data analysis, functional regression and graph learning are discussed. The notations used in this paper have been summarized in Section 4. The proposed framework for solving a functional regression problem using a graphical structure is covered in Section 5. The graph-induced OVK is introduced and a representer theorem for the learning problem with an unknown graph structure is presented in Section 5.1. An alternating minimization framework is proposed in Section 5.1 for jointly learning an unknown graphical structure on the functional inputs and the map $F$. Inducing sparsity using a MCP regularization when learning the unknown graphical structure is also discussed in Section 5.1. An extension of the proposed framework for solving multi-dimensional functional regression problems and a sample-based approximation algorithm for scaling up to large training data are also presented in Section 5.1. In Section 6, the bounds on generalization error of the learned map $F$ are established by incorporating learning of graph-induced OVK. Experiments using the proposed alternating minimization framework and comparative results have been illustrated in Section 7. Section 8 provides the conclusion of the paper.

## 3    Related Work

As our work lies in the confluence of various areas of research, we divide the related work based on the different areas below.

**Functional Data Analysis**: Continuous functions over a time interval have been explored as the central part of functional data analysis (FDA) in (Ramsay, 1982) and (Ramsay & Dalzell, 2018). FDA techniques have evolved significantly with non-parametric approaches (Ferraty & Vieu, 2006) and functional principal component analysis (FPCA) (Happ & Greven, 2018) becoming prevalent tools. These approaches have found applications in sparse longitudinal data (Yao et al., 2005), classification involving functional data (Rossi & Villa, 2006) and clustering for multivariate functional data (Jacques & Preda, 2014).

**Functional Regression**: In the context of functional regression, Oliva et al. (2015) uses projections on orthonormal basis systems for input and output spaces to estimate regression maps based on random basis functions from random Fourier features (Rahimi & Recht, 2007). Operator-valued kernel methods have found applications for vector-valued data (Micchelli & Pontil, 2005) and function-valued data (Kadri et al., 2016) for solving corresponding vector-valued and functional regression problems. The construction of a positive semi-definite operator-valued kernel used to learn a function-valued mapping in a corresponding reproducing kernel Hilbert space (RKHS) is considered in (Kadri et al., 2016), while Saha & Palaniappan (2020) uses indefinite operator-valued kernels to learn a function-valued function in a corresponding reproducing kernel Krein space (RKKS). Hullait et al. (2021) uses a robust functional linear regression model based on robust FPCA (Bali et al., 2011) to predict a response function using a predictor function without considering multiple input functional data and their graphical structure. Another approach in (Bouche et al., 2021) uses kernel-based projection learning with a finite (not necessarily orthogonal) basis for the output space. High dimensional functional data has been used in (Gahrooei et al., 2020) to perform function-to-function regression for a functional response output using a linear combination of functional inputs considered as covariates. However, associations among the functional inputs have not been considered in (Gahrooei et al., 2020). Functional deep learning methods have been developed to solve regression problems using functional direct neural network

and functional basis neural network (Rao & Reimherr, 2023). In functional direct neural network, continuous neurons interact with learned weight functions, whereas in functional basis neural network, basis functions are used to encode the continuous neurons as well as weight functions. Both functional direct neural network and functional basis neural network in (Rao & Reimherr, 2023) require large amount of data for training. On the contrary, our work is related to a setting with limited number of training samples which is useful in various practical scenarios where data availability is restricted.

**Graph Learning**: Learning graph structure has been an active area of research in machine learning. In (Dong et al., 2016), a Laplacian matrix corresponding to the graph structure on the observed input signals is learned by using a vectorized optimization problem with a smoothness assumption over the signals. The graph Laplacian learning algorithm is based on an alternating minimization scheme for learning the Laplacian matrix as well as the missing/noisy signals. Pu et al. (2021b) extends this idea by using a kernel-based learning problem for determining the Laplacian matrix of a graph structure for smooth input signals. Kernels are used to learn relationships between the input signals as well as the inter-relationships between covariates such as timestamp of recording an observation. Another popular approach for graph structure learning in (Qiao et al., 2019) is based on estimation of precision matrix (inverse of covariance matrix) for functional data corresponding to the $p$ nodes in the graph, assuming that the data arises from a $p$-dimensional multivariate Gaussian process. A differential functional graphical model has been considered in (Zhao et al., 2019) which learns a differential graph to characterize the difference between conditional dependencies of two different populations which is determined using their respective samples. Extending the idea of determining precision matrix for capturing conditional dependence between the nodes, Qiao et al. (2020) proposes doubly functional graphical models to capture the evolving conditional dependence relationship among the sampled functions corresponding to the nodes. Instead of learning a single graph based on the data, Pu et al. (2021a) learns a graph topology with topological difference variational autoencoder for graph learning.

A motivating work (Gómez et al., 2021) addresses function-to-function linear regression problem with both known/unknown directed graph structure where the main focus is on root cause analysis and the node representing output function is also a part of the graphical structure containing nodes corresponding to the input functions. The graph structure considered is learned based on a neighborhood selection method (Meinshausen & Bühlmann, 2006) to determine the set of candidate parents for each node in order to solve the linear regression problem. Multivariate Gaussian processes have been used with basis functions in (Gómez et al., 2021) to model a directed acyclic graph which enables solving a function-to-function linear regression problem. However, in our approach, the goal of learning an undirected graph structure only on the $p$ input functions is different from the parent-based directed acyclic graph structure assumption in (Gómez et al., 2021). Moreover, our OVK based approach helps to learn non-linear relations in comparison to the linear model considered in (Gómez et al., 2021).

## 4   Notations

We consider a functional regression problem with the input space as $\mathcal{X} = (L^2([0,1]))^p$ and the output space as $\mathcal{Y} = L^2([0,1])$, where $p \in \mathbb{N}$ and $L^2([a,b])$ denotes the space of equivalence classes of square integrable functions on $[a,b]$, $a,b \in \mathbb{R}$ and $a < b$. The notation $[n]$ denotes the the set $\{1,2,\ldots,n\}$, for $n \in \mathbb{N}$. In order to denote elements of the input space $\mathcal{X}$, we use the notation $x = (x_1, x_2, \ldots, x_p) \in \mathcal{X}$. We denote the graphical structure over the input functions $x_1, x_2, \ldots, x_p$ as $G = (V, E)$, where $V$ is the set of $p$ vertices corresponding to the functional input variables $x_1, x_2, \ldots, x_p$. The degree of a vertex $v \in V$ is denoted by $\deg(v)$. $K$ refers to an OVK mapping from $\mathcal{X} \times \mathcal{X}$ to $\mathcal{L}(\mathcal{Y})$, where $\mathcal{L}(\mathcal{Y})$ is the set of bounded linear operators over the output space $\mathcal{Y}$. For a matrix $M \in \mathbb{R}^{k \times k}$, where $k \in \mathbb{N}$, $M_{i,j}$ denotes the $(i,j)$-th element of $M$ and $M_i \in \mathbb{R}^k$ denotes the $i$-th column of $M$. Hence, we refer to matrix $M$ as $[M_{i,j}]_{i,j=1}^k$ or $[M_1, M_2, \ldots, M_k]$. The transpose of $M$ is denoted by $M^\top$. The notation $\text{diag}(d_1, d_2, \ldots, d_p)$ denotes a $p \times p$ diagonal matrix with the diagonal entries as $d_1, d_2, \ldots, d_p$, where $d_i \in \mathbb{R}$, for $i \in [p]$. For a matrix $D \in \mathbb{R}^{p \times p}$, $Diag(D)$ denotes the $p \times 1$ vector containing the diagonal elements of $D$. Based on the context, other relevant notations will be introduced in the paper as required and suitable descriptions will be provided for them.

## 5 Functional Regression based on a Graphical Structure

In this section, we introduce the functional regression problem with the aim of incorporating the graph structure to aid the regression task. To model the graphical structure, we assume that $G = (V, E)$ represents an undirected graph where $V$ denotes the node set with $|V| (= p)$ nodes and $E$ denotes the edge set of $G$. Let $x(t) = (x_1(t), x_2(t), \ldots, x_p(t))$ denote the functional variables for a given domain $t \in \mathcal{T}$ where each $x_i$ is represented by node $v_i \in V$, for $i \in [p]$. Recall that $L^2([a, b])$ denotes the space of equivalence classes of square integrable functions on $[a, b]$, $a, b \in \mathbb{R}$ and $a < b$. For simplicity, we assume $x_i \in L^2([0, 1])$, hence $\int_0^1 x_i^2(t)dt < \infty, i \in [p]$. We discuss the realistic case of functional regression with an unknown graph structure here. For a related discussion on the case of a known graph structure, we refer the reader to Appendix A.1.

### 5.1 Learning with Unknown Graph Structure

Consider a system where a set of input functions determines the output (or response) function. Let the system be modeled based on $p$ input functional variables $x_1(t), x_2(t), \ldots, x_p(t)$, where $x_i \in L^2([0, 1])$, $i \in [p]$. A functional response variable $y(t)$ is used to model output of the system where $y \in L^2([0, 1])$. (Note that $[0, 1]$ can be replaced with any closed time interval based on the application.)

The undirected graph structure of the functional input variables is represented by a suitable graph $G = (V, E)$, where $V = \{v_1, v_2, \ldots, v_p\}$ and $E = \{\{v_i, v_j\} | v_i$ is connected to $v_j, 1 \leq i, j \leq p\}$ is the edge set which characterizes the underlying relationship between the variables. Note that the notation for an edge uses an unordered pair $\{v_i, v_j\}$ which characterizes the undirected nature of the graph $G$. In order to model the relation between functional input variables $x_1, x_2, \ldots, x_p$ and functional response variable $y$, we use the following map $F$:

$$y = F(x_1, x_2, \ldots, x_p, G). \tag{1}$$

Note that $F$ now depends explicitly on the graph $G$ in addition to the input functions $x_1, x_2, \ldots, x_p$.

For most problems in real life, the underlying graphical structure encoding the inter-relationships among the input functions $x_1, x_2, \ldots, x_p$ is not known. In such situations, the underlying graph on the input functions has to be learned with simultaneous prediction of the functional response variable $y \in \mathcal{Y}$ using $x = (x_1, x_2, \ldots, x_p) \in \mathcal{X}$. Considering an undirected simple graph structure on the functional variables may result in encountering an integer programming based optimization problem which will lead to a computationally harder problem in addition to the functional regression task. We consider a relaxation in this aspect by allowing weighted undirected simple graphs in our approach. With a slight abuse of notation, let the graph structure be given by $G = (V, W)$, where $|V| = p$ and $W \in \mathbb{R}^{p \times p}$ is symmetric with $w_{i,j} \geq 0$, where $w_{i,j} = 0$ whenever vertices $v_i$ and $v_j$ are not connected and $w_{i,j} > 0$ denotes the weight assigned to the edge between $v_i$ and $v_j$. Hence the graph Laplacian matrix can be represented as $L = \mathbb{D} - W$, where $\mathbb{D} = \text{diag}(\mathbb{D}_{1,1}, \mathbb{D}_{2,2}, \ldots, \mathbb{D}_{p,p})$ with $\mathbb{D}_{i,i} = \sum_{j=1}^p w_{i,j}$. It can be shown that $L$ is positive semi-definite by virtue of being diagonally dominant with non-negative diagonal entries (Golub & Van Loan, 1996).

We consider the notations $x = (x_1, x_2, \ldots, x_p) \in \mathcal{X} (= (L^2([0, 1]))^p)$ and $y \in \mathcal{Y} (= L^2([0, 1]))$ to represent an arbitrary sample $(x, y)$. To learn the mapping $F$, consider the training data of $n$ samples given as $\left\{(x^{(i)}, y^{(i)})\right\}_{i=1}^n$, where $x^{(i)} = (x_1^{(i)}, x_2^{(i)}, \ldots, x_p^{(i)}) \in \mathcal{X}$ and $y^{(i)} \in \mathcal{Y}$. In order to learn $F$, we develop an operator-valued kernel which can leverage the structural information of $G$.

**Definition 5.1** (**Graph-induced Operator-valued Kernel**). A graph-induced operator-valued kernel is defined as

$$(K^G(x, x')y)(t) = k_1(x, x'; G) \int_0^1 k_2(s, t)y(s)ds, \tag{2}$$

where $k_2$ is a scalar-valued kernel on $\mathbb{R}^2$, $G$ is a graph associating the $p$ input functions in $(x_1, \ldots, x_p) \in \mathcal{X}$ and $k_1$ is defined as

$$k_1(x, x'; G) = e^{-\gamma(x-x')^\top (L+D)(x-x')}, \ \gamma > 0,$$

where $L$ is the Laplacian matrix of graph $G$ and $D$ is a diagonal matrix consisting of non-negative entries.

$K^G$ associates a pair $x, x' \in \mathcal{X}$ with output function $y \in \mathcal{Y}$ where $G$ is the graph which incorporates the interactions of $p$ constituent input functions of $x$ and $x'$. $k_2$ inside the Hilbert-Schmidt Integral (HSI) operator $\int_0^1 k_2(s,t)y(s)ds$ is a scalar-valued kernel on $\mathbb{R} \times \mathbb{R}$. The radial basis function (RBF) type kernel construction of $k_1$ involves $L + D$, where the addition of the Laplacian $L$ with a diagonal matrix $D$ with non-negative entries results in a diagonally perturbed Laplacian matrix (Bapat et al., 2001) (see (Kurras et al., 2014; Aliakbarisani et al., 2022) for applications of perturbed Laplacians). If $k_1$ is positive semi-definite and if $k_2$ is positive semi-definite (implying that the HSI operator is positive semi-definite), then the construction in (2) is known to be positive semi-definite (Kadri et al., 2016). The addition of diagonal matrix $D$ (with non-negative entries) to $L$ in (2) preserves the positive semi-definiteness of the kernel $k_1$. Note that the notation of the form $x^\top L x'$ used in $k_1$ denotes an inner product structure given by $x^\top L x' = \sum_{i,j=1}^p \int_0^1 x_i(t)L_{ij}x'_j(t)dt$ (see Appendix A.1 for details regarding this inner product structure).

Without loss of generality, henceforth we refer to the graph-induced operator-valued kernel $K^G$ as $K$ for simplicity. The matrix $L = [L_{i,j}]_{i,j=1}^p$ satisfies the conditions: $L\mathbf{1} = 0$ and $L_{i,j} = L_{j,i} \leq 0, \forall i \neq j$. Note that the graph-induced operator-valued kernel $(K(x,x')y)(t) = k_1(x,x';G)\int_0^1 k_2(s,t)y(s)ds$ as defined in (2) with $k_1(x,x';G) = e^{-\gamma(x-x')^\top(L+D)(x-x')}$, $\gamma > 0, x, x' \in \mathcal{X}, y \in \mathcal{Y}$ is positive semi-definite (see proof of Proposition A.1 in Appendix A.1), which ensures that there exists a unique function-valued RKHS $\mathcal{H}_K$ induced by $K$ (Theorem 1 (Kadri et al., 2016)). Now to simultaneously learn $F$ including the graph structure represented by $L$ and $D$, the following optimization problem is formulated:

$$\widetilde{F}, \widetilde{L}, \widetilde{D} = \underset{F \in \mathcal{H}_K, L \in \mathcal{L}, D \in \mathcal{D}}{\arg\min} \sum_{i=1}^n \|y^{(i)} - F(x^{(i)})\|_{\mathcal{Y}}^2 + \lambda\|F\|_{\mathcal{H}_K}^2 + \rho_L \sum_{i=1}^n x^{(i)^\top} L x^{(i)} + \rho_D\|D\|_F^2, \qquad (3)$$

where $\mathcal{L} = \{L \in \mathbb{R}^{p \times p} | L\mathbf{1} = 0, L_{i,j} = L_{j,i} \leq 0, \forall i \neq j\}$ denotes the set of all matrices satisfying the constraints associated with Laplacian matrices of the graph $G = (V, W)$ with $W = [w_{i,j}]_{i,j=1}^p$. Note that $\mathcal{D} = \{D \in \mathbb{R}^{p \times p} | D_{i,i} \geq 0, D_{i,j} = 0, \forall i \neq j\}$ denotes the set of all diagonal matrices with non-negative diagonal entries, $\|.\|_F$ is the Frobenius norm and $\lambda, \rho_L, \rho_D > 0$. The regularization of $D$ in (3) using Frobenius norm provides control on the values in $D$. Note the absence of a Frobenius norm based regularizer for $L$ in (3). A different smoothness term $x^\top L x = \frac{1}{2}\sum_{i,j=1}^p w_{i,j}\|x_i - x_j\|_{\mathcal{Y}}^2$ is considered (a similar term is considered in (Humbert et al., 2021)), which provides an improved interaction-based data-oriented regularization instead of using a simple matrix norm based regularization of $L$.

Using the representer theorem A.2 and the reproducing property of OVK $K$, the minimization problem (3) is equivalently reduced to be in terms of $\mathbf{u} \in \mathcal{Y}^n$ instead of $F \in \mathcal{H}_K$ as follows (see Appendix A.2):

$$\min_{F,L,D} J(F, L, D) = \sum_{i=1}^n \|y^{(i)} - F(x^{(i)})\|_{\mathcal{Y}}^2 + \lambda\|F\|_{\mathcal{H}_K}^2 + \rho_L \sum_{i=1}^n x^{(i)^\top} L x^{(i)} + \rho_D\|D\|_F^2 \qquad (4)$$

$$\implies \min_{\mathbf{u},L,D} J(\mathbf{u}, L, D) = \sum_{i=1}^n \left\|y^{(i)} - \sum_{j=1}^n K(x^{(i)}, x^{(j)})u_j\right\|_{\mathcal{Y}}^2 + \lambda \sum_{i,j=1}^n \langle K(x^{(i)}, x^{(j)})u_i, u_j\rangle_{\mathcal{Y}} \qquad (5)$$

$$+ \rho_L \sum_{i=1}^n x^{(i)^\top} L x^{(i)} + \rho_D\|D\|_F^2.$$

To solve (3) (or (5) equivalently) we now propose an alternating minimization framework where $J(\mathbf{u}, L, D)$ is optimized alternatively with respect to $\mathbf{u} \in \mathcal{Y}^n, L \in \mathcal{L}$ and $D \in \mathcal{D}$. We now discuss the steps involved in alternating minimization of $J(\mathbf{u}, L, D)$.

### 5.1.1 Minimization with respect to $\mathbf{u}$ for fixed $L, D$

Assuming fixed $L, D$, and from the reproducibility property of $K$ and representer theorem A.2, $J(\mathbf{u}, L, D)$ from (5) simplifies to the following system of linear operator equations in $\mathbf{u}$ (see Appendix A.8):

$$(\mathbf{K} + \lambda I)\mathbf{u} = \mathbf{y}, \qquad (6)$$

where $\mathbf{K}_{i,j}u = K(x^{(i)}, x^{(j)})u = k_1(x^{(i)}, x^{(j)}; G)\bar{k}_2(u), \forall u \in \mathcal{Y}$ with $k_1(x, x'; G) = e^{-\gamma(x-x')^\top(L+D)(x-x')}$, $\bar{k}_2(u)(t)$ is defined using an exponential kernel in (2) as $\bar{k}_2(u)(t) = \int_0^1 e^{-\gamma_{op}|s-t|}u(s)ds, \gamma_{op} > 0, s, t \in \mathbb{R}$, $\mathbf{u} = [u_1, u_2, \ldots, u_n]^\top \in \mathcal{Y}^n$ and $\mathbf{y} = [y^{(1)}, y^{(2)}, \ldots, y^{(n)}]^\top$. In our framework, we consider a particular choice of kernel $k_2$ on $\mathbb{R}^2$ as $k_2(s,t) = e^{-\gamma_{op}|s-t|}, \gamma_{op} > 0$. The OpMINRES algorithm (Saha & Palaniappan, 2020) solves the system (6) by using an iterative Krylov subspace minimal residual method. Consider $\mathbf{P} := \mathbf{K} + \lambda I$, OpMINRES minimizes $\|\mathbf{y} - \mathbf{Pu}\|_{\mathcal{Y}^n}$, where the norm is defined as $\|\xi\|_{\mathcal{Y}^n} = \sqrt{\sum_{i=1}^n \int_0^1 \xi_i^2(t)dt}$, for $\xi = (\xi_1, \xi_2, \ldots, \xi_n) \in \mathcal{Y}^n$. The steps involved in $k$-th iteration of OpMINRES algorithm are given below:

1. Transforming the linear operator system $(\mathbf{K} + \lambda I)\mathbf{u} = \mathbf{y}$ into a linear system in $\mathbb{R}^k$ using a Lanczos-based method (Lanczos, 1950), called Operator-valued Lanczos (or OpLanczos) scheme.

2. Solving the linear system of the previous step using QR decomposition.

3. Transforming the result obtained in step 2 appropriately to retrieve a solution in $\mathcal{Y}^n$.

Using the Krylov subspace $\mathcal{K}_k(\mathbf{P}, \mathbf{y}) = \text{span}\{\mathbf{y}, \mathbf{Py}, \mathbf{P}^2\mathbf{y}, \ldots, \mathbf{P}^{k-1}\mathbf{y}\}$ obtained at the $k$-th iteration, OpMINRES obtains an approximation $\mathbf{u}^k$ to the original solution using the following:

$$\mathbf{u}^k = \arg\min_{\boldsymbol{\theta} \in \mathcal{K}_k(\mathbf{P}, \mathbf{y})} \|\mathbf{y} - \mathbf{P}\boldsymbol{\theta}\|_{\mathcal{Y}^n}. \tag{7}$$

The problem in (7) is transformed into a problem in $\mathbb{R}^k$ by using OpLanczos method. The OpLanczos method at the $k$-th iteration, tridiagonalizes $\mathbf{P}$ to get $\mathbf{P}Q_k = Q_k T_k$, where $T_k$ has a tridiagonal structure given by

$$T_k = \begin{bmatrix} \alpha_1 & \beta_2 & & & & 0 \\ \beta_2 & \alpha_2 & \beta_3 & & & \\ & \beta_3 & \alpha_3 & \ddots & & \\ & & \ddots & \ddots & \beta_{k-2} & \\ & & & \beta_{k-1} & \alpha_{k-1} & \beta_k \\ 0 & & & & \beta_k & \alpha_k \end{bmatrix}, \tag{8}$$

and $Q_k = [q_1, q_2, \ldots, q_k]$, where the $q_i$'s belonging to $\mathcal{Y}^n$ are orthonormal and $q_1$ is generally assumed to be $\mathbf{y}/\|\mathbf{y}\|_{\mathcal{Y}^n}$. Further, the relation $\mathbf{P}Q_k = Q_{k+1}\overline{T}_k$ is also satisfied for a suitably defined $\overline{T}_k$. Using $Q_k$, $\boldsymbol{\theta} \in \mathcal{Y}^n$ can be written as $\boldsymbol{\theta} = Q_k\vartheta$ for an appropriate $\vartheta \in \mathbb{R}^k$. Hence we have:

$$\min_{\boldsymbol{\theta} \in \mathcal{K}_k(\mathbf{P}, \mathbf{y})} \|\mathbf{y} - \mathbf{P}\boldsymbol{\theta}\|_{\mathcal{Y}^n} = \min_{\vartheta \in \mathbb{R}^k} \|\mathbf{y} - \mathbf{P}Q_k\vartheta\|_{\mathcal{Y}^n} = \min_{\vartheta \in \mathbb{R}^k} \|\mathbf{y} - Q_{k+1}\overline{T}_k\vartheta\|_{\mathcal{Y}^n} \tag{9}$$

$$= \min_{\vartheta \in \mathbb{R}^k} \|Q_{k+1}(\beta_1 e_1 - \overline{T}_k\vartheta)\|_{\mathcal{Y}^n}, \tag{10}$$

$$(\text{where } \beta_1 = \|\mathbf{y}\|_{\mathcal{Y}^n}, e_1 = [1, 0, \ldots, 0]^\top \text{ and } q_1 = \mathbf{y}/\|\mathbf{y}\|_{\mathcal{Y}^n})$$

$$= \min_{\vartheta \in \mathbb{R}^k} \|\beta_1 e_1 - \overline{T}_k\vartheta\|_2. \quad (\text{where } \|.\|_2 \text{ is the standard Euclidean norm.}) \tag{11}$$

Equation (10) reduces to (11) owing to the orthonormality of $\{q_1, q_2, \ldots, q_{k+1}\}$ (columns of $Q_{k+1}$). Solving for $\vartheta_k = \arg\min_{\vartheta \in \mathbb{R}^k} \|\beta_1 e_1 - \overline{T}_k\vartheta\|_2$ is done using QR decomposition. Now, the transformation from $\mathbb{R}^k$ back to $\mathcal{Y}^n$ to obtain $\mathbf{u}^k$ is achieved using the following: $\mathbf{u}^k = Q_k\vartheta_k = Q_k \left(\arg\min_{\vartheta \in \mathbb{R}^k} \|\beta_1 e_1 - \overline{T}_k\vartheta\|_2\right)$. In summary, we note that the minimization of $J$ with respect to $\mathbf{u}$ is obtained by using OpMINRES for fixed $L$ and $D$. Now, we proceed to the next step in the alternating minimization of $J(\mathbf{u}, L, D)$.

### 5.1.2 Minimization with respect to $L$ for fixed $\mathbf{u}, D$

The minimization of $J(\mathbf{u}, L, D)$ with respect to $L$ for fixed $\mathbf{u}, D$ is simplified by considering the symmetry of $L \in \mathcal{L}$. To simplify the computations, we introduce vectorization of matrices and half-vectorization

of symmetric matrices (Henderson & Searle, 1979). For a matrix $Z = [Z_{i,j}]_{i,j=1}^{q} \in \mathbb{R}^{q \times q}$ for $q \in \mathbb{N}$, the vectorization of $Z$ is defined as

$$\text{vec}(Z) = [Z_{1,1}, \ldots, Z_{q,1}, Z_{1,2}, \ldots, Z_{q,2}, \ldots, Z_{1,q}, \ldots, Z_{q,q}]^{\top}.$$

The half-vectorization of a symmetric matrix $Z = [Z_{i,j}]_{i,j=1}^{q} \in \mathbb{R}^{q \times q}$ is the vectorization of the lower triangular part of $Z$ given by

$$\text{vech}(Z) = [Z_{1,1}, \ldots, Z_{q,1}, Z_{2,2}, \ldots, Z_{q,2}, \ldots, Z_{q-1,q-1}, Z_{q,q-1}, Z_{q,q}]^{\top}.$$

By introducing vectorization and half-vectorization of $L$, we reduce the minimization of $J(\mathbf{u}, L, D)$ with respect to matrix $L$ into minimization with respect to the vector $\text{vech}(L)$. To tackle the constraint set $\mathcal{L}$, we can reduce it to a simpler form by using half-vectorization and vectorization of $L$ given by $\text{vech}(L) \in \mathbb{R}^{\frac{p(p+1)}{2}}$ and $\text{vec}(L) \in \mathbb{R}^{p^2}$, respectively. The following relations are used to relate $\text{vech}(L)$ and $\text{vec}(L)$ using an appropriate transformation matrix called duplication matrix $\mathcal{M}$:

$$\mathcal{M}\text{vech}(L) = \text{vec}(L), \text{ where } \mathcal{M} \in \mathbb{R}^{p^2 \times \frac{p(p+1)}{2}}.$$

The constraint set $\mathcal{L}$ can then be rewritten as $A\,\text{vech}(L) = 0, B\,\text{vech}(L) \leq 0$, where $A$ and $B$ are matrices which handle $L\mathbf{1} = 0$ and $L_{i,j} \leq 0, i \neq j$, respectively (Dong et al., 2016; Pu et al., 2021b). The construction and properties of $A, B$ and $\mathcal{M}$ can be found in Appendix A.5. For notational simplicity, we consider a slight abuse of notations when referring to function $J(\mathbf{u}, L, D)$ to be equivalent to $J(\mathbf{u}, \text{vech}(L), \text{vech}(D))$ as $\text{vech}(L)$ and $\text{vech}(D)$ can be used to represent $L$ and $D$, respectively. Similarly, when considering fixed $\mathbf{u}, \text{vech}(L)$ or $\text{vech}(D)$, we denote the function $J$ as a function of non-fixed entities, without referring to the fixed variables. For example, in the current step, since $\text{vech}(L)$ is the non-fixed entity and $\mathbf{u}, \text{vech}(D)$ are fixed, we denote $J(\mathbf{u}, \text{vech}(L), \text{vech}(D))$ simply as $J_{\mathbf{u}, D}(\text{vech}(L))$. We employ a projected gradient descent procedure to solve $\min J_{\mathbf{u}, D}(\text{vech}(L))$. The $(k + 1)$-th iterate $\text{vech}(L)^{k+1}$ is obtained from the $k$-th iterate $\text{vech}(L)^k$ by the following projected gradient descent step :

$$\text{vech}(L)^{k+1} = \Pi_{\mathcal{L}}(\text{vech}(L)^k - \eta_L \nabla_{\text{vech}(L)} J_{\mathbf{u}, D}(\text{vech}(L)^k)), \tag{12}$$

where $\eta_L > 0$ is the learning rate for the descent step, $\Pi_{\mathcal{L}}$ denotes the projection operator onto the set $\mathcal{L}$. The expression for gradient term $\nabla_{\text{vech}(L)} J$ has been derived in Appendix A.6. For a fixed $\hat{z} \in \mathbb{R}^{\frac{p(p+1)}{2}}$, the projection operator $\Pi_{\mathcal{L}}$ is defined as follows:

$$\Pi_{\mathcal{L}}(\hat{z}) = \underset{z \in \mathbb{R}^{\frac{p(p+1)}{2}}}{\arg \min} \|z - \hat{z}\|^2 \text{ such that } Az = 0, Bz \leq 0. \tag{13}$$

The projection operator defined in (13) ensures that $\text{vech}(L)$ obtained satisfies the constraints of a Laplacian matrix of a graph. Projection operator $\Pi_{\mathcal{L}}$ is evaluated by solving the quadratic program (13) using well-known interior point methods (Dikin, 1967; Andersen et al., 2012).

For a large matrix $L$, capturing meaningful relationships between input functions becomes important; otherwise, elements of $L$ may contain many non-zero values which are close to each other in magnitude and may lead to lack of interpretability. Integrating sparsity-inducing regularizers on $L$ would ensure that the most important interactions get captured and can improve the predictions for output function. The learned sparse graphs would then become useful for interpretation. In Section 5.1.4, we discuss about incorporating sparsity-inducing regularizers in the $L$-based minimization problem $\min J_{\mathbf{u}, D}(\text{vech}(L))$. Now that projected gradient descent is proposed for minimization of $J$ with respect to $L$ for a fixed $\mathbf{u}$ and $D$, we proceed with a similar approach for the next step of the alternating minimization framework.

### 5.1.3 Minimization with respect to $D$ for fixed $\mathbf{u}, L$

Similar to the minimization of $J(\mathbf{u}, L, D)$ with respect to $L$ for fixed $\mathbf{u}, D$, as discussed in the previous section, we proceed with the simplification of $D \in \mathcal{D}$ using half-vectorization given as $\text{vech}(D) \in \mathbb{R}^{\frac{p(p+1)}{2}}$ to solve $\min J_{\mathbf{u}, L}(\text{vech}(D))$. We further use notation $Diag(D)$ to denote the vector containing diagonal elements of

$D$. In order to deal with the constraint $D_{i,i} \geq 0, \forall i \in [p]$ of $D$, we construct a matrix $C \in \mathbb{R}^{p \times \frac{p(p+1)}{2}}$ which consists of 0's and 1's satisfying $C\text{vech}(D) = Diag(D)$. Construction of a suitable $C$ has been discussed in Appendix A.5. For solving $\min J_{\mathbf{u},L}(\text{vech}(D))$, we use projected gradient descent steps given by:

$$\text{vech}(D)^{k+1} = \Pi_{\mathcal{D}}(\text{vech}(D)^k - \eta_D \nabla_{\text{vech}(D)} J(\text{vech}(D)^k)), \tag{14}$$

where $\eta_D > 0$ is learning rate for the descent step, $\Pi_{\mathcal{D}}$ denotes the projection operator onto the set $\mathcal{D}$, and $\text{vech}(D)^k$ denotes the $k$-th iterate for $\text{vech}(D)$. The required expression for gradient $\nabla_{\text{vech}(D)} J$ has been derived in Appendix A.6. The projection operator $\Pi_{\mathcal{D}}$ is defined for $\hat{d} \in \mathbb{R}^{\frac{p(p+1)}{2}}$ as

$$\Pi_{\mathcal{D}}(\hat{d}) = \underset{d \in \mathbb{R}^{\frac{p(p+1)}{2}}}{\arg\min} \ \|d - \hat{d}\|^2, \text{ such that } Cd \geq 0. \tag{15}$$

The explicit solution of (15) can be easily obtained for $i \in [p(p+1)/2]$, as the following:

$$d_i = \begin{cases} 0, & \text{if } i \in [p(p+1)/2] \setminus \mathcal{J} \\ \max(\hat{d}_i, 0), & \text{otherwise,} \end{cases} \tag{16}$$

where the set $\mathcal{J}$ consists of indices $j$ such that $(\text{vech}(C))_j = 1$ (see Appendix A.5).

Thus, the proposed alternating minimization framework discussed in Sections 5.1.1-5.1.3 can be summarized in the following steps:

1. **Minimization with respect to** $F \in \mathcal{H}_K$ (or $\mathbf{u} \in \mathcal{Y}^n$): Solving for $\mathbf{u}$ in $(\mathbf{K} + \lambda I)\mathbf{u} = \mathbf{y}$.

2. **Minimization with respect to** $\text{vech}(L)$: Projected gradient descent of $J$ with respect to $\text{vech}(L)$ such that $L \in \mathcal{L}$.

3. **Minimization with respect to** $\text{vech}(D)$: Projected gradient descent of $J$ with respect to $\text{vech}(D)$ such that $D \in \mathcal{D}$.

### 5.1.4 Sparsity Inducing Regularization

The functional regression problem aims at predicting the output function based on the interactions and influences of the input functions. In a large-scale setting with numerous input functions influencing the output function, providing each interaction of a pair of input functions with similar weights may hamper the prediction capability as well as interpretability of the proposed graph-induced operator-valued kernel method. Sparse graphs ensure a focus on picking more pivotal associations between the pairs of input functions which drive the functional regression problem, as noisy interactions are given negligible weights and significant interactions contribute more to the prediction. A popular method to obtain sparsity on the graph structure is to consider the trace of $L$ as a regularizer (Qiao et al., 2018). But as $L$ is obtained based on solving a constrained minimization problem (13), we introduce a constraint that can regularize the values in $L$ (Pu et al., 2021b). To facilitate this, we consider a vector $c \in \mathbb{R}^{\frac{p(p+1)}{2}}$ consisting of 0's and 1's, such that $c^\top \text{vech}(L) = \text{trace}(L)$ (Appendix A.5). Let $m_{\text{trace}}(> 0)$ be a hyperparameter controlling the trace of $L$ which is equivalent to controlling the off-diagonal entries of weight matrix $W$ corresponding to the graph learned. As our formulation requires solving a quadratic program in the projected gradient descent for $L$ (Section 5.1.2), a direct way to control the trace of $L$ involves modifying the quadratic program (13) to the following:

$$\Pi_{\mathcal{L}}(\hat{z}) = \underset{z \in \mathbb{R}^{\frac{p(p+1)}{2}}}{\arg\min} \ \|z - \hat{z}\|^2, \text{ such that } Az = 0, c^\top z = m_{\text{trace}}, Bz \leq 0,$$

where $\hat{z} \in \mathbb{R}^{\frac{p(p+1)}{2}}$ and $m_{\text{trace}} > 0$. The constraint $c^\top z = m_{\text{trace}}$ is appended with $Az = 0$ to obtain the transformed quadratic program for projection operator $\Pi_{\mathcal{L}}$:

$$\Pi_{\mathcal{L}}(\hat{z}) = \underset{z \in \mathbb{R}^{\frac{p(p+1)}{2}}}{\arg\min} \ \|z - \hat{z}\|^2, \text{ such that } \tilde{A}z = \tilde{0}, Bz \leq 0, \tag{17}$$

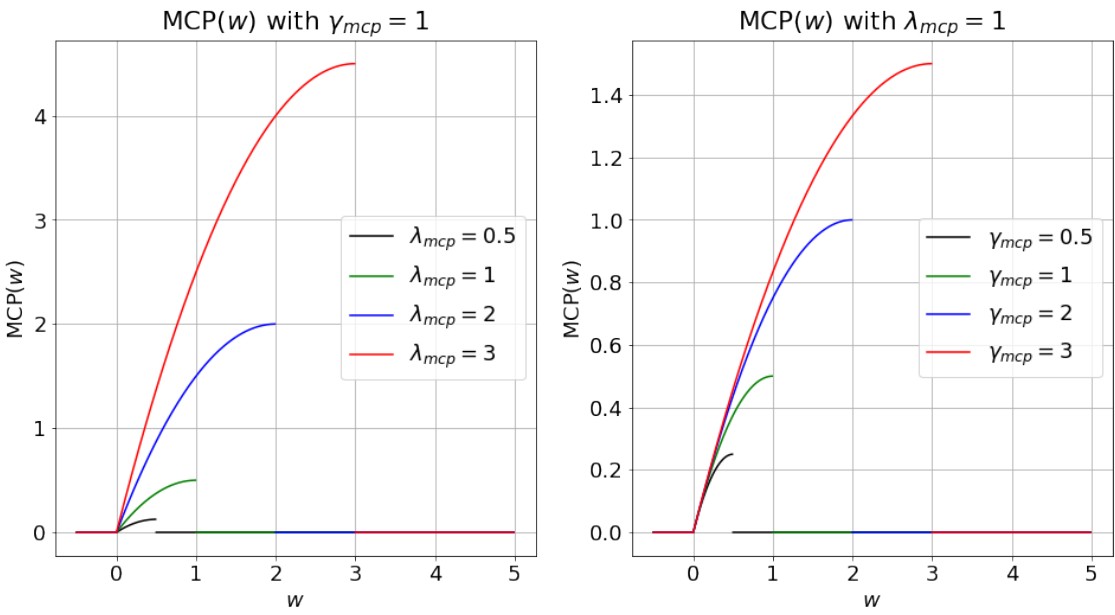

Figure 2: Comparison of $h(w) \coloneqq \mathrm{MCP}(w)$ values for different $\lambda_{mcp}$ and $\gamma_{mcp}$ values for varying $w$. [Best viewed in color]

where $\tilde{A} = \begin{bmatrix} A \\ c^\top \end{bmatrix}$ and $\tilde{0} = \begin{bmatrix} 0 \\ m_{\mathrm{trace}} \end{bmatrix}$. For our experiments the value of $m_{\mathrm{trace}}$ is considered as $p/2$ (discussed in Section 7) to promote sparsity in the learned graph structure. Assigning $m_{\mathrm{trace}}$ the value $p/2$, allows a cumulative weight of $p/2$ to be distributed among the edges in the learned graph, promoting sparsity. Though we have used $m_{trace} = p/2$ for all our experiments, we showcase the impact of different values of $m_{trace}$ in Appendix A.10.5 which illustrate that $m_{trace}$ can be considered as a hyperparameter.

In addition to restricting the trace of $L$ with $m_{\mathrm{trace}}$, we also utilize a sparsity-inducing regularizer in $\min_{\mathbf{u}, D} J(\mathrm{vech}(L))$. Nonconvex regularizers have gained popularity in recent literature (Zhang et al., 2020; Ying et al., 2020; Vargas Vieyra, 2022) for promoting sparsity in learned graphs. These nonconvex regularizers acting on weights on individual graph edges are more effective than traditional $\ell_1$-norm based graphical lasso regularization which is a classical approach in graph learning (Yuan & Lin, 2007; Banerjee et al., 2008; d'Aspremont et al., 2008). A nonconvex regularizer, minimax concave penalty (MCP) (Zhang, 2010) denoted by $h$ is characterized by the definition of its derivative $h'$ given by:

$$h'(w) = \begin{cases} \lambda_{mcp} - \frac{w}{\gamma_{mcp}}, & w \in [0, \gamma_{mcp}\lambda_{mcp}], \\ 0, & w \in [\gamma_{mcp}\lambda_{mcp}, \infty), \end{cases} \tag{18}$$

for $\lambda_{mcp}, \gamma_{mcp} > 0$. The above definition of $h'$ with $h(0) = 0$ ensures that $h$ is monotonically increasing in the interval $[0, \gamma_{mcp}\lambda_{mcp}]$. The impact of $\lambda_{mcp}$ and $\gamma_{mcp}$ on value of the regularizer $h$ has been compared in Figure 2. The illustration in Figure 2 shows that MCP function $h$ magnifies the small values till $\gamma_{mcp}\lambda_{mcp}$ where $\lambda_{mcp}$ and $\gamma_{mcp}$ control the slope and curvature of the truncated concave quadratic function $h$. This ensures that most smaller off-diagonal values are penalized by choosing proper $\lambda_{mcp}$ and the differences between large and small values are exaggerated which promotes sparsity. Though $h'$ in (18) (or equivalently $h$) is defined over $\mathbb{R}_+$, we overload the notation to define $h(z) = [h(z)_i]_{i=1}^d$ where $z \in \mathbb{R}_+^d$. Ying et al. (2020) proposes MCP to obtain a nonconvex penalized maximum likelihood estimation method for learning sparse Laplacian matrices for graphs. However, here we introduce MCP regularizer in the projection step considered for $\mathrm{vech}(L)$. Therefore, we reformulate the minimization problem with respect to $L$ for fixed $\mathbf{u}$

and $D$ as follows:

$$\min_{\substack{\text{vech}(L)\in\tilde{\mathcal{L}} \\ \text{vech}(L)\triangleright\mathcal{MCP}}} J_{\mathbf{u},D}(\text{vech}(L)), \tag{19}$$

where the constraint set is $\tilde{\mathcal{L}} = \left\{ z \in \mathbb{R}^{\frac{p(p+1)}{2}} \Big| \tilde{A}z = \tilde{0}, Bz \leq 0 \right\}$ and the property $\text{vech}(L) \triangleright \mathcal{MCP}$ signifies that $\text{vech}(L)$ is sparse in the sense of MCP regularization.

The MCP regularizer operates on the off-diagonal entries of $L$ to induce sparsity. To solve (19), we consider a two-level update procedure for $\text{vech}(L)$. In the first level, we use $J_{\mathbf{u},D}(\text{vech}(L))$, to perform an iteration of gradient descent to obtain $(q+1)$-th iterate from $q$-th iterate, as follows:

$$\text{vech}(L)^{q+1} = \text{vech}(L)^q - \eta_L \nabla_{\text{vech}(L)} J_{\mathbf{u},D}(\text{vech}(L)^q), \tag{20}$$

where $\nabla_{\text{vech}(L)} J$ denotes the gradient of $J$ with respect to $\text{vech}(L)$ and $\eta_L > 0$ is the learning rate for the descent step. In the second level, we ensure that $\text{vech}(L)^{q+1}$ obtained in (20) is projected onto set $\tilde{\mathcal{L}}$ and is also sparse in the sense of MCP regularization captured using the property $\text{vech}(L) \triangleright \mathcal{MCP}$. To apply the MCP regularizer on off-diagonal entries of $L$ for $\text{vech}(L)$, we construct a matrix $H \in \mathbb{R}^{\frac{p(p+1)}{2} \times \frac{p(p+1)}{2}}$ comprising 0's and 1's such that $H\text{vech}(L)$ produces a vector having the same structure as $\text{vech}(L)$ with 0's corresponding to diagonal entries of $L$ and the same off-diagonal entries as in $L$ (see Appendix A.5 for the construction). Then we aim to solve the following minimization problem:

$$\min_{z \in \mathbb{R}^{\frac{p(p+1)}{2}}} \|z - \text{vech}(L)^{q+1}\|^2 + \sum_{i=1}^{\frac{p(p+1)}{2}} (h(H(-z)))_i, \text{ such that } \tilde{A}z = \tilde{0}, Bz \leq 0, \tag{21}$$

where the negative sign before $z$ in MCP regularization is used since $h$ operates on non-negative entries and $L$ contains negative weights corresponding to off-diagonal entries. Further the product $H(-z)$ eliminates the positive diagonal entries of $L$. Note that the first term in the objective function of (21) is convex in $z$ and the second term is concave in $z$. Hence we adopt a majorization-minimization iterative approach, similar to (Ying et al., 2020) to solve (21). In each iteration $l$ of this approach, we obtain a majorization of the objective function in (21) using the linearization of concave function $h$ at $z^l$ as:

$$\|z - \text{vech}(L)^{q+1}\|^2 + \sum_{i=1}^{\frac{p(p+1)}{2}} (h'(H(-z^l)))_i (H(-z))_i. \tag{22}$$

Then we solve a minimization step, which when combined with the linearization in (22) leads to the following quadratic program:

$$z^{l+1} = \arg\min_{z \in \mathbb{R}^{\frac{p(p+1)}{2}}} \|z - \text{vech}(L)^{q+1}\|^2 + \sum_{i=1}^{\frac{p(p+1)}{2}} (h'(H(-z^l)))_i (H(-z))_i, \text{ such that } \tilde{A}z = \tilde{0}, Bz \leq 0. \tag{23}$$

We solve the problem (23) iteratively till convergence to obtain a sequence of $z^l$'s. The procedure for MCP based sparsity-inducing regularization has been summarized in Algorithm 1. The solution obtained from Algorithm 1 is thus a valid Laplacian matrix of a graph, and satisfies trace constraint as well as MCP based sparsity-inducing regularization property captured by $\text{vech}(L) \triangleright \mathcal{MCP}$. Note that other nonconvex regularizers such as smoothly clipped absolute deviation (SCAD) (Fan & Li, 2001; Ying et al., 2020) can also be used instead of MCP in our approach which can provide comparable sparse graph structures.

Using the learned $\mathbf{u}, L, D$, the graph-induced operator-valued kernel is used to predict the output function corresponding to the $p$ input functions $\hat{x} = (\hat{x}_1, \hat{x}_2, \ldots, \hat{x}_p)$ based on (114) given by

$$F(\hat{x}) = \sum_{i=1}^{n} K(x^{(i)}, \hat{x}) u_i, \text{ where } u_i \in \mathcal{Y}. \tag{24}$$

---

**Algorithm 1 MCP Regularization of $L$**

---

    **Input:** vech$(L)$
    **Output:** vech$(L_{mcp})$
    **Initialize** $z^0 = $ vech$(L), H$
    $l = 0$
    **while** stopping criteria for $z^l$ not satisfied **do**
        Find $h'(H(-z^l))$
        Find $z^{l+1}$ as the solution of (23)
        $l \leftarrow l + 1$
    **end while**
    vech$(L_{mcp}) = z^l$

---

Algorithm 2 summarizes the entire alternating minimization procedure with sparsity-inducing regularization. For our experiments, vech$(L)$ is initialized using a Laplacian matrix $L$ with $L_{i,j} = -m_{\text{trace}}/(p(p-1))$, for $i \neq j \in [p]$ and $L_{i,i} = m_{\text{trace}}/p$, for $i \in [p]$, which satisfies the $m_{\text{trace}}$ condition in (17). The initial vech$(D)$ is considered corresponding to $D = I_p$, identity matrix of order $p$. Note that the objective function $J$ in problem (19) is not jointly convex with respect to $\mathbf{u}$, vech$(L)$ and vech$(D)$. To prove the convergence of alternating minimization of a non-convex function $J$ with respect to a heterogeneous collection of three variables where $u_i$'s $\in \mathcal{Y}$ and vech$(L),$ vech$(D) \in \mathbb{R}^{\frac{p(p+1)}{2}}$ requires development of fundamental results which is out of scope of our current work, and we aim to take this up in future. However, we observed empirical convergence of the proposed alternating minimization framework in our experiments.

### 5.1.5 Extension of Learning Graph Structure for Multi-dimensional Outputs

In many scenarios, a functional regression problem involves multi-dimensional input and output functions which requires learning multiple functions corresponding to each dimension in the output space. Consider $\mathcal{X} = (L^2([0,1]))^p, \mathcal{Y} = L^2([0,1])$ with $\{(\mathbf{x}^{(i)}, \mathbf{y}^{(i)})\}_{i=1}^n$ as the training data, where $\mathbf{x}^{(i)} \in \mathcal{X}^r, \mathbf{y}^{(i)} \in \mathcal{Y}^s$, for $i \in [n]$. Notice that the input space is now $\mathcal{X}^r$ and output space is $\mathcal{Y}^s$, which are both multi-dimensional. In this case, we have $\mathbf{x}^{(i)} = (x_1^{(i)}, x_2^{(i)}, \ldots, x_r^{(i)})$ where $x_j^{(i)} = (x_{j1}^{(i)}, x_{j2}^{(i)}, \ldots, x_{jp}^{(i)}) \in \mathcal{X}, \forall j \in [r]$ and $\mathbf{y}^{(i)} = (y_1^{(i)}, y_2^{(i)}, \ldots, y_s^{(i)})$ where $y_j^{(i)} \in \mathcal{Y}, \forall j \in [s]$. Consider a setting where the input space $\mathcal{X}^r$ is mapped to output space $\mathcal{Y}^s$ by learning distinct maps of the from $F^i : \mathcal{X}^r \to \mathcal{Y}$, for $i \in [s]$. Note that to find these maps $F^i$, an extension to the framework developed in Sections A.1 and 5.1 can be used, as long as the proposed graph-induced OVK framework is adapted to handle the multi-dimensional inputs $\mathbf{x}^{(i)}$. For a motivating example, consider the data of movement of players for a fixed time interval in a basketball game comprising the $\mathbf{x}$ and $\mathbf{y}$ coordinates of the playing area belonging to the input space $\mathcal{X}^2$. Similarly, movement of the ball for the fixed time interval is characterized by $\mathbf{x}$ and $\mathbf{y}$ coordinates of the court which belong to the output space $\mathcal{Y}^2$. The corresponding regression problem involves learning $F = [F^1, F^2]^\top$, where $F^1 : \mathcal{X}^2 \mapsto \mathcal{Y}$ and $F^2 : \mathcal{X}^2 \mapsto \mathcal{Y}$.

For the multi-dimensional setting, we propose the following extension of the scalar-valued kernel based on $L$ and $D$ as

$$k_1\left(\mathbf{x}^{(i)}, \mathbf{x}^{(j)}; G\right) = e^{-\sum_{k=1}^r \left[\gamma_k \left(x_k^{(i)} - x_k^{(j)}\right)^\top (L+D) \left(x_k^{(i)} - x_k^{(j)}\right)\right]}, \tag{25}$$

where $\gamma_k > 0, \forall k \in [r]$ and $k_1$ maps pair of elements from the input space $\mathcal{X}^r$ to a real number. Now, $k_1\left(\mathbf{x}^{(i)}, \mathbf{x}^{(j)}; G\right)$ can be appropriately used to create a graph-based operator-valued kernel in higher dimensions using the construction of a suitable OVK as follows:

$$\begin{bmatrix} K^1\left(\mathbf{x}^{(i)}, \mathbf{x}^{(j)}\right) y_1(t) \\ \vdots \\ K^s\left(\mathbf{x}^{(i)}, \mathbf{x}^{(j)}\right) y_s(t) \end{bmatrix} = k_1\left(\mathbf{x}^{(i)}, \mathbf{x}^{(j)}; G\right) \begin{bmatrix} \int_a^b k_2^1(t', t) y_1(t') dt' \\ \vdots \\ \int_a^b k_2^s(t', t) y_s(t') dt' \end{bmatrix}, \tag{26}$$

---

**Algorithm 2 Alternating Minimization of $J$**

---

**Input:** $\{(x^{(i)}, y^{(i)})\}_{i=1}^n, x^{(i)} = (x_1^{(i)}, x_2^{(i)}, \ldots, x_p^{(i)}) \in \mathcal{X}, y^{(i)} \in \mathcal{Y}$
**Output:** $\mathbf{u}, \text{vech}(L), \text{vech}(D)$
**Initialize** $\text{vech}(L)^0 \in \mathcal{L}, \text{vech}(D)^0 \in \mathcal{D}$
$\mathbf{y} = [y^{(1)}, y^{(2)}, \ldots, y^{(n)}]^\top$
$k = 0$
**while** True **do**
    Compute $k_1(x^{(i)}, x^{(j)}; G) = e^{-\gamma R_{ij}(\text{vech}(L)^k + \text{vech}(D)^k)}, \forall i, j \in [n]$ using (137) in Appendix A.6
    Solve for $\mathbf{u}$ in $(\mathbf{K} + \lambda I)\mathbf{u} = \mathbf{y}$ to obtain $\mathbf{u}^k$ using OpMINRES
    **if** stopping criterion for $\mathbf{u}^k$ is satisfied **then**
        break from the outermost while loop                    //exit based on convergence of **u** iterates
    **end**
    $q = 0$
    **while** stopping criterion for $\text{vech}(L)^q$ not satisfied **do**
        //stopping criterion is based on convergence of $\text{vech}(L)$ iterates
        $\text{vech}(L)^{q+1} = \text{vech}(L)^q - \eta_L \nabla_{\text{vech}(L)} J_{\mathbf{u},D}(\text{vech}(L)^q)$
        $\text{vech}(L)_{mcp}$ obtained based on MCP regularization using $\text{vech}(L)^{q+1}$ as input to Algorithm 1
        $\text{vech}(L)^{q+1} \leftarrow \text{vech}(L)_{mcp}$
        $q \leftarrow q + 1$
    **end while**
    $\text{vech}(L)^k \leftarrow \text{vech}(L)^q$
    $m = 0$
    **while** stopping criterion for $\text{vech}(D)^m$ not satisfied **do**
        //stopping criterion is based on convergence of $\text{vech}(D)$ iterates
        $\text{vech}(D)^m = \text{vech}(D)^k$
        $\text{vech}(D)^{m+1} = \Pi_{\mathcal{D}}(\text{vech}(D)^m - \eta_D \nabla_{\text{vech}(D)} J_{\mathbf{u},L}(\text{vech}(D)^m))$ using (14)
        $m \leftarrow m + 1$
    **end while**
    $\text{vech}(D)^k \leftarrow \text{vech}(D)^m$
    $k \leftarrow k + 1$
**end while**
$\mathbf{u} = \mathbf{u}^k, \text{vech}(L) = \text{vech}(L)^k, \text{vech}(D) = \text{vech}(D)^k$

---

where $k_2^i : \mathbb{R} \times \mathbb{R} \to \mathbb{R}$, for $i \in [s]$ are scalar-valued kernels on $\mathbb{R}^2$. In order to use the output functions $\mathbf{y}^{(i)} = (y_1^{(i)}, y_2^{(i)}, \ldots, y_s^{(i)})$ for learning $s$ maps from input space $\mathcal{X}^r$ to the output space $\mathcal{Y}$, the problem in (3) is extended as follows:

$$
\widetilde{F}, \widetilde{L}, \widetilde{D} = \underset{F=[F^1, F^2, \ldots, F^s]^\top \in \mathcal{H}_K, L \in \mathcal{L}, D \in \mathcal{D}}{\arg\min} \sum_{l=1}^s \left[ \sum_{i=1}^n \|y_l^{(i)} - F^l(\mathbf{x}^{(i)})\|_{\mathcal{Y}}^2 + \lambda \|F^l\|_{\mathcal{H}_K^l}^2 \right] \\
+ \sum_{k=1}^r \rho_k x_k^{(i)\top} L x_k^{(i)} + \rho_D \|D\|_F^2,
\tag{27}
$$

where $\lambda, \rho_k, \rho_D$ are positive reals, $\mathcal{H}_K = \mathcal{H}_K^1 \times \mathcal{H}_K^2 \times \cdots \times \mathcal{H}_K^s, F^l : \mathcal{X}^r \to \mathcal{Y}$ for $l \in [s]$. Consider the objective function of (27) as $J(F, L, D)$. Similar to Sections 5.1.1-5.1.4, on applying an alternating minimization framework, the steps involved in $L, D$ minimization remain the same. In order to solve the minimization problem in (27) (for fixed $L, D$) an extension of the representer theorem A.2 is required which follows based on the construction of graph-induced operator-valued kernel in (26).

**Theorem 5.1 (Extended representer theorem).** Let $K$ be an operator-valued kernel as defined in (26) and $\mathcal{H}_K = \mathcal{H}_K^1 \times \mathcal{H}_K^2 \times \cdots \times \mathcal{H}_K^s$ be its corresponding function-valued reproducing kernel Hilbert space based

on kernels $k_1, k_2^1, \ldots, k_2^s$. The solution $\widetilde{F}_\lambda \in \mathcal{H}_K$ of the regularized optimization problem:

$$\widetilde{F}_\lambda = \underset{F=[F^1, F^2, \ldots, F^s]^\top \in \mathcal{H}_K}{\arg\min} \sum_{l=1}^{s} \left( \sum_{i=1}^{n} \|y_l^{(i)} - F^l(\mathbf{x}^{(i)})\|_{\mathcal{Y}}^2 + \lambda \|F^l\|_{\mathcal{H}_K^l}^2 \right),$$

where $\lambda > 0, F = [F^1, F^2, \ldots, F^s]^\top \in \mathcal{H}_K = \mathcal{H}_K^1 \times \mathcal{H}_K^2 \times \cdots \times \mathcal{H}_K^s$, has the following form

$$\widetilde{F}_\lambda(.) = \begin{bmatrix} \widetilde{F}_\lambda^1(.) \\ \vdots \\ \widetilde{F}_\lambda^s(.) \end{bmatrix} = \begin{bmatrix} \sum_{i=1}^{n} K^1(\mathbf{x}^{(i)}, .)u_i^1 \\ \vdots \\ \sum_{i=1}^{n} K^s(\mathbf{x}^{(i)}, .)u_i^s \end{bmatrix}, \text{ where } u_i^1, u_i^2, \ldots, u_i^s \in \mathcal{Y}. \tag{28}$$

*Proof.* The proof follows as a consequence of the representer theorem proof in Appendix A.4. $\qquad\square$

In order to solve the minimization problem (27), we use the representer theorem and reproducibility property of the OVKs $K^l$, for $l \in [s]$. The optimization problem in (27) is solved by using the alternating minimization of the objective function with respect to $F = [F^1, F^2, \ldots, F^s]^\top \in \mathcal{H}_K, L \in \mathcal{L}$ and $D \in \mathcal{D}$. For a constant $L$ and $D$, we use the representer theorem (Theorem 5.1) to transform the objective function in terms of $F^1 \in \mathcal{H}_K^1, \ldots, F^s \in \mathcal{H}_K^s$ to functions $u_i^1, u_i^2, \ldots, u_i^s \in \mathcal{Y}, i \in [n]$, respectively. The objective function $J$ is defined as the following (see Appendix A.2):

$$J(F^1, F^2, \ldots, F^s, L, D) = \sum_{l=1}^{s} \left( \sum_{i=1}^{n} \|y_l^{(i)} - F^l(\mathbf{x}^{(i)})\|_{\mathcal{Y}}^2 + \lambda \|F^l\|_{\mathcal{H}_K^l}^2 \right) + \sum_{k=1}^{r} \rho_k \left( \sum_{i=1}^{n} x_k^{(i)\top} L x_k^{(i)} \right)$$

$$+ \rho_D \|D\|_F^2$$

$$\implies J(\mathbf{u}^1, \mathbf{u}^2, \ldots, \mathbf{u}^s, L, D) = \sum_{l=1}^{s} \left( \sum_{i=1}^{n} \left\| y_l^{(i)} - \sum_{j=1}^{n} K^l(\mathbf{x}^{(i)}, \mathbf{x}^{(j)})u_j^l \right\|_{\mathcal{Y}}^2 + \lambda \sum_{i,j=1}^{n} \langle K^l(\mathbf{x}^{(i)}, \mathbf{x}^{(j)})u_i^l, u_j^l \rangle_{\mathcal{Y}} \right) \tag{29}$$

$$+ \sum_{k=1}^{r} \left( \rho_k \sum_{i=1}^{n} x_k^{(i)\top} L x_k^{(i)} \right) + \rho_D \|D\|_F^2.$$

For solving the multi-dimensional functional regression problem, the alternating minimization framework discussed in Section 5.1 is extended with the major difference in the step concerning minimization with respect to $F = [F^1, F^2, \ldots, F^s]^\top$ (or $\mathbf{u}^1, \mathbf{u}^2, \ldots, \mathbf{u}^s$) for fixed $L$ and $D$. The multi-dimensionality leads to solving the following system of linear operator equations:

$$\left[ (\mathbf{K}^1 + \lambda I)\mathbf{u}^1 \quad \ldots \quad (\mathbf{K}^s + \lambda I)\mathbf{u}^s \right] = \begin{bmatrix} \Theta_1 & \ldots & \Theta_s \end{bmatrix}, \tag{30}$$

where $\mathbf{K}_{i,j}^l u = K^l(\mathbf{x}^{(i)}, \mathbf{x}^{(j)})u = k_1(\mathbf{x}^{(i)}, \mathbf{x}^{(j)}; G)\bar{k}_2^k(u), \forall u \in \mathcal{Y}, \Theta_l = [y_l^{(1)}, y_l^{(2)}, \ldots, y_l^{(n)}]^\top$ with $\bar{k}_2^l = \int_0^1 e^{-\gamma_{op}^l |t'-t|} u(t')dt', \gamma_{op}^l > 0, \forall l \in [s]$ and $t', t \in [0, 1]$. For all $K^l, l \in [s]$, the scalar-valued kernel $k_1$ (given in (25)) used remains the same and is built on common $L$ and $D$. Therefore, $s$ possibly different graph-induced OVKs are obtained by using exponential kernels on $\mathbb{R}^2$ with $\gamma_{op}^l$, for $l \in [s]$.

In order to solve for $\mathbf{u}^1, \mathbf{u}^2, \ldots, \mathbf{u}^s$ in (30), we use the OpMINRES algorithm discussed in Section 5.1.1 to solve the systems $(\mathbf{K}^l + \lambda I)\mathbf{u} = \mathbf{y}_l$, for $l \in [s]$. OpMINRES algorithm solves the $s$ systems in parallel with a stopping criteria which combines $s$ relative residuals.

Thus, for alternating minimization, the steps discussed can be summarized as:

1. **Minimization with respect to** $F = [F^1, F^2, \ldots, F^s]^\top \in \mathcal{H}_K$ (or $\mathbf{u}^1, \mathbf{u}^2, \ldots, \mathbf{u}^s \in \mathcal{Y}^n$): Solving for $\mathbf{u}^1, \mathbf{u}^2, \ldots, \mathbf{u}^s$ in

$$\left[ (\mathbf{K}^1 + \lambda I)\mathbf{u}^1 \quad \ldots \quad (\mathbf{K}^s + \lambda I)\mathbf{u}^s \right] = \begin{bmatrix} \Theta_1 & \ldots & \Theta_s \end{bmatrix}.$$

2. **Minimization with respect to** vech($L$): Projected gradient descent of $J$ with respect to vech($L$) in $\mathcal{L}$ with sparsity inducing regularization.

3. **Minimization with respect to** vech($D$): Projected gradient descent of $J$ with respect to vech($D$) in $\mathcal{D}$.

Using the learned graph-induced operator-valued kernel the output function is used to predict for input functions $\hat{\mathbf{x}} = (\hat{x}_1, \ldots, \hat{x}_r)$, where $\hat{x}_j \in \mathcal{X}$, for $j \in [r]$ as

$$
F(\hat{\mathbf{x}}) = \begin{bmatrix} \sum_{i=1}^n K^1(\mathbf{x}^{(i)}, \hat{\mathbf{x}}) u_i^1 \\ \vdots \\ \sum_{i=1}^n K^s(\mathbf{x}^{(i)}, \hat{\mathbf{x}}) u_i^s \end{bmatrix}, \text{ where } u_i^1, \ldots, u_i^s \in \mathcal{Y}, \text{ for } i \in [s].
$$

### 5.1.6 Sample-based Approximation for Functional Regression Problem

The kernel-based alternating minimization framework proposed earlier in this section helps to learn appropriate $\mathbf{u}, L$, and $D$ for the prediction of functional output using (24). For a setting where the number of training samples is large, the training can become computationally expensive as OpMINRES iteration scales in $O(n^3)$, where $n$ is the number of training samples. This issue of scalability is a well-known problem in kernel methods, which becomes more pronounced in an OVK-based framework. There are many popular methods for handling scalability issues in kernel methods (Williams & Seeger, 2000; Meanti et al., 2020; Bach & Jordan, 2005). In most cases, the approaches handling scalability issues for kernel methods are incorporated into the learning problem for approximating the kernel Gram matrices arising in large datasets, by using low-rank Cholesky decomposition (Bach & Jordan, 2005), Nyström approximation (Williams & Seeger, 2000) and GPU-based acceleration and parallelization (Meanti et al., 2020). For vector-valued regression problems, random Fourier features have been used for building OVKs (Brault et al., 2016; Brault, 2017) which cannot be directly extended to functional regression problems due to the following reasons. Extension of the random Fourier features to a functional setting requires developing a new theoretical framework for a functional version of operator-valued Bochner's theorem. Moreover, spectral decomposition of OVKs for functional data is beyond the scope of our current work, hence we leave it for future work. In our approach, we aim for a sample-based approximation heuristic algorithm which enables us to perform a greedy sample selection procedure followed by the training with only those selected samples.

The motivation of the sample-based approximation lies in characterizing the action of the considered OVK $K$ on $i$-th sample $(x^{(i)}, y^{(i)}) \in \mathcal{X} \times \mathcal{Y}$. Recall the learning problem discussed in Section 5.1 given by

$$
\widetilde{F}, \widetilde{L}, \widetilde{D} = \underset{F \in \mathcal{H}_K, L \in \mathcal{L}, D \in \mathcal{D}}{\arg\min} \sum_{i=1}^n \|y^{(i)} - F(x^{(i)})\|_{\mathcal{Y}}^2 + \lambda \|F\|_{\mathcal{H}_K}^2 + \rho_L \sum_{i=1}^n {x^{(i)}}^\top L x^{(i)} + \rho_D \|D\|_F^2,
$$

which requires solving for $\mathbf{u}$ in $(\mathbf{K} + \lambda I)\mathbf{u} = \mathbf{y}$ in the first step of alternating minimization with respect to $F$ (or $\mathbf{u}$) for fixed $L$ and $D$. We consider the notations $y_i$ and $K_i$ respectively as equivalent to $y^{(i)}$ and $K(x^{(i)}, \cdot)$ in this section for simplicity. Though approximating $u_i \in \mathcal{Y}$ in $F(.) = \sum_{i=1}^n K_i u_i$ corresponding to sample $(x^{(i)}, y^{(i)})$ may provide a better option for performing a sample-based approximation, we do not have the luxury to perform the inversion required in $(\mathbf{K} + \lambda I)\mathbf{u} = \mathbf{y}$. One way to assess the importance of a training sample $(x^{(i)}, y^{(i)})$ is to investigate the action of operator $K$ with $x^{(i)}$ on the output function $y^{(i)}$. Towards that we build $\bar{K}_i : \mathcal{X} \to \mathcal{L}(\mathcal{Y})$ by choosing samples which minimize the squared norm of the difference $\sum_{i=1}^n \|K_i y_i - \bar{K}_i y_i\|_{\mathcal{H}_K}^2$, defined over the RKHS $\mathcal{H}_K$. A working set of samples is constructed iteratively from the training data to formulate $\bar{K}_i y_i$ as a linear combination of $K_{i_j} y_{i_j}$'s, where $i_j$'s correspond to the indices of a working set of samples in training data. Inspired by (Smola & Schölkopf, 2000), we propose the following approach to construct $\bar{K}_i$'s iteratively. Consider indices in $I = \{i_1, i_2, \ldots, i_{|I|}\} \subset [n]$ as the set of indices for the working set $I_W = \{(x^{(i)}, y^{(i)}) : i \in I\}$ of samples from the training data. The aim is to

approximate the action of $K_i$ on $y_i$ using samples in $I_W$ as

$$\bar{K}_i y_i = \sum_{j=1}^{|I|} T_{i,j} K_{i_j} y_{i_j}, \text{ for } i \in [n], \tag{31}$$

$$\implies \begin{bmatrix} \bar{K}_1 y_1 \\ \bar{K}_2 y_2 \\ \vdots \\ \bar{K}_n y_n \end{bmatrix} = \begin{bmatrix} T_{1,1} & T_{1,2} & \dots & T_{1,|I|} \\ T_{2,1} & T_{2,2} & \dots & T_{2,|I|} \\ \vdots & \vdots & \ddots & \vdots \\ T_{n,1} & T_{n,2} & \dots & T_{n,|I|} \end{bmatrix} \begin{bmatrix} K_{i_1} y_{i_1} \\ K_{i_2} y_{i_2} \\ \vdots \\ K_{i_{|I|}} y_{i_{|I|}} \end{bmatrix} =: T \begin{bmatrix} K_{i_1} y_{i_1} \\ K_{i_2} y_{i_2} \\ \vdots \\ K_{i_{|I|}} y_{i_{|I|}} \end{bmatrix}, \tag{32}$$

where $T \in \mathbb{R}^{n \times |I|}$. The values $\left\| K_i y_i - \bar{K}_i y_i \right\|_{\mathcal{H}_K}$ are treated as residuals of the approximation for the $i$-th training sample. Approximations $\bar{K}_i$ corresponding to each sample $(x^{(i)}, y^{(i)})$, for $i \in [n]$, created by the working set $I_W$ of samples in (31) are bounded linear operators on the output space $\mathcal{Y}$. For every sample $(x^{(i)}, y^{(i)})$, where $i \in [n]$, the action of operator $K_i$ on $y_i$ is approximated by using scalars $T_{i,1}, T_{i,2}, \dots, T_{i,|I|}$ with $K_{i_j} y_{i_j} \in \mathcal{H}_K$, for $j \in [|I|]$. Minimization of $\sum_{i=1}^{n} \|K_i y_i - \bar{K}_i y_i\|_{\mathcal{H}_K}^2$ ensures that the working set of samples can characterize closely the impact of $K_i y_i$. The approximation of $K_i, \forall i \in [n]$ using the working set of samples is described next.

**Approximation of Operators using Samples:** Initially, suppose $I = \emptyset$ and let $i_1 \in [n]$ be the best candidate index. Then the index set $I$ is updated as $I = \{i_1\}$, and the working set $I_W$ contains only a single sample corresponding to the index $i_1 \in I$. Now the optimization problem is to determine $T_{1,1}, T_{2,1}, \dots, T_{n,1}$ in $\bar{K}_i y_i = T_{i,1} K_{i_1} y_{i_1}, \forall i \in [n]$ and is given by

$$\underset{T_{1,1}, T_{2,1}, \dots, T_{n,1}}{\arg\min} \sum_{i=1}^{n} \left\| K_i y_i - \bar{K}_i y_i \right\|_{\mathcal{H}_K}^2 = \sum_{i=1}^{n} \left\| K_i y_i - T_{i,1} K_{i_1} y_{i_1} \right\|_{\mathcal{H}_K}^2. \tag{33}$$

The solution of (33) is obtained as

$$T_{i,1} = \frac{\langle K_i y_i, K_{i_1} y_{i_1} \rangle_{\mathcal{H}_K}}{\langle K_{i_1} y_{i_1}, K_{i_1} y_{i_1} \rangle_{\mathcal{H}_K}} \tag{34}$$

$$= \frac{\langle K_{ii_1} y_i, y_{i_1} \rangle_{\mathcal{Y}}}{\langle K_{i_1 i_1} y_{i_1}, y_{i_1} \rangle_{\mathcal{Y}}}, \text{ for } i \in [n], \tag{35}$$

where $K_{ii_1} = K(x^{(i)}, x^{(i_1)})$ and (34) is obtained by differentiating the objective function in (33) with respect to $T_{i,1}$ and equating it to 0, to obtain the minima for $i \in [n]$. The reproducing property of OVK is used to obtain (35). This construction of $T_{i,1}$ yields

$$\langle K_i y_i - \bar{K}_i y_i, K_{i_1} y_{i_1} \rangle_{\mathcal{H}_K} = 0, \ \forall i \in [n]. \tag{36}$$

The equality in (36) denotes that the space $\text{span}\{(K_i y_i - \bar{K}_i y_i), \forall i \in [n]\}$ is orthogonal to $K_{i_1} y_{i_1}$ (denoted by $\text{span}\{(K_i y_i - \bar{K}_i y_i)\}, \forall i \in [n] \perp K_{i_1} y_{i_1}$). Note that this orthogonality property holds for index $i_1$. We shall show later that a similar property indeed holds for all samples which will be added to the working set. In general, the number of samples for the functional regression problem can be large and searching for the best candidates in the complete training set may become costly and defeat the cause for developing a sample based approximation. Hence we consider a random subset of $R$ samples for an efficient approximation, where $R < n$ and $|I| < R$.

For the iterative process to build the working set of indices $I$ and samples $I_W$, let us assume that $I^{old}$ be the index set with $|I^{old}| = k$ (say) and $T^{old} \in \mathbb{R}^{n \times k}$ be the matrix formed based on (31) for obtaining $\bar{K}_i^{old} y_i = \sum_{j=1}^{k} T_{i,j} K_{i_j} y_{i_j}, \forall i \in [n]$. Suppose $i_{k+1}$ be the index of next best sample to be added to get $I^{new} = I^{old} \cup \{i_{k+1}\}$ which provides $\bar{K}_i^{new} = \bar{K}_i^{old} y_i + T_{i,k+1} K_{i_{k+1}} y_{i_{k+1}}$. The minimization problem as in (33) is written as

$$\underset{T_{1,k+1}, T_{2,k+1}, \dots, T_{n,k+1}}{\arg\min} \sum_{i=1}^{n} \left\| K_i y_i - \bar{K}_i^{new} y_i \right\|_{\mathcal{H}_K}^2 = \sum_{i=1}^{n} \left\| K_i y_i - \bar{K}_i^{old} y_i - T_{i,k+1} K_{i_{k+1}} y_{i_{k+1}} \right\|_{\mathcal{H}_K}^2, \tag{37}$$

where the solution of (37) results in

$$T_{i,k+1} = \frac{\langle K_i y_i - \bar{K}_i^{old} y_i, K_{i_{k+1}} y_{i_{k+1}} \rangle_{\mathcal{H}_K}}{\langle K_{i_{k+1}} y_{i_{k+1}}, K_{i_{k+1}} y_{i_{k+1}} \rangle_{\mathcal{H}_K}} \tag{38}$$

$$= \frac{\langle K_{ii_{k+1}} y_i, y_{i_{k+1}} \rangle_{\mathcal{Y}} - \sum_{j=1}^{k} T_{i,j} \langle K_{i_j i_{k+1}} y_{i_j}, y_{i_{k+1}} \rangle_{\mathcal{Y}}}{\langle K_{i_{k+1} i_{k+1}} y_{i_{k+1}}, y_{i_{k+1}} \rangle_{\mathcal{Y}}}, \text{ for } i \in [n]. \tag{39}$$

$T_{i,k+1}$ in (38) is obtained similar to the procedure for (34) by differentiating the objective function in (37) with respect to $T_{i,k+1}$'s and equating it to 0, obtaining the minima for $i \in [n]$. Equation (39) follows from properties of inner-product and reproducing property of OVK. The iterative construction ensures that the following property holds:

$$\langle K_i y_i - \bar{K}_i^{new} y_i, K_{i_j} y_{i_j} \rangle_{\mathcal{H}_K} = 0, \ \forall i \in [n], \forall j \in [k+1] \tag{40}$$

$$\implies \text{span}\{K_i y_i - \bar{K}_i^{new} y_i | i \in [n]\} \perp \text{span}\{K_{i_j} y_{i_j} | j \in [k+1]\}. \tag{41}$$

Similar to (36), the iterative procedure ensures that the orthogonality property is extended to (41) in $\mathcal{H}_K$, which will be used in the iterative selection process discussed below. For each iteration, it remains to find the best sample from the $R$ randomly selected candidate set of samples, which is to be included in $I_W$. We discuss this next.

**Selecting the Best Samples Iteratively:** In order to find the training sample which will minimize the residuals most effectively, let $C$ be the candidate set of indices for training samples given by $C \subseteq [n] \setminus I$. Suppose for a particular iteration, let $I = \{i_1, i_2, \ldots, i_k\}$ and let the randomly selected candidate set of indices be $C = \{c_1, c_2, \ldots, c_M\}$. For each $c_r \in C$, we calculate the improvement in the sum of residuals which can result in including $c_r$ in $I$. Assume $\bar{K}_i^{old} y_i$ be given by

$$\bar{K}_i^{old} y_i = \sum_{j=1}^{k} T_{i,j} K_{i_j} y_{i_j}, \text{ for } i \in [n], \tag{42}$$

from which we obtain $\bar{K}_i^{new} y_i$ as follows:

$$\bar{K}_i^{new} y_i = \bar{K}_i^{old} y_i + T_{i,r} K_{c_r} y_{c_r}, \text{ for } c_r \in C. \tag{43}$$

In order to select the best sample index from the candidate set $C$, we need to find the index $c_r$ in $C$ which best approximates $\sum_{i=1}^{n} \|K_i y_i - \bar{K}_i^{new} y_i\|$, when $I^{new} = I^{old} \cup \{c_r\}$ is considered as the index set for the new working set of samples. Let the improvement in the sum of residuals by adding $c_r$ in $I^{old}$ be denoted by Improvement$(c_r)$, given by

$$\text{Improvement}(c_r) = \sum_{i=1}^{n} \|K_i y_i - \bar{K}_i^{old} y_i\|_{\mathcal{H}_K}^2 - \sum_{i=1}^{n} \|K_i y_i - \bar{K}_i^{new} y_i\|_{\mathcal{H}_K}^2 \tag{44}$$

$$= \frac{\sum_{i=1}^{n} \left[ \langle K_i y_i - \bar{K}_i^{old} y_i, K_{c_r} y_{c_r} \rangle_{\mathcal{H}_K} \right]^2}{\langle K_{c_r} y_{c_r}, K_{c_r} y_{c_r} \rangle_{\mathcal{H}_K}} \tag{45}$$

$$= \frac{\sum_{i=1}^{n} \left[ \langle K_{ic_r} y_i, y_{c_r} \rangle_{\mathcal{Y}} - \sum_{j=1}^{k} T_{i,j} \langle K_{i_j c_r} y_{i_j}, y_{c_r} \rangle_{\mathcal{Y}} \right]^2}{\langle K_{c_r c_r} y_{c_r}, y_{c_r} \rangle_{\mathcal{Y}}}. \tag{46}$$

Equation (44) quantifies the reduction in the residual value by the addition of $c_r$ to working set $I$ of indices. Equation (45) is obtained from (44) by using the properties of inner product and $\bar{K}_i^{old}$, for $i \in [n]$ and the orthogonality property (41). Equation (46) follows from the reproducing property of operator-valued kernel $K$. The sample which achieves the maximum improvement is considered to be the best sample to be added to the working set. In terms of indices, this selection becomes

$$\text{Best Index}_k = \underset{c_r \in C}{\arg\max} \text{ Improvement}(c_r).$$

---

**Algorithm 3 Sample-based Approximation**

---

**Input:** $\{(x^{(i)}, y^{(i)})\}_{i=1}^n, x^{(i)} = (x_1^{(i)}, x_2^{(i)}, \ldots, x_p^{(i)}) \in \mathcal{X}, y^{(i)} \in \mathcal{Y}$
**Output:** $I$, the index set of working set of samples.
**Initialize** $\text{vech}(L)^0, \text{vech}(D)^0$
$y_i \leftarrow y^{(i)}, K_i \leftarrow K(x^{(i)}, .), K_{ij} \leftarrow K(x^{(i)}, x^{(j)}), \ \forall i, j \in [n]$
**Initialize** $k = 0, I = \emptyset, T = \mathbf{0}$
**while** stopping criterion based on residual (49) is not satisfied **do**
    Construct $C$ by drawing random subset of $M$ elements from $[n] \setminus I, C = \{c_1, c_2, \ldots, c_M\}$
    Compute $T_{i,k+1} = \dfrac{\langle K_{ii_{k+1}} y_i, y_{i_{k+1}} \rangle_{\mathcal{Y}} - \sum_{j=1}^k T_{i,j} \langle K_{i_j i_{k+1}} y_{i_j}, y_{i_{k+1}} \rangle_{\mathcal{Y}}}{\langle K_{i_{k+1} i_{k+1}} y_{i_{k+1}}, y_{i_{k+1}} \rangle_{\mathcal{Y}}}$, for $i \in [n]$
    Improvement$(c_m) = \dfrac{\sum_{i=1}^n \left[ \langle K_{ic_m} y_i, y_{c_m} \rangle_{\mathcal{Y}} - \sum_{j=1}^k T_{i,j} \langle K_{i_j c_m} y_{i_j}, y_{c_m} \rangle_{\mathcal{Y}} \right]^2}{\langle K_{c_m c_m} y_{c_m}, y_{c_m} \rangle_{\mathcal{Y}}}$, for $m \in [M]$
    Best Index$_k = \arg\max_{c_m \in C}$ Improvement$(c_m)$
    $I = I \cup \{\text{Best Index}_k\}$
    $k \leftarrow k + 1$
**end while**

---

**Stopping Criterion:** As is the case for any iterative algorithm, an appropriate stopping criterion is required for ending the sample selection process which may be based on the number of iterations or accuracy. For using accuracy-based stopping criterion, residual is calculated for the $k$-th iteration as

$$\text{Residual}_k = \sum_{i=1}^n \left\| (K_i - \bar{K}_i) y_i \right\|_{\mathcal{H}_K}^2 \tag{47}$$

$$= \sum_{i=1}^n \left[ \langle K_i y_i, K_i y_i \rangle_{\mathcal{H}_K} - 2 \sum_{j=1}^k T_{ij} \langle K_i y_i, K_{i_j} y_{i_j} \rangle_{\mathcal{H}_K} + \sum_{j=1}^k \sum_{l=1}^k T_{ij} T_{il} \langle K_{i_j} y_{i_j}, K_{i_l} y_{i_l} \rangle_{\mathcal{H}_K} \right] \tag{48}$$

$$= \sum_{i=1}^n \langle K_{ii} y_i, y_i \rangle_{\mathcal{Y}} - 2 \sum_{i=1}^n \sum_{j=1}^k T_{i,j} \langle K_{ii_j} y_i, y_{i_j} \rangle_{\mathcal{Y}} + \sum_{i=1}^n \sum_{j=1}^k \sum_{l=1}^k T_{i,j} T_{i,l} \langle K_{i_j i_l} y_{i_j}, y_{i_l} \rangle_{\mathcal{Y}}. \tag{49}$$

Equation (48) follows from the properties of inner product of RKHS and (49) is obtained using the reproducibility property of $K$. As the first part of the summation in (49) remains constant for each iteration, a threshold for residual value can be used to determine convergence of the last two terms. For a very large set of training samples, a budget on the number of samples to consider can also be an effective tool for approximation. As the aim is to learn a kernel encapsulating the graphical structure between the input variables, the sample approximation can still be costly. An effective strategy is to start with an initial $L$ representing a fully connected graph and an initial $D$ which is the identity matrix. After the sample selection process, the final working set $I_W$ of samples indexed by $I$ are used throughout in the alternating minimization framework. In our implementations, we used the residual calculation using a validation set instead of the training set which provided a faster convergence and better generalization. Algorithm 3 illustrates the sample-based approximation for functional regression problem.

## 6 Generalization Analysis

Let $\mathcal{X} = (L^2([0,1]))^p$ be the input space and $\mathcal{Y} = L^2([0,1])$ be the output space. Consider the training samples given as $z = \{(x^{(i)}, y^{(i)}) : i \in [m]\} \subseteq \mathcal{X} \times \mathcal{Y} =: \mathfrak{Z}$ where $z_i := (x^{(i)}, y^{(i)}), \forall i \in [m]$ are drawn i.i.d. from a probability distribution $\mu$. The empirical error of a learned function-valued function $F$ on the data $z$ is given as the following:

$$\mathscr{E}_z(F) = \frac{1}{m} \sum_{i=1}^m \mathscr{L}(y^{(i)}, F(x^{(i)})), \tag{50}$$

where $\mathscr{L} : \mathcal{Y} \times \mathcal{Y} \to \mathbb{R}_+$ is a loss function defined on the output space $\mathcal{Y}$. A typical learning problem involves estimating a function-valued $F$ which is the solution of the following problem:

$$\min_{F \in \mathcal{H}_K} \mathscr{E}_\lambda(F, K), \tag{51}$$

where $\mathscr{E}_\lambda(F, K) := \mathscr{E}_z(F) + \lambda \|F\|^2_{\mathcal{H}_K}$. In our problem $K$ is parameterized by $L, D, \gamma, \gamma_{op}$ and belongs to a class of OVKs $\mathcal{K}$ and hence in this work, for the given data, we aim to learn the following:

$$(K_z, F_z) := \arg\min\{\mathscr{E}_\lambda(F, K) : K \in \mathcal{K}, F \in \mathcal{H}_K\}. \tag{52}$$

The problem (52) can be reformulated as a regularized empirical error minimization problem. Our focus is on the problem of bounding the generalization error of $F_z$, namely $\mathscr{E}(F_z) - \mathscr{E}(F^*)$, where $\mathscr{E}(F)$ is the expected error of $F$ given by $\mathscr{E}(F) := \mathbb{E}[\mathscr{L}(y, F(x))]$, the expectation $\mathbb{E}$ is taken over the probability measure $\mu$, and $F^*$ is the target function defined as

$$F^* = \arg\min \mathscr{E}(F), \tag{53}$$

where the minimum is taken over all measurable functions $F : \mathcal{X} \to \mathcal{Y}$.

## 6.1 Error Bounds

In this section, we introduce quantities which will be useful for our generalization bound analysis. We use the approach in (Micchelli et al., 2016; Ying & Zhou, 2007; Wu & Zhou, 2006) and introduce sample error as the following:

$$S_z(m, \lambda, F) = [\mathscr{E}(F_z) - \mathscr{E}_z(F_z)] + [\mathscr{E}_z(F) - \mathscr{E}(F)]. \tag{54}$$

The sample error $S_z(m, \lambda, F)$ in (54) consists of two terms $[\mathscr{E}(F_z) - \mathscr{E}_z(F_z)]$ and $[\mathscr{E}_z(F) - \mathscr{E}(F)]$. The first term $[\mathscr{E}(F_z) - \mathscr{E}_z(F_z)]$ is the difference between the expected value of $\mathscr{L}(y, F_z(x))$ with respect to $\mu$ and its empirical mean over a fixed random data set $z \subseteq \mathfrak{Z}$. To bound this term we use the notion of Rademacher averages which enables us to control and analyze the random variables $z_i$ associated with the data $z \subseteq \mathfrak{Z}$. Similarly, the second term $[\mathscr{E}_z(F) - \mathscr{E}(F)]$ denotes the difference between the empirical mean of $\mathscr{L}(y, F(x))$ for a fixed $z \subseteq \mathfrak{Z}$ and its expectation with respect to $\mu$. We follow the approach in (Micchelli et al., 2016) to bound both the terms.

In addition to the sample error, we introduce another quantity known as the regularization error $\mathcal{R}(F)$ for a function $F \in \mathcal{H}_K$ defined as

$$\mathcal{R}(F) = \mathscr{E}(F) - \mathscr{E}(F^*) + \lambda \|F\|^2_{\mathcal{H}_K}, \tag{55}$$

where $F^*$ is the target function. A regularized version of problem (53) is given by

$$(K^*_\lambda, F^*_\lambda) := \arg\min_{K \in \mathcal{K}, F \in \mathcal{H}_K} \{\mathscr{E}(F) + \lambda \|F\|^2_{\mathcal{H}_K} : K \in \mathcal{K}, F \in \mathcal{H}_K\}. \tag{56}$$

The regularization error of $F^*_\lambda$ is denoted by $\mathcal{R}^*(\lambda)$ as follows:

$$\mathcal{R}^*(\lambda) = \min_{K \in \mathcal{K}} \min_{F \in \mathcal{H}_K} \left[\mathscr{E}(F) - \mathscr{E}(F^*) + \lambda \|F\|^2_{\mathcal{H}_K}\right]. \tag{57}$$

In order to determine a generalization bound, we use the following result which enables us to relate generalization error using sample error and regularization error.

**Proposition 6.1.** For every $K \in \mathcal{K}, F \in \mathcal{H}_K$, the following inequality holds

$$\mathscr{E}(F_z) - \mathscr{E}(F^*) \le S_z(m, \lambda, F) + \mathcal{R}(F). \tag{58}$$

*Proof.* In order to prove the inequality, we start with the generalization error,

$$\mathscr{E}(F_z) - \mathscr{E}(F^*) = S_z(m, \lambda, F) + \mathscr{E}_z(F_z) - \mathscr{E}_z(F) + \mathscr{E}(F) - \mathscr{E}(F^*) + \lambda \|F\|^2_{\mathcal{H}_K} - \lambda \|F\|^2_{\mathcal{H}_K} \tag{59}$$

$$= S_z(m, \lambda, F) + \mathcal{R}(F) + \mathscr{E}_z(F_z) - \mathscr{E}_z(F) - \lambda \|F\|^2_{\mathcal{H}_K} \tag{60}$$

$$\le S_z(m, \lambda, F) + \mathcal{R}(F). \tag{61}$$

Equation (59) is obtained by adding and subtracting $S_z(m, \lambda, F)$ and $\lambda \|F\|_{\mathcal{H}_K}^2$. Equation (60) follows from the definition of $R(F)$ in (55). The inequality in (61) is obtained using the facts $\lambda \|F\|_{\mathcal{H}_K}^2 \geq 0$ and $\mathscr{E}_z(F_z) - \mathscr{E}_z(F) \leq 0$, as $F_z = \arg\min_{F \in \mathcal{H}_K} \mathscr{E}_z(F)$. □

Now, we intend to bound the term $\mathscr{E}(F_z) - \mathscr{E}_z(F_z)$ in (54). Towards this we define the following notion of Rademacher average of a suitable class of functions.

**Definition 6.1** (**Rademacher Average**). Let $\mathcal{F}$ denote a class of functions from $\mathcal{X}$ to $\mathcal{Y}$. Let $\mu_{\mathcal{X}}$ denote the marginal distribution over the input space $\mathcal{X}$. Consider a $m$-tuple of samples from input space as $(x^{(1)}, x^{(2)}, \ldots, x^{(m)}) \in \mathcal{X}^m$, where $x^{(i)} \sim \mu_{\mathcal{X}}, i \in [m]$. Then the Rademacher average of class $\mathcal{F}$ is defined as

$$\mathscr{R}_{m;\mathcal{Y}}(\mathcal{F}) = \mathbb{E}\left[\sup_{F \in \mathcal{F}} \frac{1}{m} \left\|\sum_{i=1}^m \varepsilon_i F(x^{(i)})\right\|_{\mathcal{Y}}\right], \tag{62}$$

where $\varepsilon_i$'s are Rademacher random variables uniformly distributed over $\{+1, -1\}$ and $\mathbb{E}$ represents the expectation over both i.i.d. Rademacher variables $\varepsilon_i$'s and i.i.d. variables $x^{(i)}$'s based on $\mu_{\mathcal{X}}$.

Recall that for a class of functions $\mathsf{F}$ from an input space $X$ to $\mathbb{R}$, and a sample $(x_1, x_2, \ldots, x_m) \in X^m$, the Radamacher average of $\mathsf{F}$ is defined as

$$\mathscr{R}_{m;\mathbb{R}}(\mathsf{F}) = \mathbb{E}\left[\sup_{F \in \mathsf{F}} \frac{1}{m} \sum_{i=1}^m \varepsilon_i F(x_i)\right]. \tag{63}$$

Comparing this expression with that in Definition 6.1, we note that a suitable norm is used in Definition 6.1. Thus our definition of a suitable Radamacher average accommodates the nature of the function class $\mathcal{F}$ which contains function-valued functions, unlike $\mathsf{F}$ which is composed of simple real-valued functions.

In order to proceed with the upcoming proofs, we require some assumptions which we state next. Assume $\exists \beta > 0$ such that $\|y\|_{\mathcal{Y}} \leq \beta, \ \forall y \in \mathcal{Y}$, which provides a uniform upper bound on the norm of the outputs. In addition, assume that the class $\mathcal{K}$ is uniformly bounded, that is,

$$\kappa = \sup_{K \in \mathcal{K}} \sup_{x \in \mathcal{X}} \sup_{y \in \mathcal{Y}} \|K(x, .)y\|_{\mathcal{H}_K} = \sup_{K \in \mathcal{K}} \sup_{x \in \mathcal{X}} \sup_{y \in \mathcal{Y}} \sqrt{\langle K(x,x)y, y \rangle_{\mathcal{Y}}} < \infty. \tag{64}$$

This assumption holds for OVKs which satisfy the trace class assumption (Kadri et al., 2016).

For any $K \in \mathcal{K}$ and $F \in \mathcal{H}_K$, we define the following norm

$$\|F\|_\infty = \max_{x \in \mathcal{X}} \max_{y \in \mathcal{Y}} |\langle F(x), y \rangle_{\mathcal{Y}}| = \max_{x \in \mathcal{X}} \max_{y \in \mathcal{Y}} |\langle F, K(x, .)y \rangle_{\mathcal{H}_K}| \leq \kappa \|F\|_{\mathcal{H}_K}, \tag{65}$$

where we use the reproducing property, $\langle F, K(x, .)y \rangle_{\mathcal{H}_K} = \langle F(x), y \rangle_{\mathcal{Y}}$ and the inequality in (65) follows from Cauchy-Schwarz inequality. We define for $t \geq 0$,

$$\Xi(t) := \sup_{y \in \mathcal{Y}} \sup_{\|s\|_{\mathcal{Y}} \leq t} \mathscr{L}(y, s). \tag{66}$$

The function $\Xi(t)$ provides a bound on the loss function when the second argument $s$ in the loss function has restricted norm. $\Xi(t)$ enables bounding the norm of the function $F$ via the loss function $\mathscr{L}$ by considering $t = 0$. Let $L : \mathcal{Y} \to \mathbb{R}$ be defined as the following:

$$L(t) = \sup_{y \in \mathcal{Y}} \sup_{\substack{\|s_1\|_{\mathcal{Y}} \leq t, \\ \|s_2\|_{\mathcal{Y}} \leq t}} \frac{|\mathscr{L}(y, s_1) - \mathscr{L}(y, s_2)|}{\|s_1 - s_2\|_{\mathcal{Y}}}. \tag{67}$$

$L(t)$ provides a Lipschitz constant for the loss function $\mathscr{L}$ with respect to the second argument when the norm of the argument is bounded by $t$.

**Lemma 6.2.** Let $\mathcal{F}$ be a class of functions from $\mathcal{X}$ to $\mathcal{Y}$. Consider a $m$-tuple of samples from input space as $(x^{(1)}, x^{(2)}, \ldots, x^{(m)}) \in \mathcal{X}^m$. Then the following hold:

1. $\mathbb{E}\left[\sup_{F \in \mathcal{F}} \|\frac{1}{m}\sum_{i=1}^{m} F(x^{(i)}) - \mathbb{E}F\|_{\mathcal{Y}}\right] \leq 2\mathscr{R}_{m;\mathcal{Y}}(\mathcal{F})$.

2. For every $c \in \mathbb{R}$, $\mathscr{R}_{m;\mathcal{Y}}(c\mathcal{F}) = |c|\mathscr{R}_{m;\mathcal{Y}}(\mathcal{F})$.

3. For $\phi : \mathcal{Y} \to \mathbb{R}$, if $\phi$ is a Lipschitz function with Lipschitz constant $L$, then $\mathscr{R}_{m;\mathbb{R}}(\phi \circ \mathcal{F}) \leq L\mathscr{R}_{m;\mathcal{Y}}(\mathcal{F})$.

*Proof.* The lemma has been proved as Lemma A.6 in Appendix A.9. □

We further define the following constants:

$$\rho = \sqrt{\Xi(0)/\lambda},\ \tau = \kappa\rho, \tag{68}$$

which will be useful in the upcoming results. The forthcoming results will use the class of kernels given by:

$$\mathcal{K}_0 = \{K(x,.)y : K \in \mathcal{K}, x \in \mathcal{X}, y \in \mathcal{Y}\}. \tag{69}$$

The following result provides a bound on the sample error which involves the Rademacher average of the class of kernels $\mathcal{K}_0$.

**Theorem 6.3.** If $F \in \mathcal{H}_{\mathcal{K}}$, then with confidence $1 - \delta$, where $\delta \in (0, 1)$, there holds

$$S_z(m, \lambda, F) \leq 2\rho L(\tau)\beta^{1/4}(\mathscr{R}_{m;\mathcal{Y}}(\mathcal{K}_0))^{1/4} + (\Xi(\tau) + \Xi(\|F\|_\infty))\sqrt{\frac{\log\frac{1}{\delta}}{2m}}. \tag{70}$$

The proof will be provided later as a consequence of the results covered in the upcoming section. Theorem 6.3 provides a probabilistic upper bound for the sample error in terms of Rademacher average for the class of graph-induced operator-valued kernels. Later in Section 6.3, we will consider $\mathcal{K}$ to be the class of graph-induced OVKs and derive a bound on $\mathscr{R}_{m;\mathcal{Y}}(\mathcal{K}_0)$.

## 6.2 Estimating Sample Error

In this section, we derive results which aid in establishing the result in Theorem 6.3. We use Hoeffding inequality for bounding the term $\mathscr{E}_z(F) - \mathscr{E}(F)$ using random data set $z \subseteq \mathfrak{Z} := \mathcal{X} \times \mathcal{Y}$.

**Lemma 6.4.** Let $F$ be a bounded function. For every $\delta \in (0, 1)$, with confidence $1 - \delta$ there holds

$$\mathscr{E}_z(F) - \mathscr{E}(F) \leq \Xi(\|F\|_\infty)\sqrt{\frac{\log\frac{1}{\delta}}{2m}}. \tag{71}$$

*Proof.* Consider the random variable $\zeta = \mathscr{L}(y, F(x))$. Note that $\mathscr{E}_z = \frac{1}{m}\sum_{i=1}^{m}\zeta(z_i)$, where $z_i = (x^{(i)}, y^{(i)})$ and $\mathscr{E} = \mathbb{E}(\zeta)$. By our assumption, $0 < \zeta \leq \Xi(\|F\|_\infty)$ we have $|\zeta - \mathbb{E}[\zeta]| \leq \Xi(\|F\|_\infty)$. Using one-sided Hoeffding inequality, we obtain

$$P\left(\frac{1}{m}\sum_{i=1}^{m}(\zeta_i - \mathbb{E}[\zeta]) \geq t\right) \leq \exp\left(-\frac{2mt^2}{\Xi^2(\|F\|_\infty)}\right). \tag{72}$$

Consider,

$$\delta = \exp\left(-\frac{2mt^2}{\Xi^2(\|F\|_\infty)}\right) \implies \log\frac{1}{\delta} = \left(\frac{2mt^2}{\Xi^2(\|F\|_\infty)}\right) \implies t = \Xi(\|F\|_\infty)\sqrt{\frac{\log\frac{1}{\delta}}{2m}}. \tag{73}$$

Therefore, for every $\delta \in (0, 1)$, with confidence $1 - \delta$ the following holds:

$$\mathscr{E}_z(F) - \mathscr{E}(F) \leq \Xi(\|F\|_\infty)\sqrt{\frac{\log\frac{1}{\delta}}{2m}}. \tag{74}$$

□

Now that the second term in the sample error (54) has been bounded, we focus on the first term by considering a union of unit balls in the space $\mathcal{H}_K$ where the notion of Rademacher average can be defined for obtaining bounds. We define a function $\Theta$ such that

$$\mathscr{E}(F_z) - \mathscr{E}_z(F_z) \leq \Theta(z) := \sup_{F \in \rho\mathcal{B}_{\mathcal{K}}} \left(\mathscr{E}(F) - \mathscr{E}_z(F)\right), \tag{75}$$

where $\mathcal{B}_{\mathcal{K}}$ is the union of unit balls in $\mathcal{H}_K$ over $K \in \mathcal{K}$ given by

$$\mathcal{B}_{\mathcal{K}} = \bigcup_{K \in \mathcal{K}} \left\{F \in \mathcal{H}_K : \|F\|_{\mathcal{H}_K} \leq 1\right\}. \tag{76}$$

The supremum is defined over $\rho\mathcal{B}_{\mathcal{K}}$ in (75) by the following reasoning: $\lambda\|F_z\|^2_{\mathcal{H}_{K_z}} \leq \mathscr{E}(F_z) - \mathscr{E}(F^*) + \lambda\|F_z\|^2_{\mathcal{H}_{K_z}} \leq \mathscr{E}_z(0) - \mathscr{E}(F^*) \leq \sup_{y \in \mathcal{Y}} \mathscr{L}(y, 0) = \sup_{y \in \mathcal{Y}} \sup_{s:\|s\|_{\mathcal{Y}} \leq 0} \mathscr{L}(y, s) = \Xi(0)$. This gives a bound on $\|F_z\|_{\mathcal{H}_{K_z}}$, note that $F_z$ is the minimizer in problem (52). In fact, using a similar approach, for any general $F \in \mathcal{H}_K$, we have $\|F\|_{\mathcal{H}_K} \leq \sqrt{\Xi(0)/\lambda} =: \rho$ which leads to the set $\rho\mathcal{B}_{\mathcal{K}}$ in (75).

Thus to bound $\mathscr{E}(F_z) - \mathscr{E}_z(F_z)$, we find a suitable upper bound on $\Theta(z)$ in the following lemma.

**Lemma 6.5.** Consider $\Theta(z) = \sup_{F \in \rho\mathcal{B}_{\mathcal{K}}} \left(\mathscr{E}(F) - \mathscr{E}_z(F)\right)$, then for every $\delta \in (0, 1)$, with confidence $1 - \delta$ the following holds

$$\Theta(z) \leq \mathbb{E}[\Theta(z)] + \Xi(\tau)\sqrt{\frac{\log\frac{1}{\delta}}{2m}}. \tag{77}$$

*Proof.* Let $z_i'$ be the data set which is obtained by replacing $i$-th pair $z_i = (x^{(i)}, y^{(i)})$ of $z$ with $(x_i', y_i')$. Then,

$$\Theta(z) - \Theta(z_i') = \sup_{F \in \rho\mathcal{B}_{\mathcal{K}}} \left(\mathscr{E}(F) - \mathscr{E}_z(F)\right) - \sup_{F \in \rho\mathcal{B}_{\mathcal{K}}} \left(\mathscr{E}(F) - \mathscr{E}_{z_i'}(F)\right) \tag{78}$$

$$\leq \sup_{F \in \rho\mathcal{B}_{\mathcal{K}}} \left(\mathscr{E}_{z_i'}(F) - \mathscr{E}_z(F)\right) \tag{79}$$

$$= \frac{1}{m} \sup_{F \in \rho\mathcal{B}_{\mathcal{K}}} \left(\mathscr{L}(y_i', F(x_i')) - \mathscr{L}(y^{(i)}, F(x^{(i)}))\right) \tag{80}$$

$$\leq \frac{1}{m}\Xi(\tau), \tag{81}$$

where (79) is obtained by using properties of supremum and (81) follows from the definition of $\kappa$, $\tau$ and $\Xi$. By interchanging $z$ and $z_i'$, we obtain

$$|\Theta(z) - \Theta(z_i')| \leq \frac{1}{m}\Xi(\tau). \tag{82}$$

Using McDiarmid's inequality, we obtain

$$P\left(\Theta(z) - \mathbb{E}[\Theta(z)] \geq \epsilon\right) \leq \exp\left(-\frac{2m\epsilon^2}{\Xi^2(\tau)}\right). \tag{83}$$

Using a similar argument as in the proof of Lemma 6.4, the proof follows that for every $\delta \in (0, 1)$, with confidence $1 - \delta$ the following holds:

$$\Theta(z) - \mathbb{E}[\Theta(z)] \leq \Xi(\tau)\sqrt{\frac{\log\frac{1}{\delta}}{2m}}. \tag{84}$$

$\square$

The next lemma helps to bound $\mathbb{E}[\Theta(z)]$ using Rademacher average of the class $\mathcal{K}_0$.

**Lemma 6.6.** $\mathbb{E}[\Theta(z)]$ is bounded above as follows:

$$\mathbb{E}[\Theta(z)] \leq 2\rho L(\tau)\beta^{1/4}(\mathscr{R}_{m;\mathcal{Y}}(\mathcal{K}_0))^{1/4}.$$

*Proof.* The lemma has been proved as Lemma A.7 in Appendix A.9. □

Using Lemmas 6.4, 6.5 and 6.6, the result in Theorem 6.3 is proved.

### 6.3 Learning with Graph-Induced Operator-valued Kernels

In this section, we consider the following class of functions

$$\mathcal{K}_0 = \{K(x,.)y : K \in \mathcal{K}, x \in \mathcal{X}, y \in \mathcal{Y}\}, \tag{85}$$

where $K$ is defined as the graph-induced OVK given by

$$K(x,x')y = e^{-\gamma(x-x')^\top (L+D)(x-x')} \int_0^1 e^{-\gamma_{op}|s-t|}y(s)ds, \tag{86}$$

$$= \mathfrak{g}(x,x')Ty, \tag{87}$$

with $\gamma, \gamma_{op} > 0, L \in \mathcal{L}$ and $D \in \mathcal{D}$. Note that $\mathfrak{g}(x,x') = e^{-\gamma(x-x')^\top (L+D)(x-x')}$ and $Ty = \int_0^1 e^{-\gamma_{op}|s-t|}y(s)ds$.

Now, in order to bound the Rademacher average $\mathscr{R}_{m;\mathcal{Y}}(\mathcal{K}_0)$, we follow an approach inspired by Maurer (2016) and split the OVK $K$ using properties on $\mathfrak{g}$ and $T$. Consider the class of functions defined as $\mathcal{G} = \{\mathfrak{g}(x,.) = e^{-\gamma(x-.)^\top (L+D)(x-.)} \in \mathcal{H}_{\mathcal{G}} : \|\mathfrak{g}(x,.)\|_{\mathcal{H}_{\mathcal{G}}} \leq R_{\mathcal{G}}, L \in \mathcal{L}, D \in \mathcal{D}, \gamma > 0\}$ where $\mathcal{H}_{\mathcal{G}}$ is the RKHS corresponding to the scalar-valued kernel $\mathfrak{g}$ on $\mathcal{X} \times \mathcal{X}$.

$$\mathscr{R}_{m;\mathcal{Y}}(\mathcal{K}_0) = \mathbb{E}\left[\sup_{k \in \mathcal{G}} \sup_{y \in \mathcal{Y}} \sup_{t \in \mathcal{X}} \left\|\frac{1}{m}\sum_{i=1}^m \varepsilon_i k(x^{(i)}, t)Ty\right\|_{\mathcal{Y}}\right] \tag{88}$$

$$\leq \mathbb{E}\left[\sup_{k \in \mathcal{G}} \sup_{y \in \mathcal{Y}} \sup_{t \in \mathcal{X}} \left|\frac{1}{m}\sum_{i=1}^m \varepsilon_i k(x^{(i)}, t)\right| \|Ty\|_{\mathcal{Y}}\right] \tag{89}$$

$$= \left(\sup_{y \in \mathcal{Y}} \|Ty\|_{\mathcal{Y}}\right) \mathbb{E}\left[\sup_{k \in \mathcal{G}} \sup_{t \in \mathcal{X}} \left|\frac{1}{m}\sum_{i=1}^m \varepsilon_i k(x^{(i)}, t)\right|\right] \tag{90}$$

$$= \left(\sup_{y \in \mathcal{Y}} \|Ty\|_{\mathcal{Y}}\right) \mathscr{R}_{m;\mathbb{R}}^+(\mathcal{G}). \tag{91}$$

In Equation (91), $\mathscr{R}_{m;\mathbb{R}}^+(\mathcal{G})$ denotes a Rademacher average involving absolute values of real-valued functions in $\mathcal{G}$. For bounding Rademacher average $\mathscr{R}_{m;\mathbb{R}}^+(\mathcal{G})$, we use the notion of covering numbers. Next, we provide the definition of covering numbers.

**Definition 6.2** (**Covering Numbers**). Let $(\mathbb{F}, d)$ be a pseudo-metric space and $S$ be a subset of $\mathbb{F}$. For every $\epsilon > 0$, the covering number of $S$ by balls of radius $\epsilon$ with respect to $d$, denoted by $\mathcal{N}(S, \epsilon, d)$ is defined as the minimal number of balls of radius $\epsilon$ whose union covers $S$, namely,

$$\mathcal{N}(S, \epsilon, d) = \min\left\{n \in \mathbb{N} : \exists\{s_j\}_{j=1}^n \subset \mathbb{F} \text{ such that } S \subseteq \bigcup_{j=1}^n B(s_j, \epsilon)\right\},$$

where $B(s_j, \epsilon) = \{s \in \mathbb{F} : d(s, s_j) \leq \epsilon\}$.

Let $\mathcal{Q}$ be a class of bounded real-valued functions defined on $\mathcal{X}$, $\mathbf{x} = (x^{(i)} : i \in [m]) \in \mathcal{X}^m$ and $\mathcal{Q}|_{\mathbf{x}} = \{(Q(x^{(i)}) : i \in [m]) : Q \in \mathcal{Q}\} \subseteq \mathbb{R}^m$. For a norm induced by $d$ on $\mathcal{X}$, we define the $d$-norm empirical covering number of $\mathcal{Q}$ associated with $\mathbf{x}$ as $\mathcal{N}_d(\mathcal{Q}, \epsilon, m) = \sup_{\mathbf{x} \in \mathcal{X}^m} \mathcal{N}(\mathcal{Q}|_{\mathbf{x}}, \epsilon, d)$.

Let $U = \sup_{\mathfrak{g} \in \mathcal{G}} \mathbb{E}[\mathfrak{g}^2]$. Using a construction similar to (124) and (125) in Appendix A.3, we obtain

$$\mathfrak{g}(x, .) = e^{-\gamma(x-.)^\top (L+D)(x-.)} = e^{-\gamma \|A(x-.)\|_p^2},$$

where $A = \sqrt{\Lambda} V$, for a diagonal matrix $\Lambda$ with non-negative eigenvalues of $L + D$ and V is a orthonormal matrix. Consider an appropriate bound as $\mathbb{E}[\mathfrak{g}^2] \leq a$, $\forall \mathfrak{g} \in \mathcal{G}$ which is reasonable owing to the RBF-based construction of scalar-valued kernels in $\mathcal{G}$. Based on Corollary 2.2.8 in (Van Der Vaart & Wellner, 1996), we can bound the Rademacher average using covering number as

$$\mathscr{R}_{m;\mathbb{R}}^+(\mathcal{G}) \leq \frac{1}{\sqrt{m}} \int_0^U \sqrt{\log \mathcal{N}_d(\mathcal{G}, \epsilon, m)} d\epsilon, \tag{92}$$

$$\leq \frac{1}{\sqrt{m}} \int_0^a \sqrt{\log \mathcal{N}_d(\mathcal{G}, \epsilon, m)} d\epsilon. \tag{93}$$

To bound $\mathcal{N}_d(\mathcal{G}, \epsilon, m)$, we use Remark 11 in (Cucker & Smale, 2002), to state that there exists $C > 0$ and $q > 0$ such that

$$\log \mathcal{N}_d(\mathcal{G}, \epsilon, m) \leq \left( \frac{R_{\mathcal{G}} C}{\epsilon} \right)^{\frac{1}{q}}. \tag{94}$$

Based on our assumptions, there exists $\mathbb{K} > 0$ such that $\sup_{y \in \mathcal{Y}} \|Ty\|_{\mathcal{Y}} \leq \mathbb{K}$. Using (93) and (94) in (91), we obtain

$$\mathscr{R}_{m;\mathcal{Y}}(\mathcal{K}_0) \leq \frac{2aq\mathbb{K}(R_{\mathcal{G}}C/a)^{1/2q}}{(2q-1)\sqrt{m}}. \tag{95}$$

Using the result in (95) with (70), we can establish the generalization bounds for the class of kernels constructed with graph-induced operator-valued kernels. For the problem considered in this work, the loss is defined as $\mathscr{L}(y, y') = \int_0^1 (y(t) - y'(t))^2 dt$ with $\Xi(t) \leq (\beta + t)^2$ and $L(t) \leq 2(\beta + t)$. We obtain the following for $\lambda < 1, \delta \in (0, 1)$, with confidence $1 - \delta$ as

$$S_z(m, \lambda, F_\lambda^*) \leq 4\rho(\beta + \tau)\beta^{1/4} \left( \frac{2aq\mathbb{K}(R_{\mathcal{G}}C/a)^{1/2q}}{(2q-1)\sqrt{m}} \right)^{1/4} + ((\beta + \tau)^2 + (\beta + \|F_\lambda^*\|_\infty)^2)\sqrt{\frac{\log \frac{1}{\delta}}{2m}} \tag{96}$$

$$\leq \frac{4\beta}{\sqrt{\lambda}} \left( \beta + \frac{\kappa\beta}{\sqrt{\lambda}} \right) \beta^{1/4} \left( \frac{2aq\mathbb{K}(R_{\mathcal{G}}C/a)^{1/2q}}{(2q-1)\sqrt{m}} \right)^{1/4} + 2 \left( \beta + \frac{\kappa\beta}{\sqrt{\lambda}} \right)^2 \sqrt{\frac{\log \frac{1}{\delta}}{2m}} \tag{97}$$

$$= \frac{4\beta^{9/4}}{\lambda}(\kappa + \sqrt{\lambda}) \left( \frac{2aq\mathbb{K}(R_{\mathcal{G}}C/a)^{1/2q}}{(2q-1)\sqrt{m}} \right)^{1/4} + \frac{2\beta^2}{\lambda}(\kappa + \sqrt{\lambda})^2 \sqrt{\frac{\log \frac{1}{\delta}}{2m}} \tag{98}$$

$$< \left[ \beta^{1/4} \left( \frac{2aq\mathbb{K}(R_{\mathcal{G}}C/a)^{1/2q}}{(2q-1)} \right)^{1/4} + \sqrt{\frac{\log \frac{1}{\delta}}{2}} \right] \frac{\beta^2}{\lambda m^{1/8}} \max\{4(\kappa + 1), 2(\kappa + 1)^2\}. \tag{99}$$

The inequality (97) is obtained using $\rho = \sqrt{\Xi(0)/\lambda} \leq \beta/\sqrt{\lambda}$, $\tau = \kappa\rho$ and $\|F_\lambda^*\|_\infty \leq \kappa\rho$. Now, a common assumption for smooth kernels is of logarithmic decay of regularization error, i.e., $\mathcal{R}^*(\lambda) \leq c'\lambda^\eta$, where $\eta \in (0, 1]$ and $c' > 0$ (Micchelli et al., 2016). Then the generalization error is bounded by

$$\mathscr{E}(F_z) - \mathscr{E}(F^*) \leq \frac{c}{\lambda} + c'\lambda^\eta. \tag{100}$$

Consider the function $\mathscr{H}(\lambda) = \frac{c}{\lambda} + c'\lambda^\eta$, then the minimizer is obtained for $\lambda^* = (c/\eta c')^{1/(1+\eta)}$ with $\mathscr{H}(\lambda^*) = (\eta c')^{1/(1+\eta)} [1 + 1/\eta] c^{\eta/(1+\eta)}$. Therefore, for $\delta \in (0, 1)$, with confidence $1 - \delta$ we obtain

$$\mathscr{E}(F_z) - \mathscr{E}(F^*) \leq (\eta c')^{1/(1+\eta)} (1 + 1/\eta) \left[ \left( \beta^{1/4} \left( \frac{2aq\mathbb{K}(R_{\mathcal{G}}C/a)^{1/2q}}{(2q-1)} \right)^{1/4} + \sqrt{\frac{\log \frac{1}{\delta}}{2}} \right) \frac{\beta^2}{m^{1/8}} \mathfrak{A} \right]^{\eta/(1+\eta)}, \tag{101}$$

where $\mathfrak{A} = \max\{4(\kappa + 1), 2(\kappa + 1)^2\}$. (101) ensures an upper bound for the generalization error with the help of a bound on the Rademacher average for the problem (52) of learning the OVK $K_z$ and the functional map $F_z$ in the induced RKHS corresponding to $K_z$ by using a ball $\mathcal{B}_\mathcal{K}$ in corresponding RKHS with a fixed radius (considered as 1). The task of establishing a bound on the regularization error $\mathcal{R}^*(\lambda)$ in (Micchelli et al., 2016) considers an example prescribing value for $\eta$ based on the hyperparameter in the RBF kernel. A similar pursuit in our setting is not straightforward because of the functional nature of the input space. Hence, we leave it for future work.

# 7 Experiments

In order to illustrate the effectiveness of the developed framework, we have used functional regression problem with an unknown graph structure in the input data for both synthetic and real datasets. The task of predicting output functions with the help of a Laplacian matrix denoting the relationship between the set of $p$ input functions has been illustrated in the experiments. As practical data is always available as discrete observations corresponding to functions, standard FDA techniques can be used for the conversion of functional data into vector representation using basis functions, e.g. Fourier basis, B-spline basis, etc. Let $\mathcal{X} = (L^2([a, b]))^p$ and $\mathcal{Y} = L^2([c, d])$ be the input and output spaces, respectively. For our experiments, the error metric used is residual sum of squares error (RSSE) (Kadri et al., 2016) defined as $RSSE = \sum_i \int_c^d \{y^{(i)}(t) - \hat{y}^{(i)}(t)\}^2 dt$, where $y^{(i)}$ is the actual output function and $\hat{y}^{(i)}$ is the predicted output function. RSSE is better suited to compare functional outputs. The integrals involved have been approximated by using numerical integration in our implementation. The quadratic programs involved in (23) and (15) are solved by using CVXOPT (Andersen et al., 2023).

**Experimental Setting**: All methods were coded in Python 3.7 and the codes are made public.[1] All experiments were run on a Linux box with 182 Gigabytes main memory and 28 CPU cores. As methods to solve the problem of functional regression problem simultaneously with learning $L$ and/or $D$ are not available, we use popular algorithms to first determine $L$. Then for the learned $L$, we use our alternating minimization framework to learn $D$ using projected gradient descent and $\mathbf{u}$ using OpMINRES. For the MCP-based $L$ learning and $D$ learning in the proposed alternating minimization framework, we use a decaying step-size in the projected gradient descent. The decaying step-size regime involves starting with an initial step-size (e.g. $10^{-4}$) and reducing it by a fixed factor (e.g. 2) after a set of iterations (e.g. 5) continuously till a final step-size (e.g. $10^{-9}$). In order to illustrate the effectiveness, we consider the following methods for comparison.

**fglasso-OpMINRES-D**: $L$ is determined using fglasso (Qiao et al., 2019), based on a Gaussian functional model which provides a precision matrix (inverse of covariance matrix) corresponding to the nodes with corresponding functional input data. The approach develops an extension of the glasso criterion (Yuan & Lin, 2007) to fglasso for functional data. The learned $L$ is then used with our alternating minimization regime for optimizing $\mathbf{u}$ and $D$. OpMINRES is used with $k_1(x, x'; G) = e^{-\gamma(x-x')^\top (L+D)(x-x')}$ and $k_2(s, t) = e^{-\gamma_{op}|s-t|}$, where $\gamma \in \{10^{-6}, 10^{-5}, 10^{-4}, 10^{-3}, 10^{-2}, 10^{-1}, 1, 10, 100\}$ and $\gamma_{op} \in \{10^{-6}, 10^{-5}, 10^{-4}, 10^{-3}, 10^{-2}, 10^{-1}, 1, 10, 100\}$.

**KGL-OpMINRES-D**: $L$ is obtained by using Kernel Graph Learning (KGL) (Pu et al., 2021b) problem with respect to two kernel Gram matrices obtained for input signals and their timestamps which have been used to establish the relationship between input functions. RBF kernels have been considered for the input functions. The hyperparameters for KGL are tuned using cross-validation in our implementation. The learned $L$ is then used with our alternating minimization regime for optimizing $\mathbf{u}$ and $D$. OpMINRES is used with $k_1(x, x'; G) = e^{-\gamma(x-x')^\top (L+D)(x-x')}$ and $k_2(s, t) = e^{-\gamma_{op}|s-t|}$, where $\gamma \in \{10^{-6}, 10^{-5}, 10^{-4}, 10^{-3}, 10^{-2}, 10^{-1}, 1, 10, 100\}$ and $\gamma_{op} \in \{10^{-6}, 10^{-5}, 10^{-4}, 10^{-3}, 10^{-2}, 10^{-1}, 1, 10, 100\}$.

**Sparse OpMINRES-L-D**: This denotes our proposed method where we used Algorithm 2 to learn $\mathbf{u}$, $L$ and $D$. Projected gradient descent is used in minimization with respect to $L$ and $D$ based on a decaying step-size. The sparsity is aided by the MCP regularization considered in learning of $L$. The graph-induced operator-

---

[1]Codes used for the experiments can be found at `https://github.com/akashsaha06/graph-inducedOVK`.

valued kernels are obtained using $k_1(x, x'; G) = e^{-\gamma(x-x')^\top (L+D)(x-x')}$ and $k_2(s,t) = e^{-\gamma_{op}|s-t|}$, where $\gamma \in \{10^{-6}, 10^{-5}, 10^{-4}, 10^{-3}, 10^{-2}, 10^{-1}, 1, 10, 100\}$ and $\gamma_{op} \in \{10^{-6}, 10^{-5}, 10^{-4}, 10^{-3}, 10^{-2}, 10^{-1}, 1, 10, 100\}$.

**Sparse Non-Pos-OpMINRES-L-D**: As our framework is developed for OVKs, the proposed alternating minimization is well adaptive to consider generalized non-positive semi-definite OVKs (Saha & Palaniappan, 2020) as graph-induced OVKs. We call this extension Sparse Non-Pos-OpMINRES-L-D. Here too, projected gradient descent is used in minimization with respect to $L$ and $D$ using the decaying step-size similar to Sparse OpMINRES-L-D. The sparsity is aided by the MCP regularization considered in learning of $L$. The graph-induced operator-valued kernels are obtained using $k_1(x, x'; G) = e^{-\gamma(x-x')^\top (L+D)(x-x')}$ and $k_2(s,t) = e^{-\gamma_{op1}|s-t|} - e^{-\gamma_{op2}|s-t|}$, where $\gamma \in \{10^{-6}, 10^{-5}, 10^{-4}, 10^{-3}, 10^{-2}, 10^{-1}, 1, 10, 100\}$ and $\gamma_{op1}, \gamma_{op2} \in \{10^{-6}, 10^{-5}, 10^{-4}, 10^{-3}, 10^{-2}, 10^{-1}, 1, 10, 100\}$. Note that $k_2$ is not necessarily a positive semi-definite kernel.

**Stopping Criteria**: We elaborate on the stopping criteria for the different algorithms used in the alternating minimization framework in Algorithm 2.

- **OpMINRES**: The stopping criterion for OpMINRES is based on the following condition: the loop exits if the value of relative residual norms between the current residual norm and the initial residual norm is less than a threshold (e.g. $10^{-3}$ was used in our implementation).

- **Projected Gradient Descent**: The projected gradient descent steps for both vech($L$) and vech($D$) in Algorithm 2 use similar stopping criterion where the norm of difference between two consecutive iterates is compared to be less than a threshold (e.g. $10^{-3}$ was used in our implementation).

- **MCP Regularization**: The sparsity-inducing MCP regularization of vech($L$) in Algorithm 1 compares the norm of difference between two consecutive iterates against a threshold (considered as $10^{-3}$ in our implementation) as the stopping criterion.

## 7.1 Experiments with synthetic data

**Data Generation**: For synthetic experiments, three sets of experiments have been considered with input functions for graph structures having 3-nodes, 12-nodes and 25-nodes, respectively. The input functions are generated based on weighted cosine functions and constant functions with random noise. The corresponding output function is based on weighted sine functions sharing the weights between input functions and output function (details are given in Appendix A.10).

For all the methods, a truncated trigonometric basis of $L^2([0, 2\pi])$ with 30 basis functions has been considered for encoding the functional data. The experiments were run for three settings where the data has been divided randomly into a training set, a validation set and a test set. The following data splits have been considered: $(80/20/20)$, $(160/40/40)$ and $(320/80/80)$, representing the number of training samples/validation samples/test samples.

The results for synthetic data with 12 nodes are summarized in Tables 1-2. From Table 2, we observe that Sparse OpMINRES-L-D obtains comparable performance based on mean RSSE on the test data in all three settings where 80 samples, 160 samples and 320 samples have been used for training. Both fglasso-OpMINRES-D and KGL-OpMINRES-D essentially predict a graph structure first and then use it for the functional regression problem. Sparse OpMINRES-L-D and Sparse Non-Pos-OpMINRES-L-D present a unified approach which incorporates sparse graph learning with the functional regression task. In Table 1, the learned graphs are illustrated where darker colors of edges indicate larger edge weights. Table 1 illustrates that fglasso-OpMINRES-D fails to differentiate between the interactions of input functions and results in a fully connected graph consistently. Though, KGL-OpMINRES-L-D in comparison to fglasso-OpMINRES-L-D produces a sparser graph structure, Sparse-OpMINRES-L-D learns sparse graph structures exhibiting relations that can incorporate the associations enforced in the generation process. The learned associations provide required correlations which can benefit the functional regression task. Further details of the synthetic data including experiments for 3 nodes and 25 nodes are given in Appendix A.10. Sparse Non-Pos-OpMINRES-L-D also learns sparse graph structures which are informative of synthetic data used for the functional regression task.

Table 1: Graphs corresponding to learned $L$ for 12-node synthetic data. [Best viewed in color]

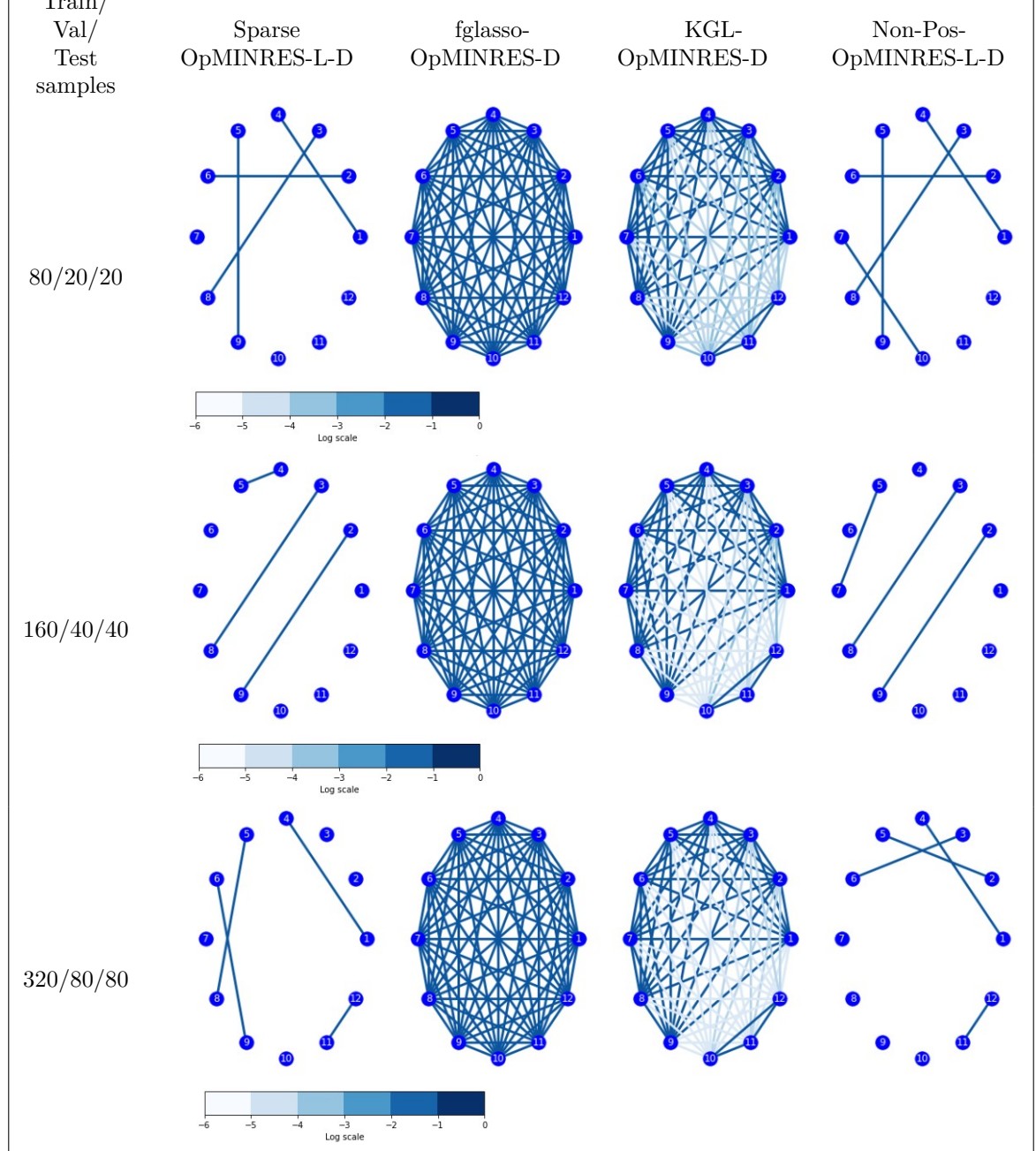

**Additional Experiments:** Appendix A.10.5 contains the results of experiments conducted as an ablation study in the 12-node setting.

## 7.2 Experiments on weather data

Weather data is dynamic and inter-relationships between different parameters can be hard to predict. As our problem solves a functional regression problem based on a relationship between a set of input functions, we intend to showcase the effectiveness of the proposed algorithm by predicting average dew-point temperature

Table 2: Mean RSSE results for 12-node synthetic data.

| Train/Val/Test samples | Methods | Mean RSSE | | |
|---|---|---|---|---|
| | | Train | Val | Test |
| 80/20/20 | Sparse OpMINRES-L-D | 1.140691 | 1.780445 | 1.583640 |
| | fglasso-OpMINRES-D | 1.243734 | 1.821687 | 1.700265 |
| | KGL-OpMINRES-D | 1.061473 | 1.775388 | 1.554853 |
| | Sparse Non-Pos-OpMINRES-L-D | 1.167264 | 1.806093 | 1.618175 |
| 160/40/40 | Sparse OpMINRES-L-D | 0.888574 | 1.229568 | 1.385952 |
| | fglasso-OpMINRES-D | 0.956907 | 1.285154 | 1.305025 |
| | KGL-OpMINRES-D | 0.983432 | 1.260719 | 1.286481 |
| | Sparse Non-Pos-OpMINRES-L-D | 1.154356 | 1.362239 | 1.417921 |
| 320/80/80 | Sparse OpMINRES-L-D | 1.062102 | 1.294110 | 1.239181 |
| | fglasso-OpMINRES-D | 0.947426 | 1.336192 | 1.271646 |
| | KGL-OpMINRES-D | 0.980995 | 1.299266 | 1.252706 |
| | Sparse Non-Pos-OpMINRES-L-D | 1.073292 | 1.295140 | 1.243346 |

(F) across 12 weather stations based on their respective air temperatures (F). We consider 1 minute data of Wyoming ASOS data collected from IEM ASOS One Minute Data (Iowa Environmental Mesonet, 2022). The data has been collected for an interval of 2 hours for both input functions and output function from January, 2022 to August, 2022. Data collected at one minute interval for different 12 weather stations in Wyoming was pre-processed to create 2 hour interval data by disregarding intervals where data was missing in any of the 12 stations. A total of 718 samples have been collected after removing missing data.

For all the methods, a truncated trigonometric basis of $L^2([0, 1])$ with 80 basis functions has been considered for encoding the functional data. We segregate the weather data experiments into small weather data experiments (Appendix A.10) by considering 120 samples and full weather data experiments. The following random data splits have been considered: (80/20/20) and (472/123/123), representing the number of training samples/validation samples/test samples in small weather data and full weather data settings, respectively.

Tables 3-4 showcase the performance of the algorithms for full weather data considering all 718 samples. Sparse OpMINRES-L-D performs the best in terms of mean RSSE on the test data compared to fglasso-OpMINRES-L-D and KGL-OpMINRES-L-D (Table 4). The maps in Table 3 describe the geographic positioning of the weather stations in Wyoming and the edges between them indicate potential inter-relations between the stations. In Table 3, fglasso-OpMINRES-L-D and KGL-OpMINRES-L-D learn dense fully connected graphs which do not provide much information regarding the impact of different weather stations on the relationship of respective air temperature to the average dew point temperature. Sparse OpMINRES-L-D learns a sparse $L$ where stations BPI(1) and CPR(2), P60(7) and SHR(10) along with RIW(8) and WRL(12) are connected. BPI(1) $(42.58507, -110.11115)$ and CPR(2) $(42.908, -106.46442)$ are 300.7 km apart with an elevation of 2124 m and 1612 m, respectively. P60(7) $(44.54444, -110.42111)$ and SHR(10) $(44.77, -106.97)$ are 274.85 km apart with an elevation of 2368 m and 1209 m, respectively. RIW(8) $(43.06423, -108.45984)$ and WRL(12) $(43.96571, -107.95083)$ are 108.28 km apart with an elevation of 1688 m and 1294 m. It can be observed that the connections in the learned graph structure have been established between stations with varying elevations lying in close proximity latitude-wise.

To illustrate the utility of our proposed sample-based approximation algorithm, we use the full weather data and evaluate it to produce the results in Table 5 for all algorithms. The results in Table 5 show that the sample-based approximation algorithm provides comparable results using only a few samples. In 5 runs, out of 472 training samples, the number of samples in the working set of the sample-based approximation algorithm varies between 123 to 200. Sparse OpMINRES-L-D performs the best in terms of the mean RSSE on test data.

Table 3: Graphs corresponding to learned $L$ for full weather (472/123/123) data. [Best viewed in color]

Table 4: Mean RSSE results for full weather data.

| Train/Val/Test samples | Methods | Mean RSSE | | |
|---|---|---|---|---|
| | | Train | Val | Test |
| 472/123/123 | Sparse OpMINRES-L-D | 0.002938 | 0.009891 | 0.010743 |
| | fglasso-OpMINRES-D | 0.021094 | 0.013476 | 0.044216 |
| | KGL-OpMINRES-D | 0.003474 | 0.010877 | 0.012797 |

Table 5: RSSE (mean ± standard deviation) results over 5 runs for sample-based approximation algorithm using full weather data.

| Methods | Mean RSSE | | | |
|---|---|---|---|---|
| | Full Train | Train subset | Val | Test |
| Sparse OpMINRES-L-D | $0.083688 \pm 0.011229$ | $0.010584 \pm 0.006664$ | $0.013568 \pm 0.000914$ | $0.097118 \pm 0.01392$ |
| fglasso-OpMINRES-D | $0.112057 \pm 0.004981$ | $0.011457 \pm 0.006908$ | $0.014289 \pm 0.000710$ | $0.130591 \pm 0.006171$ |
| KGL-OpMINRES-D | $0.131076 \pm 0.016523$ | $0.010166 \pm 0.005781$ | $0.013702 \pm 0.000561$ | $0.100619 \pm 0.017939$ |

Table 6: Mean RSSE results for NBA data.

| Train/Val/Test samples | Methods | Mean RSSE | | |
|---|---|---|---|---|
| | | Train | Val | Test |
| 233/59/59 | Sparse OpMINRES-L-D | 0.025200 | 0.087748 | 0.106344 |
| | OpMINRES-D | 0.023261 | 0.191459 | 0.265513 |

Table 7: RSSE (mean ± standard deviation) results for sample-based approximation algorithm for NBA data.

| Methods | Mean RSSE | | | |
|---|---|---|---|---|
| | Full Train | Train subset | Val | Test |
| Sparse OpMINRES-L-D | $0.215943 \pm 0.002837$ | $0.018371 \pm 0.001624$ | $0.066631 \pm 0.003058$ | $0.147214 \pm 0.004665$ |
| OpMINRES-D | $9.070595 \pm 0.138830$ | $0.005005 \pm 0.002031$ | $0.180238 \pm 0.013534$ | $0.452081 \pm 0.020617$ |

## 7.3 Experiments on NBA data

The movement of basketball and 21 players involved on the court (`x-y` coordinates) in the Atlanta Hawks (ATL) vs Utah Jazz (UTA) match on November 15, 2015 has been considered in this experiment. This data is available in the Github repo NBA Movement Data (Seward, 2018). The data has been collected for different plays for both input functions of 21 players and output function denoting the position of the ball, which includes missing data corresponding to some players in different plays. As plays in a basketball game are of different time duration, we use a truncated trigonometric basis of $L^2([0, 1])$ with 80 basis functions to sample the functions at fixed 100 points on $[0, 1]$. A total of 351 samples have been collected after removing missing data. A random data split of $(233/59/59)$ representing the number of training samples/validation samples/test samples has been considered. The problem requires solving a multi-dimensional functional regression problem which is incompatible with fglasso and KGL algorithms, as both fglasso & KGL are based on single dimensional input functions. Hence, we compare our method with the algorithm OpMINRES-D where a fixed $L$ is incorporated in our alternating minimization framework.

**OpMINRES-D**: A fixed $L$ is considered corresponding to a fully connected network of 21 nodes. This decision was made as fglasso mostly learns a fully connected graph in earlier experiments. Thus, a fixed $L$ (with no sparsity-inducing MCP) is used in the proposed alternating minimization regime for optimizing **u** and $D$. OpMINRES is used with $k_1(x, x'; G) = e^{-\gamma_x(x-x')^\top (L+D)(x-x') - \gamma_y(x-x')^\top (L+D)(x-x')}$ and $k_2^1(s, t) = e^{-\gamma_{op}^1|s-t|}, k_2^2(s, t) = e^{-\gamma_{op}^2|s-t|}$, where $\gamma_x, \gamma_y \in \{10^{-6}, 10^{-5}, 10^{-4}, 10^{-3}, 10^{-2}, 10^{-1}, 1, 10, 100\}$ and $\gamma_{op}^1, \gamma_{op}^2 \in \{10^{-6}, 10^{-5}, 10^{-4}, 10^{-3}, 10^{-2}, 10^{-1}, 1, 10, 100\}$.

**Sparse OpMINRES-L-D**: We consider the graph-induced operator-valued kernels using $k_1(x, x'; G) = e^{-\gamma_x(x-x')^\top (L+D)(x-x') - \gamma_y(x-x')^\top (L+D)(x-x')}$ and $k_2^1(s, t) = e^{-\gamma_{op}^1|s-t|}, k_2^2(s, t) = e^{-\gamma_{op}^2|s-t|}$, where $\gamma_x, \gamma_y \in \{10^{-6}, 10^{-5}, 10^{-4}, 10^{-3}, 10^{-2}, 10^{-1}, 1, 10, 100\}$ and $\gamma_{op}^1, \gamma_{op}^2 \in \{10^{-6}, 10^{-5}, 10^{-4}, 10^{-3}, 10^{-2}, 10^{-1}, 1, 10, 100\}$. Projected gradient descent is used in minimization with respect to $L$ and $D$ based on a decaying step-size. The sparsity is aided by the MCP regularization considered in learning of $L$.

The results are illustrated in Tables 6 and 8 where comparison method OpMINRES-D uses a fully connected graph, however Sparse OpMINRES-L-D performs better with a sparse learned graph in terms of mean RSSE on the test data. Observations for the match have been published in the match reports ESPN match recap and ESPN match scoreboard (ESPN, 2015a;b). In Table 8, the depiction of a basketball court is provided where the players have been arranged on the court with ATL players on the left and UTA players on the right. The graphical structure corresponding to the learned $L$ in Table 8 illustrates strategic relationships between players of both ATL and UTA. The connection between Derrick Favors—Trevor Booker (6—8) had been pivotal for Utah Jazz. The performance of Al Horford in (4—7) and Kent Bazemore in (2—11) for Atlanta Hawks has been captured. Though the partnership of Alec Burks—Trey Burke (9—18) for Utah

Jazz is not evident in the match reports, their ball carrying interactions may be the reason for being learned in $L$.

Table 8: Graphs corresponding to learned $L$ for NBA data (233/59/59) of ATL (left) vs UTA (right) match using Sparse OpMINRES-L-D. [Best viewed in color]

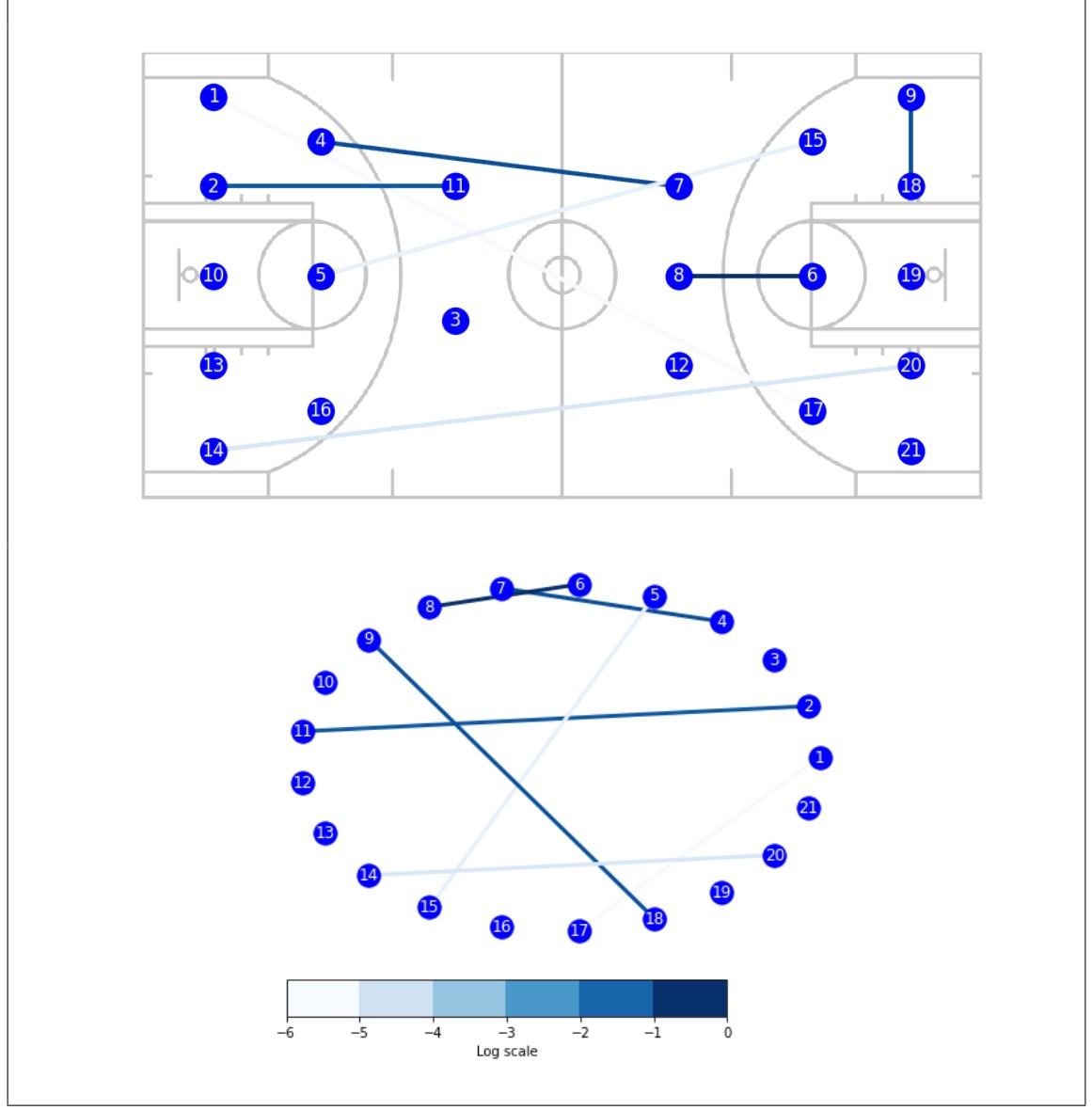

The performance of sample-based approximation algorithm has been showcased on NBA data in Table 7. Using the sample-based approximation algorithm, out of 233 training samples, the number of samples chosen in the working set of samples in 5 runs varies between 57 to 62. Sparse OpMINRES-L-D performs the best in terms of the mean RSSE on test data. The remaining results and details of experiments are in Appendix A.10. The best hyperparameters for the experiments conducted have been listed in Appendix A.10.4.

## 8   Conclusion

In this work, we incorporate learning of a suitable graphical structure which drives a functional regression problem where the output function depends on the input functions and also upon their inter-relationships with each other. An alternating minimization based algorithm has been proposed to learn the Laplacian matrix $L$, a non-negative diagonal matrix $D$ characterizing the graphical structure, along with the map from input space to the output space. For a fixed $L$ and $D$, the functional regression learning problem is formulated as an operator system of equations which is solved by using OpMINRES algorithm. Projected gradient descent is used to learn the Laplacian matrix and the non-negative diagonal matrix in the alternating minimization framework. A sparsity-inducing regularizer (e.g. MCP) in $L$ has been incorporated during the alternating minimization, which helps in learning a graphical structure and allows for improved interpretability and can highlight interactions which are most relevant among input functions useful for the prediction. To make the proposed algorithm scalable, a sample-based approximation algorithm has been proposed which helps reduce the computations required for solving linear system of operator equations using OpMINRES algorithm. An extension of the alternating minimization framework has also been proposed to solve the multi-dimensional functional regression problem assuming a single graphical structure on the input variables. The generalization analysis provides a bound on generalization error for learning a graph-induced OVK. Experiments establish the utility of proposed graph-induced operator-valued kernels in functional regression problems from diverse applications.

### Broader Impact Statement

The framework and algorithms introduced in the paper with graph-induced operator-valued kernels aid in learning a sparse graphical structure which drives a functional regression problem where the output function depends on the input functions and their inter-relationships with each other. This will promote research in investigating more sophisticated techniques for handling functional data with an inherent graphical structure ingrained among them. To the best of our knowledge, our work does not have any negative impact.

### Acknowledgments

We thank our anonymous reviewers for their insightful comments and suggestions. We declare no competing interests.

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

# A   Appendix

## A.1   Functonal Regression with Known Graph Structure

In this section, we motivate functional regression problem with known graph structure. Consider a system where a set of input functions determines the output (or response) function. Let the system be modeled based on $p$ input functional variables $x_1(t), x_2(t), \ldots, x_p(t)$, where $x_i \in L^2([0,1])$, $i \in [p]$. A functional response variable $y(t)$ is used to model output of the system where $y \in L^2([0,1])$. (Note that $[0,1]$ can be replaced with any closed time interval based on the application.)

The undirected graph structure of the functional input variables is represented by $G = (V, E)$, where $V = \{v_1, v_2, \ldots, v_p\}$ and $E = \{\{v_i, v_j\} | v_i$ is connected to $v_j, 1 \le i, j \le p\}$ is the edge set which characterizes the underlying relationship between the variables. Note that the notation for an edge uses an unordered pair $\{v_i, v_j\}$ which characterizes the undirected nature of the graph $G$. In order to model the relation between functional input variables $x_1, x_2, \ldots, x_p$ and functional response variable $y$, we use the following map $F$:

$$y = F(x_1, x_2, \ldots, x_p, G). \tag{102}$$

Note that $F$ now depends explicitly on the graph $G$ in addition to the input functions $x_1, x_2, \ldots, x_p$. Here, we consider a scenario where $G$ is known. Recall the example of a manufacturing factory, where the output of emissions depends on the metrics of different components involved in the manufacturing process. The graph $G$ is determined in this case by understanding the components which are connected during the manufacturing process.

We consider the notations $x = (x_1, x_2, \ldots, x_p) \in \mathcal{X} \ (= (L^2([0,1]))^p)$ and $y \in \mathcal{Y} \ (= L^2([0,1]))$ to represent an arbitrary sample $(x, y)$. To learn the mapping $F$, consider the training data of $n$ samples given as $\{(x^{(i)}, y^{(i)})\}_{i=1}^n$, where $x^{(i)} = (x_1^{(i)}, x_2^{(i)}, \ldots, x_p^{(i)}) \in \mathcal{X}$ and $y^{(i)} \in \mathcal{Y}$. In order to learn $F$, we develop an operator-valued kernel which can leverage the structural information of $G$.

Towards this, we first introduce an operator-valued kernel which maps the elements of $\mathcal{X} \times \mathcal{X}$ to a set of bounded linear operators over the output space $\mathcal{Y}$, denoted by $\mathcal{L}(\mathcal{Y})$. We formally define OVK as follows.

**Definition A.1** (**Operator-valued Kernel**). (Kadri et al., 2016) An $\mathcal{L}(\mathcal{Y})$-valued kernel $K$ on $\mathcal{X}^2$ is a function $K(.,.) : \mathcal{X} \times \mathcal{X} \to \mathcal{L}(\mathcal{Y})$, satisfying the following properties:

1. $K$ is Hermitian, that is $\forall w, z \in \mathcal{X}, K(w, z) = K(z, w)^*$ (* denotes the adjoint operator),

2. $K$ is positive semi-definite on $\mathcal{X}^2$, that is $K$ is Hermitian and for every natural number $r$ and all $\{(w^{(i)}, u^{(i)})_{i \in [r]}\} \in \mathcal{X} \times \mathcal{Y}$, the matrix with $(i, j)$-th entry given by $\langle K(w^{(i)}, w^{(j)}) u^{(i)}, u^{(j)} \rangle_{\mathcal{Y}}$ is positive semi-definite.

Constructing an operator-valued kernel based on Definition A.1 is a challenge as verifying both properties of being Hermitian and positive semi-definiteness becomes non-trivial. A construction of OVK that satisfies both the properties in Definition A.1 has been proposed in (Lian, 2007; Kadri et al., 2016). The OVK construction in (Lian, 2007; Kadri et al., 2016) uses a scalar-valued kernel $k_1$ on $\mathcal{X} \times \mathcal{X}$ and a Hilbert-Schmidt integral (HSI) operator defined on the output space $\mathcal{Y}$, and is given as follows:

$$(K(x, x')y)(t) = k_1(x, x') \int_0^1 k_2(s, t)y(s)ds, \tag{103}$$

where $k_2$ inside the HSI operator $\int_0^1 k_2(s, t)y(s)ds$ is a scalar-valued kernel on $\mathbb{R} \times \mathbb{R}$. If $k_1$ is positive semi-definite and if $k_2$ is positive semi-definite (implying that the HSI operator is positive semi-definite), then the construction in (103) is known to be positive semi-definite (Kadri et al., 2016). We will now adapt the OVK in (103) to include the graph structure information present in $G$. An obvious choice for using the influence of graphical structure $G$ in the functional regression task is to use the adjacency matrix of $G$, but the adjacency matrix of $G$ not being necessarily positive semi-definite makes its utility restrictive. The Laplacian matrix of a graph $G$, on the other hand is useful in this respect as it has the desired property of

being positive semi-definite which is useful in a kernel-based learning framework. The Laplacian matrix of an undirected graph $G = (V, E)$, with $V$ as the node set and $E$ as edge set is defined as $L = \mathbb{D} - A$, where $\mathbb{D} = \text{diag}(\deg(v_1), \deg(v_2), \ldots, \deg(v_p))$ is the degree matrix and $A$ is the adjacency matrix of the graph $G$. The elements of $L$ are given by

$$
L_{i,j} = \begin{cases} \deg(v_i), & \text{if } i = j, \\ -1, & \text{if } i \neq j \text{ and } \{v_i, v_j\} \in E, \\ 0, & \text{otherwise.} \end{cases} \tag{104}
$$

We propose to incorporate the graphical structure in Equation (103) within the scalar-valued kernel $k_1$ itself, as follows:

$$
k_1(x, x'; G) = \gamma x^\top L x', \ \gamma > 0, \tag{105}
$$

$$
= \gamma \sum_{i,j=1}^{p} \int_0^1 x_i(t) L_{ij} x_j'(t) dt. \tag{106}
$$

The construction of $k_1$ in (105) involves capturing the graphical structure of $G$ using $L$ in $x^\top L x'$. We use an equivalent expression $\langle x, Lx' \rangle_p := x^\top L x'$, where the inner product $\langle \cdot, \cdot \rangle_p : \mathcal{X} \times \mathcal{X} \to \mathbb{R}$ is defined as $\langle x, x' \rangle_p = \sum_{i=1}^{p} \int_0^1 x_i(t) x_i'(t) dt$. The inner product $\langle x, x' \rangle_p$ measures the similarity between $x$ and $x'$ in $\mathcal{X}$. Let $L = [L_1, L_2, \ldots, L_p]$, where $L_i$ represents the $i$-th column of $L$, then $Lx'$ is computed based on standard matrix-vector multiplication where elements of $L$ are multiplied with $x'$ by using scalar multiplication and addition of functions as $Lx' = \sum_{i=1}^{p} L_i x_i'$. Equation (105) provides a tool to measure similarity between $x, x' \in \mathcal{X}$ where the interactions are encoded in the underlying graph $G$ using the Laplacian matrix $L$. If $k_1$ defined in (105) can be proved to be a valid positive semi-definite scalar-valued kernel on $\mathcal{X} \times \mathcal{X}$, then an OVK can be defined similar to the construction in Equation (103). Next, we prove that such a construction is indeed possible.

**Proposition A.1.** For an underlying graph $G = (V, E)$ with $|V| = p$, functional variables $x = (x_1, x_2, \ldots, x_p) \in \mathcal{X} (= (L^2([0, 1]))^p)$ and a functional response variable $y \in \mathcal{Y} (= L^2([0, 1]))$, consider an operator-valued kernel $K : \mathcal{X} \times \mathcal{X} \to \mathcal{L}(\mathcal{Y})$ defined as

$$
(K(x, x')y)(t) = k_1(x, x'; G) \int_0^1 k_2(s, t) y(s) ds,
$$

where $k_1(x, x'; G) = \gamma x^\top L x', \gamma > 0$ and $k_2$ is a positive semi-definite scalar-valued kernel on $\mathbb{R} \times \mathbb{R}$. Then $K$ is positive semi-definite.

*Proof.* Please see Appendix A.3 for the proof. □

Recall the construction of scalar-valued radial basis function (RBF) kernel defined over $\mathbb{R}^d \times \mathbb{R}^d$ ($d \in \mathbb{Z}_+$) as $e^{-\gamma \|\mathbb{x} - \mathbb{x}'\|^2}$, for $\gamma > 0$, $\mathbb{x}, \mathbb{x}' \in \mathbb{R}^d$ based on the kernel $\mathbb{x}^\top \mathbb{x}$. Similar to that construction, we now describe an extension for kernel $k_1$ defined in (105). The kernel notation $k_1(x, x'; G) = \gamma x^\top L x'$ is overloaded to represent the following RBF-type kernel:

$$
k_1(x, x'; G) = e^{-\gamma (x - x')^\top L (x - x')}, \ \gamma > 0. \tag{107}
$$

capturing the interaction of $x, x'$ using $L$ (see Appendix A.3). The RBF-type kernel in (107) is an improved version of the kernel in (105), as it can approximate higher dimensional relationships better owing to the exponential nature and shift invariant property given by $k_1(x+h, x'+h; G) = k_1(x, x'; G), \forall x, x', h \in \mathcal{X}$. Note that when computing $(x - x')^\top L (x - x')$, where $x = (x_1, \ldots, x_p), x' = (x_1', \ldots, x_p') \in \mathcal{X}$, the interactions between the unlike pair $(x_i, x_j'), i \neq j$ would negate the influence of the like pair $(x_i, x_i')$, because of the structure of $L$ (defined in (104)). Therefore, we propose to use a diagonally perturbed Laplacian (Bapat et al., 2001) to aid the functional regression task performance. Perturbed Laplacians have found applications in spectral clustering, analysis of graphs (Kurras et al., 2014) and missing link prediction in networks

(Aliakbarisani et al., 2022). A natural perturbation of Laplacian for the scalar-valued kernel $k_1$ in (107) involves the degree matrix $\mathbb{D}$ leading to the following definition for $k_1$:

$$k_1(x, x'; G) = e^{-\gamma(x-x')^\top (L+\mathbb{D})(x-x')}, \ \gamma > 0, \ \text{for } x, x' \in \mathcal{X}. \tag{108}$$

Incorporating $\mathbb{D}$ with $L$ in $k_1$ improves the representation of individual components of $x, x'$ in prediction of $y$ as the like pair $(x_i, x'_i)$ gets weighed by $D_{i,i} + L_{i,i}$ in the kernel expression which compensates for the negation effect described above. This enables us to define a family of operator-valued kernels discussed next, which is induced by the graphical structure information.

**Definition A.2 (Graph-induced Operator-valued Kernel for known $G$).** A graph-induced operator-valued kernel is defined as

$$(K^G(x, x')y)(t) = k_1(x, x'; G) \int_0^1 k_2(s, t)y(s)ds, \tag{109}$$

where $k_2$ is a scalar-valued kernel on $\mathbb{R}^2$, $G$ is a graph associating the $p$ input functions in $(x_1, \ldots, x_p) \in \mathcal{X}$ and $k_1$ is defined as

$$k_1(x, x'; G) = e^{-\gamma(x-x')^\top (L+\mathbb{D})(x-x')},$$

for $\gamma > 0$ where $L$ is the Laplacian matrix and $\mathbb{D}$ is the degree matrix of the graph $G$.

$K^G$ associates a pair $x, x' \in \mathcal{X}$ with output function $y \in \mathcal{Y}$ where $G$ is the graph which incorporates the interaction of $p$ constituent input functions of $x$ and $x'$. The addition of $\mathbb{D}$ to $L$ in (108) preserves the positive semi-definiteness of the kernel $k_1$ as $\mathbb{D}$ is a diagonal matrix with positive entries. Using graph-induced OVK for functional regression problem requires associating $K^G$ with a function-valued reproducing kernel Hilbert space where the map $F$ from (102) resides. The existence of a bijection between the set of positive semi-definite (Mercer) operator-valued kernels and function-valued reproducing kernel Hilbert spaces has been established in (Kadri et al., 2016). A function-valued reproducing kernel Hilbert space (RKHS) is defined as follows.

**Definition A.3 (Function-valued RKHS).** (Kadri et al., 2016) A Hilbert space $\mathcal{H}$ of functions from $\mathcal{X}$ to $\mathcal{Y}$ is called a reproducing kernel Hilbert space if there is a positive semi-definite $\mathcal{L}(\mathcal{Y})$-valued kernel $K$ on $\mathcal{X}^2$ such that:

1. the function $z \mapsto K(w, z)g$ belongs to $\mathcal{H}$, $\forall w, z \in \mathcal{X}$ and $\forall g \in \mathcal{Y}$,

2. for every $F \in \mathcal{H}, w \in \mathcal{X}$ and $g \in \mathcal{Y}, \langle F, K(w, .)g \rangle_{\mathcal{H}} = \langle F(w), g \rangle_{\mathcal{Y}}$.     **(reproducing property)**

Property 1 in Definition A.3 provides an association of OVK $K$ with the space $\mathcal{H}$ which contains maps from $\mathcal{X}$ to $\mathcal{Y}$. The reproducing property in Definition A.3 helps to relate the inner product in $\mathcal{H}$ to the inner product in $\mathcal{Y}$. Using Proposition A.1 and Definition A.3, the positive semi-definiteness of $K^G$ constructed using (109) ensures that there exists a unique RKHS $\mathcal{H}_{K^G}$ corresponding to $K^G$ (Theorem 1 (Kadri et al., 2016)). This enables us to formulate a learning problem in $\mathcal{H}_{K^G}$ as follows:

$$\widetilde{F}_\lambda = \arg\min_{F \in \mathcal{H}_{K^G}} \sum_{i=1}^n \|y^{(i)} - F(x^{(i)})\|_{\mathcal{Y}}^2 + \lambda \|F\|_{\mathcal{H}_{K^G}}^2, \tag{110}$$

where $\mathcal{H}_{K^G}$ is the function-valued RKHS induced by the graph-induced operator-valued kernel $K^G$ and $\| \cdot \|_{\mathcal{H}_{K^G}}$ denotes the norm in $\mathcal{H}_{K^G}$. In order to solve the optimization problem, we utilize the reproducing property of operator-valued kernel (Definition A.3) given by

$$\langle F, K^G(x, .)h \rangle_{\mathcal{H}_{K^G}} = \langle F(x), h \rangle_{\mathcal{Y}}, \ \forall x \in \mathcal{X}, h \in \mathcal{Y}. \tag{111}$$

The minimization problem in (110) is not tractable using a search based procedure over $\mathcal{H}_{K^G}$, hence the reproducing property of operator-valued kernel $K^G$ in (111) can be leveraged to simplify the problem and characterize the solution of (110) using elements of the output space $\mathcal{Y}$. We now provide a representer theorem for the minimization problem (110) in the function-valued RKHS $\mathcal{H}_{K^G}$ corresponding to the operator-valued kernel $K^G$.

**Theorem A.2** (**Representer theorem**). Let $K^G$ be an operator-valued kernel and $\mathcal{H}_{K^G}$ be its corresponding function-valued reproducing kernel Hilbert space. The solution $\widetilde{F}_\lambda \in \mathcal{H}_{K^G}$ of the regularized optimization problem: $\widetilde{F}_\lambda = \arg\min_{F \in \mathcal{H}_{K^G}} \sum_{i=1}^n \|y^{(i)} - F(x^{(i)})\|_{\mathcal{Y}}^2 + \lambda \|F\|_{\mathcal{H}_{K^G}}^2$, where $\lambda > 0, F \in \mathcal{H}_{K^G}$, has the following form

$$\widetilde{F}_\lambda(.) = \sum_{i=1}^n K^G(x^{(i)}, .)u_i, \text{ where } u_i \in \mathcal{Y}. \tag{112}$$

*Proof.* Please find the proof in Appendix A.4. $\qquad\square$

Using the representer theorem with reproducing property of operator-valued kernel, we provide below (Appendix A.8) the linear system of operators to determine $u_i$, for $i \in [n]$:

$$(\mathbf{K} + \lambda I)\mathbf{u} = \mathbf{y}, \tag{113}$$

where $\mathbf{K}$ is a block operator matrix given by $\mathbf{K}_{i,j} = K^G(x^{(i)}, x^{(j)})$, $\mathbf{u} = [u_1, u_2, \ldots, u_n]^\top$ and $\mathbf{y} = [y^{(1)}, y^{(2)}, \ldots, y^{(n)}]^\top$. A simple inversion of $\mathbf{K} + \lambda I$ may not be straightforward to obtain $\mathbf{u}$ in (113), for an arbitrary choice of OVK $K^G$. Saha & Palaniappan (2020) proposed an iterative operator minimum residual (OpMINRES) algorithm which adapts a Krylov subspace minimal residual (MINRES) algorithm to solve operator-based linear system of the form in (113). We delve deeper into the details of OpMINRES in Section 5.1.1. With the learned $\mathbf{u}$ obtained by solving (113), for any sample $\hat{x} \in \mathcal{X}$, the prediction is given by

$$\widetilde{F}_\lambda(\hat{x}) = \sum_{i=1}^n K^G(x^{(i)}, \hat{x})u_i, \text{ where } u_i \in \mathcal{Y}. \tag{114}$$

In (114), functions $u_i \in \mathcal{Y}$ for $i \in [n]$, can be considered as basis functions for the space $\mathcal{Y}$ and operators $K^G(x^{(i)}, \hat{x})$ for $i \in [n]$, correspond to operator-valued coefficients of the basis functions. The term $K^G(x^{(i)}, \hat{x})u_i$ amounts to the total contribution of sample $x^{(i)}$ in determining the prediction for $\hat{x}$.

## A.2 Mathematical Derivations

We cover derivations which can transform the objective function where the optimization takes place over the output space $\mathcal{Y}$ instead of the RKHS $\mathcal{H}_K$ induced by the OVK $K$. The following derivation has been referred in Section 5.1 to obtain (5) from (4), where we simplify the expression $J(F, L, D)$ to obtain $J(\mathbf{u}, L, D)$.

$$\min_{F,L,D} J(F, L, D) = \sum_{i=1}^n \|y^{(i)} - F(x^{(i)})\|_{\mathcal{Y}}^2 + \lambda \|F\|_{\mathcal{H}_K}^2 + \rho_L \sum_{i=1}^n x^{(i)^\top} L x^{(i)} + \rho_D \|D\|_F^2 \tag{115}$$

$$\implies \min_{\mathbf{u},L,D} J(\mathbf{u}, L, D) = \sum_{i=1}^n \left\| y^{(i)} - \sum_{j=1}^n K(x^{(i)}, x^{(j)})u_j \right\|_{\mathcal{Y}}^2 + \lambda \left\langle \sum_{i=1}^n K(x^{(i)}, .)u_i, \sum_{j=1}^n K(x^{(j)}, .)u_j \right\rangle_{\mathcal{H}_K} \tag{116}$$
$$+ \rho_L \sum_{i=1}^n x^{(i)^\top} L x^{(i)} + \rho_D \|D\|_F^2$$

$$= \sum_{i=1}^n \left\| y^{(i)} - \sum_{j=1}^n K(x^{(i)}, x^{(j)})u_j \right\|_{\mathcal{Y}}^2 + \lambda \sum_{i,j=1}^n \langle K(x^{(i)}, x^{(j)})u_i, u_j \rangle_{\mathcal{Y}} \tag{117}$$
$$+ \rho_L \sum_{i=1}^n x^{(i)^\top} L x^{(i)} + \rho_D \|D\|_F^2.$$

The expression (116) is obtained from (115) by using the representer theorem and the reproducibility property of $K$ is utilized to obtain (117).

Similarly, we obtain the objective function from (29) in Section 5.1.5 using representer theorem and the reproducibility property of $K$ given as follows:

$$J(F^1, F^2, \ldots, F^s, L, D) = \sum_{l=1}^{s} \left( \sum_{i=1}^{n} \|y_l^{(i)} - F^l(\mathbf{x}^{(i)})\|_{\mathcal{Y}}^2 + \lambda \|F^l\|_{\mathcal{H}_K^l}^2 \right)$$
$$+ \sum_{l=1}^{r} \rho_l \left( \sum_{i=1}^{n} x_l^{(i)^\top} L x_l^{(i)} \right) + \rho_D \|D\|_F^2$$

$$\implies J(\mathbf{u}^1, \mathbf{u}^2, \ldots, \mathbf{u}^s, L, D) =$$

$$\sum_{l=1}^{s} \left( \sum_{i=1}^{n} \left\| y_l^{(i)} - \sum_{j=1}^{n} K^l(\mathbf{x}^{(i)}, \mathbf{x}^{(j)}) u_j^l \right\|_{\mathcal{Y}}^2 + \lambda \sum_{i,j=1}^{n} \langle K^l(\mathbf{x}^{(i)}, \mathbf{x}^{(j)}) u_i^l, K^l(\mathbf{x}^{(j)}, .) u_j^l \rangle_{\mathcal{Y}} \right) \tag{118}$$

$$+ \sum_{k=1}^{r} \left( \rho_k \sum_{i=1}^{n} x_k^{(i)^\top} L x_k^{(i)} \right) + \rho_D \|D\|_F^2$$

$$= \sum_{l=1}^{s} \left( \sum_{i=1}^{n} \left\| y_l^{(i)} - \sum_{j=1}^{n} K^l(\mathbf{x}^{(i)}, \mathbf{x}^{(j)}) u_j^l \right\|_{\mathcal{Y}}^2 + \lambda \sum_{i,j=1}^{n} \langle K^l(\mathbf{x}^{(i)}, \mathbf{x}^{(j)}) u_i^l, u_j^l \rangle_{\mathcal{Y}} \right) \tag{119}$$

$$+ \sum_{k=1}^{r} \left( \rho_k \sum_{i=1}^{n} x_k^{(i)^\top} L x_k^{(i)} \right) + \rho_D \|D\|_F^2.$$

### A.3 Positive semi-definiteness of OVK

In this section, we cover the proof of Proposition A.1 in Section A.1 which helps in building a graph-induced OVK. The major contribution is based on showing that the proposed scalar-valued kernel in graph-induced OVK is a valid positive semi-definite kernel on $\mathcal{X} \times \mathcal{X}$. We recall Proposition A.1 below.

**Proposition A.3.** For an underlying graph $G = (V, E)$ with $|V| = p$, functional variables $x = (x_1, x_2, \ldots, x_p) \in \mathcal{X}(= (L([0, 1]))^p)$ and a functional response variable $y \in \mathcal{Y}(= L^2([0, 1]))$, consider an operator-valued kernel $K : \mathcal{X} \times \mathcal{X} \to \mathcal{L}(\mathcal{Y})$ defined as

$$(K(x, x')y)(t) = k_1(x, x'; G) \int_0^1 k_2(s, t)y(s)ds,$$

where $k_1(x, x'; G) = \gamma x^\top L x', \gamma > 0$ and $k_2$ is a positive semi-definite scalar-valued kernel on $\mathbb{R} \times \mathbb{R}$. Then $K$ is positive semi-definite.

*Proof.* In order to prove the positive semi-definiteness of $K$, it is sufficient to prove the positive semi-definiteness of $k_1$ based on the construction followed in (Kadri et al., 2016). We focus on the term $x^\top L x'$ in $k_1$. Let $\{x^1, x^2, \ldots, x^l\} \subset \mathcal{X}$ be a finite set of points. Consider any vector $\alpha \in \mathbb{R}^l$ and $\mathbf{K}$ be a $l \times l$ kernel matrix given by $\mathbf{K} = [k_1(x^i, x^j; G)]_{i,j}$.

Now, we recall that the space $L^2([0, 1])$ has the following inner product and norm:

$$\langle f, g \rangle = \int_0^1 f(t)g(t)dt, \ f, g \in L^2([0, 1]),$$

$$\|f\| = \left( \int_0^1 f^2(t)dt \right)^{1/2}.$$

We define $\langle ., . \rangle_p : \mathcal{X} \times \mathcal{X} \to \mathbb{R}$ as

$$\langle x, w \rangle_p = \langle x_1, w_1 \rangle + \langle x_2, w_2 \rangle + \cdots + \langle x_p, w_p \rangle \tag{120}$$

$$= \sum_{i=1}^{p} \int_0^1 x_i(t)w_i(t)dt, \tag{121}$$

where $x = (x_1, x_2, \ldots, x_p) \in \mathcal{X}$, $w = (w_1, w_2, \ldots, w_p) \in \mathcal{X}$. We can show that $\langle ., . \rangle_p$ is an inner product on $\mathcal{X}$ over the field $\mathbb{R}$.

- The symmetry of $\langle ., . \rangle_p$ follows from the definition in equation (121) as:

$$\langle x, w \rangle_p = \langle w, x \rangle_p.$$

- Linearity:

$$\langle ax + bw, z \rangle_p = \langle ax_1 + bw_1, z_1 \rangle + \langle ax_2 + bw_2, z_2 \rangle + \cdots + \langle ax_p + bw_p, z_p \rangle$$

$$= \sum_{i=1}^{p} \int_0^1 (ax_i(t) + bw_i(t)) z(t) dt$$

$$= a \sum_{i=1}^{p} \int_0^1 x_i(t) z_i(t) dt + b \sum_{i=1}^{p} \int_0^1 w_i(t) z_i(t) dt$$

$$= a \langle x, z \rangle_p + b \langle w, z \rangle_p.$$

- Positive semi-definiteness: For any non-zero $x \in \mathcal{X}$,

$$\langle x, x \rangle_p = \langle x_1, x_1 \rangle + \langle x_2, x_2 \rangle + \cdots + \langle x_p, x_p \rangle$$

$$= \sum_{i=1}^{p} \int_0^1 x_i^2(t) dt$$

$$= \sum_{i=1}^{p} \|x_i\|^2 \geq 0.$$

The norm induced by $\langle ., . \rangle_p$ on $\mathcal{X}$ is given by

$$\|x\|_p = \sqrt{\langle x, x \rangle_p}$$

$$= \left( \sum_{i=1}^{p} \int_0^1 x_i^2(t) dt \right)^{1/2}.$$

On considering $x^i, x^j \in \mathcal{X}$, $L \in \mathbb{R}^{p \times p}$, the quantity $x^{i^\top} L x^j$ can be defined as

$$x^{i^\top} L x^j = \begin{pmatrix} x_1^i & x_2^i & \ldots & x_p^i \end{pmatrix} \begin{pmatrix} L_{1,1} & L_{1,2} & \ldots & L_{1,p} \\ L_{2,1} & L_{2,2} & \ldots & L_{2,p} \\ \vdots & \vdots & \ddots & \vdots \\ L_{p,1} & L_{p,2} & \ldots & L_{p,p} \end{pmatrix} \begin{pmatrix} x_1^j \\ x_2^j \\ \vdots \\ x_p^j \end{pmatrix}$$

$$= \begin{pmatrix} x_1^i & x_2^i & \ldots & x_p^i \end{pmatrix} \begin{pmatrix} L_{1,1} x_1^j + L_{1,2} x_2^j + \cdots + L_{1,p} x_p^j \\ L_{2,1} x_1^j + L_{2,2} x_2^j + \cdots + L_{2,p} x_p^j \\ \vdots \\ L_{p,1} x_1^j + L_{p,2} x_2^j + \cdots + L_{p,1} x_p^j \end{pmatrix}$$

$$\begin{aligned} = & \langle x_1^i, L_{1,1} x_1^j + L_{1,2} x_2^j + \cdots + L_{1,p} x_p^j \rangle \\ & + \langle x_2^i, L_{2,1} x_1^j + L_{2,2} x_2^j + \cdots + L_{2,p} x_p^j \rangle \\ & + \cdots + \langle x_p^i, L_{p,1} x_1^j + L_{p,2} x_2^j + \cdots + L_{p,p} x_p^j \rangle \end{aligned} \tag{122}$$

$$= \langle x^i, L x^j \rangle_p. \tag{123}$$

Note that from equations (122 and 123), $L x^j \in \mathcal{X}$. Now based on the positive semi-definiteness of $L$, we can decompose $L = V^\top \Lambda V$, where $V$ is an orthogonal matrix and $\Lambda$ is the diagonal matrix containing the

non-negative eigenvalues.

$$
\begin{aligned}
{x^i}^\top L x^j &= \langle x^i, L x^j \rangle_p \\
&= \langle x^i, V^\top \Lambda V x^j \rangle_p \\
&= {x^i}^\top V^\top \Lambda V x^j.
\end{aligned}
\tag{124}
$$

Consider $A = \sqrt{\Lambda} V$,

$$
\begin{aligned}
{x^i}^\top L x^j &= {x^i}^\top V^\top \Lambda V x^j \\
&= {x^i}^\top A^\top A x^j \\
&= \langle A x^i, A x^j \rangle_p.
\end{aligned}
\tag{125}
$$

Note that ${x^i}^\top L x^j = \langle A x^i, A x^j \rangle_p = \langle \phi(x^i), \phi(x^j) \rangle_p$ (say) which is a characterization of kernels. For a finite $m \in \mathbb{N}$, let $\mathscr{G} \in \mathbb{R}^{m \times m}$ be the Gram (kernel) matrix induced by using $\langle Ax, Aw \rangle$ as a kernel. Let $\beta \in \mathbb{R}^m$, then we have:

$$
\begin{aligned}
\beta' \mathscr{G} \beta &= \sum_{i,j=1}^m \beta_i \mathscr{G}_{i,j} \beta_j \\
&= \sum_{i,j=1}^m \beta_i \beta_j \langle \phi(x^i), \phi(x^j) \rangle_p \\
&= \left\langle \sum_{i=1}^m \beta_i \phi(x^i), \sum_{j=1}^m \beta_j \phi(x^j) \right\rangle_p \\
&= \left\| \sum_{i=1}^m \beta_i \phi(x^i) \right\|_p^2 .
\end{aligned}
$$

This illustrates that ${x^i}^\top L x^j$ is positive semi-definite and defines a scalar-valued kernel. Now, using the properties of a scalar-valued kernel, $\gamma x^\top L x'$, with $\gamma > 0$ is a valid positive semi-definite kernel. Thus the kernel given by

$$
k_1(x, x'; G) = \gamma x^\top L x', \ \forall x, x' \in \mathcal{X}, \gamma > 0,
$$

is a valid positive semi-definite scalar kernel on $\mathcal{X} \times \mathcal{X}$. Therefore, by the construction of OVK used in (Kadri et al., 2016),

$$
(K(x, x')y)(t) = k_1(x, x'; G) \int_0^1 k_2(s, t) y(s) ds,
\tag{126}
$$

defines an operator-valued kernel on $\mathcal{X} \times \mathcal{X}$. $\qquad\square$

Now, as exponential of a scalar-valued kernel provides us another valid scalar-valued kernel (Shawe-Taylor et al., 2004), we consider the following normalized version:

$$
\begin{aligned}
\frac{\exp(\gamma x^\top L x')}{\sqrt{\exp(\gamma x^\top L x)\exp(\gamma x'^\top L x')}} &= \exp\left(\gamma x^\top L x' - \frac{\gamma}{2} x^\top L x - \frac{\gamma}{2} x'^\top L x'\right) \\
&= \exp\left[-\frac{\gamma}{2}\left(x^\top L x + x'^\top L x' - 2 x^\top L x'\right)\right] \\
&= \exp\left[-\frac{\gamma}{2}\left(\|Ax\|_p^2 + \|Ax'\|_p^2 - 2\langle Ax, Ax'\rangle_p\right)\right] \\
&= \exp\left[-\frac{\gamma}{2}\left(\|Ax - Ax'\|_p^2\right)\right] \\
&= \exp\left[-\frac{\gamma}{2}\left(\langle A(x-x'), A(x-x')\rangle_p\right)\right] \\
&= \exp\left[-\frac{\gamma}{2}\left(\langle A(x-x'), A(x-x')\rangle_p\right)\right] \\
&= \exp\left[-\frac{\gamma}{2}(x-x')^\top L (x-x')\right].
\end{aligned}
\tag{127}
$$

As exponential of a scalar-valued kernel and normalization preserves the validity of a scalar-valued kernel, we claim that the kernel in (127) defines a valid scalar-valued kernel. This illustrates that $e^{-\gamma(x-x')^\top L(x-x')}$ is a valid positive semi-definite scalar-valued kernel on $\mathcal{X} \times \mathcal{X}$ for $\gamma > 0$ and an OVK with $k_1(x, x'; G) = e^{-\gamma(x-x')^\top L(x-x')}$, $\forall x, x' \in \mathcal{X}, \gamma > 0$ in (126) provides a valid graph-induced OVK.

## A.4 Proof of Representer theorem

We provide a proof for the Representer theorem (Theorem A.2) stated in Section A.1. In the proof we use the Gateaux derivative in an associated function-valued reproducing kernel Hilbert space for an operator-valued kernel. We recall the theorem statement first and then provide a proof.

**Theorem A.4 (Representer theorem).** Let $K^G$ be an operator-valued kernel and $\mathcal{H}_{K^G}$ be its corresponding function-valued reproducing kernel Hilbert space. The solution $\widetilde{F}_\lambda \in \mathcal{H}_{K^G}$ of the regularized optimization problem:

$$
\widetilde{F}_\lambda = \underset{F \in \mathcal{H}_{K^G}}{\arg\min} \sum_{i=1}^n \|y^{(i)} - F(x^{(i)})\|_{\mathcal{Y}}^2 + \lambda \|F\|_{\mathcal{H}_{K^G}}^2,
\tag{128}
$$

where $\lambda > 0, F \in \mathcal{H}_{K^G}$, has the following form

$$
\widetilde{F}_\lambda(.) = \sum_{i=1}^n K^G(x^{(i)}, .)u_i, \text{ where } u_i \in \mathcal{Y}.
\tag{129}
$$

*Proof.* We use the Gateaux derivative to obtain the condition for stationary point of the functional $J_\lambda(F)$, given by

$$
J_\lambda(F) = \sum_{i=1}^n \|y^{(i)} - F(x^{(i)})\|_{\mathcal{Y}}^2 + \lambda \|F\|_{\mathcal{H}_K}, \ \ \forall F \in \mathcal{H}_{K^G}.
$$

In order to find the critical points in $\mathcal{H}_{K^G}$, we use Gateaux derivative $D_\mathcal{G}$ of $J_\lambda$ with respect to $F$ in the direction $H$, which is defined by

$$
D_\mathcal{G} J_\lambda(F, H) = \lim_{\tau \to 0} \frac{J_\lambda(F + \tau H) - J_\lambda(F)}{\tau}.
$$

Let $\widetilde{F}$ be the operator in $\mathcal{H}_{K^G}$ such that

$$
\widetilde{F} = \underset{F \in \mathcal{H}_{K^G}}{\arg\min} J_\lambda(F) \implies D_\mathcal{G} J_\lambda(F, H) = 0, \ \ \forall H \in \mathcal{H}_{K^G}.
$$

$J_\lambda$ can be written as

$$J_\lambda(F) = \sum_{i=1}^{n} G_i(F) + \lambda L(F),$$

and as $D_\mathcal{G} J_\lambda(F, H) = \langle D_\mathcal{G} J_\lambda(F), H \rangle_{\mathcal{H}_{K^G}}$, $\forall F, H \in \mathcal{H}_{K^G}$, we obtain the following.

noitemsep $L(F) = \|F\|^2_{\mathcal{H}_{K^G}} = \langle F, F \rangle_{\mathcal{H}_{K^G}}$. Therefore we have

$$\lim_{\tau \to 0} \frac{\langle F + \tau H, F + \tau H \rangle_{\mathcal{H}_{K^G}} - \langle F, F \rangle_{\mathcal{H}_{K^G}}}{\tau} = 2\langle F, H \rangle_{\mathcal{H}_{K^G}}$$

$$\implies D_\mathcal{G} L(F) = 2F.$$

noiitemsep $G_i(F) = \|y^{(i)} - F(x^{(i)})\|^2_\mathcal{Y}$. Then we have

$$\lim_{\tau \to 0} \frac{\|y^{(i)} - F(x^{(i)}) - \tau H(x^{(i)})\|^2_\mathcal{Y} - \|y^{(i)} - F(x^{(i)})\|^2_\mathcal{Y}}{\tau} = -2\langle y^{(i)} - F(x^{(i)}), H(x^{(i)}) \rangle_\mathcal{Y} \qquad (130)$$

$$= -2\langle K^G(x^{(i)}, .)(y^{(i)} - F(x^{(i)})), H \rangle_{\mathcal{H}_{K_G}} \qquad (131)$$

$$= -2\langle K^G(x^{(i)}, .)u_i, H \rangle_{\mathcal{H}_{K_G}}, \qquad (132)$$

$$\implies D_\mathcal{G} G_i(F) = -2K^G(x^{(i)}, .)u_i.$$

We obtain equation (131) from equation (130) using the reproducibility property. In (131), we use $u_i = y^{(i)} - F(x^{(i)})$ to get (132). Using 1, 2, and $D_\mathcal{G} J_{\lambda_1, \lambda_2}(\widetilde{F}) = 0$, we obtain, $\widetilde{F}(.) = \frac{1}{\lambda} \sum_{i=1}^{n} K^G(x^{(i)}, .)u_i$. The constant $\frac{1}{\lambda}$ can be absorbed in functions $u_i$'s, such that $\widetilde{F}(.) = \sum_{i=1}^{n} K^G(x^{(i)}, .)u_i$. $\qquad \square$

We provide a proof for the extended represeneter theorem (Theorem 5.1) based on the arguments used in the earlier proof. We recall the theorem and then provide a proof.

**Theorem A.5 (Extended representer theorem).** Let $K$ be an operator-valued kernel as defined in (26) and $\mathcal{H}_K = \mathcal{H}_K^1 \times \mathcal{H}_K^2 \times \cdots \times \mathcal{H}_K^s$ be its corresponding function-valued reproducing kernel Hilbert space based on kernels $k_1, k_2^1, \ldots, k_2^s$. The solution $\tilde{F}_\lambda \in \mathcal{H}_K$ of the regularized optimization problem.

$$\widetilde{F}_\lambda = \underset{F = [F^1, F^2, \ldots, F^s]^\top \in \mathcal{H}_K}{\arg\min} \sum_{l=1}^{s} \left( \sum_{i=1}^{n} \|y_l^{(i)} - F^l(\mathbf{x}^{(i)})\|^2_\mathcal{Y} + \lambda \|F^l\|^2_{\mathcal{H}_K^l} \right),$$

where $\lambda > 0, F = [F^1, F^2, \ldots, F^s]^\top \in \mathcal{H}_K = \mathcal{H}_K^1 \times \mathcal{H}_K^2 \times \cdots \times \mathcal{H}_K^s$, has the following form

$$\widetilde{F}_\lambda(.) = \begin{bmatrix} \widetilde{F}_\lambda^1(.) \\ \vdots \\ \widetilde{F}_\lambda^s(.) \end{bmatrix} = \begin{bmatrix} \sum_{i=1}^{n} K^1(\mathbf{x}^{(i)}, .)u_i^1 \\ \vdots \\ \sum_{i=1}^{n} K^s(\mathbf{x}^{(i)}, .)u_i^s \end{bmatrix}, \text{ where } u_i^1, u_i^2, \ldots, u_i^s \in \mathcal{Y}. \qquad (133)$$

*Proof.* We use a similar argument as in case of the representer theorem proof. The Gateaux derivative is used to obtain the condition for stationary points which minimize $J_\lambda(F)$ written as a sum of $J_\lambda^l(F^l)$, for $l \in [s]$, given by

$$J_\lambda(F) = \sum_{l=1}^{s} J_\lambda^l(F^l)$$

$$= \sum_{l=1}^{s} \left( \sum_{i=1}^{n} \|y_l^{(i)} - F^l(\mathbf{x}^{(i)})\|^2_\mathcal{Y} + \lambda \|F^k\|_{\mathcal{H}_K^l} \right), \quad \forall F \in \mathcal{H}_K.$$

In order to find the critical points in $\mathcal{H}_K = \mathcal{H}_K^1 \times \mathcal{H}_K^2 \times \cdots \times \mathcal{H}_K^s$, we use Gateaux derivative $D_{\mathcal{G}}$ of $J_\lambda^l$ with respect to $F^l$ in the direction $H \in \mathcal{H}_K^l$, which is defined by

$$D_{\mathcal{G}} J_\lambda^l(F^l, H) = \lim_{\tau \to 0} \frac{J_\lambda^l(F^l + \tau H) - J_\lambda^l(F^l)}{\tau}.$$

Let $\widetilde{F}^l$ be the operator in $\mathcal{H}_K^l$ such that

$$\widetilde{F}^l = \arg\min_{F^l \in \mathcal{H}_K^l} J_\lambda^l(F^l) \implies D_{\mathcal{G}} J_\lambda^l(F^l, H) = 0, \ \ \forall H \in \mathcal{H}_K^l, l \in [s].$$

Now, $J_\lambda^l$ can be written as

$$J_\lambda^l(F^l) = \sum_{i=1}^n G_i^l(F^l) + \lambda L^l(F^l),$$

and as $D_{\mathcal{G}} J_\lambda^l(F^l, H) = \langle D_{\mathcal{G}} J_\lambda^l(F^l), H \rangle_{\mathcal{H}_K^l}, \ \forall F, H \in \mathcal{H}_K^l, l \in [s]$, we obtain the following.

noitemsep  $L^l(F^l) = \|F^l\|_{\mathcal{H}_K^l}^2 = \langle F^l, F^l \rangle_{\mathcal{H}_K^l}$. Therefore we have

$$\lim_{\tau \to 0} \frac{\langle F^l + \tau H, F^l + \tau H \rangle_{\mathcal{H}_K^l} - \langle F^l, F^l \rangle_{\mathcal{H}_K^l}}{\tau} = 2\langle F^l, H \rangle_{\mathcal{H}_K^l}$$

$$\implies D_{\mathcal{G}} L^l(F^l) = 2F^l.$$

noiitemsep  $G_i^l(F^l) = \|y_l^{(i)} - F^l(\mathbf{x}^{(i)})\|_{\mathcal{Y}}^2$. Then we have

$$\lim_{\tau \to 0} \frac{\|y_l^{(i)} - F(\mathbf{x}^{(i)}) - \tau H(\mathbf{x}^{(i)})\|_{\mathcal{Y}}^2 - \|y_l^{(i)} - F(\mathbf{x}^{(i)})\|_{\mathcal{Y}}^2}{\tau} = -2\langle y_l^{(i)} - F(\mathbf{x}^{(i)}), H(\mathbf{x}^{(i)}) \rangle_{\mathcal{Y}} \quad (134)$$

$$= -2\langle K^l(\mathbf{x}^{(i)}, .)(y_l^{(i)} - F(\mathbf{x}^{(i)})), H \rangle_{\mathcal{H}_K^l} \quad (135)$$

$$= -2\langle K^l(\mathbf{x}^{(i)}, .)u_i^l, H \rangle_{\mathcal{H}_K^l}, \quad (136)$$

$$\implies D_{\mathcal{G}} G_i^l(F^l) = -2K^l(\mathbf{x}^{(i)}, .)u_i^l.$$

We obtain equation (135) from equation (134) using the reproducibility property. In (135), we use $u_i^l = y_l^{(i)} - F^l(\mathbf{x}^{(i)})$ to get (136). Using 1, 2, and $D_{\mathcal{G}} J_\lambda^l(\widetilde{F}^l) = 0$, we obtain, $\widetilde{F}^l(.) = \frac{1}{\lambda} \sum_{i=1}^n K^l(\mathbf{x}^{(i)}, .)u_i^l$. The constant $\frac{1}{\lambda}$ can be absorbed in functions $u_i^l$'s, such that $\widetilde{F}^l(.) = \sum_{i=1}^n K^l(\mathbf{x}^{(i)}, .)u_i^l, l \in [s]$. Therefore, we obtain the following:

$$\widetilde{F}_\lambda(.) = \begin{bmatrix} \widetilde{F}_\lambda^1(.) \\ \vdots \\ \widetilde{F}_\lambda^s(.) \end{bmatrix} = \begin{bmatrix} \sum_{i=1}^n K^1(\mathbf{x}^{(i)}, .)u_i^1 \\ \vdots \\ \sum_{i=1}^n K^s(\mathbf{x}^{(i)}, .)u_i^s \end{bmatrix}, \ \text{where } u_i^1, u_i^2, \ldots, u_i^s \in \mathcal{Y}.$$

$\square$

## A.5  Properties of $A, B, C, H, c$ and $\mathcal{M}$ matrices

This section deals with the construction and properties of matrices $A, B, C, H, c$ and $\mathcal{M}$ which have been used in the framework.

**Determining $A$:**
$A$ is a matrix which is constructed to represent the constraint $L\mathbf{1} = 0$ as $A\,\text{vech}(L) = 0$. For a given $\text{vech}(L)$,

we can obtain the condition $L\mathbf{1} = 0$ with $A\text{vech}(L) = 0$ based on the following construction:

$$A = \begin{bmatrix} e_p & 0_{p-1} & 0_{p-2} & \ldots & 0_2 & 0_1 \\ e_p^2 & e_{p-1} & 0_{p-2} & \ldots & 0_2 & 0_1 \\ e_p^3 & e_{p-1}^2 & e_{p-2} & \ldots & 0_2 & 0_1 \\ \vdots & \vdots & \vdots & \ddots & \vdots & \vdots \\ e_p^p & e_{p-1}^{p-1} & e_{p-2}^{p-2} & \ldots & e_2^2 & e_1 \end{bmatrix},$$

where $0_k = [0, 0, \ldots, 0] \in \mathbb{R}^{1 \times k}$ denotes a row vector containing $k$ zeros, $e_k = [1, 1, \ldots, 1] \in \mathbb{R}^{1 \times k}$ denotes a row vector containing $k$ ones, $e_k^i = [0, \ldots, 1, \ldots, 0] \in \mathbb{R}^{1 \times k}$ denotes a row vector with 1 in the $i$-th position and zeros elsewhere, resulting in $A \in \mathbb{R}^{p \times \frac{p(p+1)}{2}}$.

**Determining $B$:**

$B$ is a matrix which is constructed to represent the constraint $L_{i,j} \leq 0, i \neq j$ as $B\,\text{vech}(L) \leq 0$. For a given vech$(L)$, we reformulate the condition $L_{i,j} \leq 0, \forall i \neq j$ as $B\text{vech}(L) \leq 0$. This can be achieved with the following construction:

$$B = \begin{bmatrix} e_p^2 & 0_{p-1} & 0_{p-2} & \ldots & 0_3 & 0_2 & 0_1 \\ e_p^3 & 0_{p-1} & 0_{p-2} & \ldots & 0_3 & 0_2 & 0_1 \\ \vdots & \vdots & \vdots & \ddots & \vdots & \vdots & \vdots \\ e_p^p & 0_{p-1} & 0_{p-2} & \ldots & 0_3 & 0_2 & 0_1 \\ 0_p & e_{p-1}^2 & 0_{p-2} & \ldots & 0_3 & 0_2 & 0_1 \\ \vdots & \vdots & \vdots & \ddots & \vdots & \vdots & \vdots \\ 0_p & e_{p-1}^{p-1} & 0_{p-2} & \ldots & 0_3 & 0_2 & 0_1 \\ \vdots & \vdots & \vdots & \ddots & \vdots & \vdots & \vdots \\ 0_p & 0_{p-1} & 0_{p-2} & \ldots & e_3^2 & 0_2 & 0_1 \\ 0_p & 0_{p-1} & 0_{p-2} & \ldots & e_3^3 & 0_2 & 0_1 \\ 0_p & 0_{p-1} & 0_{p-2} & \ldots & 0_3 & e_2^2 & 0_1 \end{bmatrix}.$$

Note that $B \in \mathbb{R}^{\frac{p(p-1)}{2} \times \frac{p(p+1)}{2}}$.

**Determining $C$:**

In section 5.1.3, $C$ is a matrix which is used to deal with the constraint $D_{ii} \geq 0, \forall i \in [p]$. We construct a matrix $C \in \mathbb{R}^{p \times \frac{p(p+1)}{2}}$ which consists of 0's and 1's satisfying $C\text{vech}(D) = Diag(D)$. For a given vech$(L)$, we formulate the matrix $C \in \mathbb{R}^{p \times \frac{p(p+1)}{2}}$ using 0's and 1's as follows:

$$C = \begin{bmatrix} \leftarrow & C_{1:} & \rightarrow \\ \leftarrow & C_{2:} & \rightarrow \\ \leftarrow & C_{3:} & \rightarrow \\ \vdots & \vdots & \vdots \\ \leftarrow & C_{p:} & \rightarrow \end{bmatrix},$$

where the row $C_{i:} \in \mathbb{R}^{1 \times \frac{p(p+1)}{2}}$, and $C_{i:}$ contains a 1 in the $\left(\frac{i(i+1)}{2}\right)$-th position and zeros elsewhere.

**Determining $H$:**

In section 5.1.4, we require a matrix $H \in \mathbb{R}^{\frac{p(p+1)}{2} \times \frac{p(p+1)}{2}}$ comprising 0's and 1's such that $H\text{vech}(L)$ produces a vector having the same structure as vech$(L)$ with 0's corresponding to diagonal entries of $L$ and the same off-diagonal entries as in $L$. For a given vech$(L)$, we formulate the matrix $H \in \mathbb{R}^{\frac{p(p+1)}{2} \times \frac{p(p+1)}{2}}$ using 0's and

1's as follows:

$$H = \begin{bmatrix} \leftarrow & H_{1:} & \rightarrow \\ \leftarrow & H_{2:} & \rightarrow \\ \leftarrow & H_{3:} & \rightarrow \\ \vdots & \vdots & \vdots \\ \leftarrow & H_{\frac{p(p+1)}{2}:} & \rightarrow \end{bmatrix},$$

where the rows $H_{i:} \in \mathbb{R}^{1 \times \frac{p(p+1)}{2}}$, and $H_{i:}$ contains a 0 in the $\left(\frac{i(i+1)}{2}\right)$-th position and 1's elsewhere.

**Determining $c$:**
In section 5.1.4, we consider a vector $c \in \mathbb{R}^{\frac{p(p+1)}{2}}$ consisting of 0's and 1's, such that $c^\top \mathrm{vech}(L) = \mathrm{trace}(L)$. The vector $c \in \mathbb{R}^{\frac{p(p+1)}{2}}$ is defined for obtaining the diagonal entries of $D$ from $\mathrm{vech}(D)$ using 0's and 1's such that

$$c^\top \mathrm{vech}(D) = d_{1,1} + d_{2,2} + \cdots + d_{p,p},$$

where $c$ is given as follows:

$$c = [c_1, c_2, \ldots, c_{p-1}, c_p]^\top,$$

where $c \in \mathbb{R}^{\frac{p(p+1)}{2}}$ such that $c_i \in \mathbb{R}^{1 \times \frac{i(i+1)}{2}}$ has all 0's with 1 in $\left(\frac{i(i+1)}{2}\right)$-th element.

**Determining $\mathcal{M}$:**
$\mathcal{M}$ is a matrix which is constructed to transform $\mathrm{vech}(L)$ to $\mathrm{vec}(L)$ using $\mathcal{M}\mathrm{vech}(L) = \mathrm{vec}(L)$. Let the Laplacian of the graph of $p$ nodes be denoted by $L$ which is symmetric,

$$L = \begin{bmatrix} l_{1,1} & l_{2,1} & l_{3,1} & \cdots & l_{p,1} \\ l_{2,1} & l_{2,2} & l_{3,2} & \cdots & l_{p,2} \\ l_{3,1} & l_{3,2} & l_{3,3} & \cdots & l_{p,3} \\ \vdots & \vdots & \vdots & \ddots & \vdots \\ l_{p,1} & l_{p,2} & l_{p,3} & \cdots & l_{p,p} \end{bmatrix}.$$

Now, $\mathrm{vec}(L) = [l_{1,1}, \ldots, l_{p,1}, l_{2,1}, \ldots, l_{p,2}, \ldots, l_{p,1}, \ldots, l_{p,p}]^\top$ is obtained by stacking the columns and

$$\mathrm{vech}(L) = [l_{1,1}, \ldots, l_{p,1}, l_{2,2}, \ldots, l_{p,2}, l_{3,3}, \ldots, l_{p,3}, \ldots, l_{p,p}]^\top,$$

which is obtained by eliminating the super-diagonal elements and then stacking them up. As illustrated, $\mathrm{vec}(L) \in \mathbb{R}^{p^2}$ and $\mathrm{vech}(L) \in \mathbb{R}^{p(p+1)/2}$. We can find a matrix $\mathcal{M}$ which satisfies

$$\mathrm{vec}(L) = \mathcal{M}\mathrm{vech}(L).$$

Hence observe that $\mathcal{M} \in \mathbb{R}^{p^2 \times \frac{p(p+1)}{2}}$. Therefore,

$$\mathcal{M}^\top = \sum_{i \geq j} v_{i,j}(\mathrm{vec}(T_{i,j}))^\top,$$

where $v_{i,j}$ is a vector of order $\frac{1}{2}p(p+1)$ having the value 1 in the $(j-1)n + i - \frac{1}{2}j(j-1)$-th position and 0 elsewhere and $T_{i,j}$ is a $p \times p$ matrix with 1 in positions $(i,j)$ and $(j,i)$, and 0 elsewhere.

The following relations are used in order to write the expression of objective function in terms of $\mathrm{vec}(L)$.

$$(x^{(i)} - x^{(j)})^\top (L + D)(x^{(i)} - x^{(j)}) = \mathrm{vec}((x^{(i)} - x^{(j)})(x^{(i)} - x^{(j)})^\top)^\top \mathrm{vec}(L + D),$$
$$= \mathrm{vec}((x^{(i)} - x^{(j)})(x^{(i)} - x^{(j)})^\top)^\top (\mathrm{vec}(L) + \mathrm{vec}(D)),$$
$${x^{(i)}}^\top L x^{(i)} = \mathrm{vec}(x^{(i)}{x^{(i)}}^\top)^\top \mathrm{vec}(L),$$
$$\|D\|_F = \mathrm{vec}(D)^\top \mathrm{vec}(D).$$

## A.6 Derivation of Gradients

In this section, we cover the derivation of $\nabla_{\text{vech}(L)} J$ and $\nabla_{\text{vech}(D)} J$ which are required in Algorithm 2. We use (103) and (108) in the following expression to find the gradients. Recall $\bar{k}_2(u)(t) = \int_0^1 e^{-\gamma_{op}|s-t|} u(s) ds, \gamma_{op} > 0, s, t \in \mathbb{R}$ in the following derivations.

$$
\begin{aligned}
J(\mathbf{u}, L, D) &= \sum_{i=1}^n \left\| y^{(i)} - \sum_{j=1}^n K(x^{(i)}, x^{(j)}) u_j \right\|_{\mathcal{Y}}^2 + \lambda \sum_{i,j=1}^n \langle K(x^{(i)}, x^{(j)}) u_i, u_j \rangle_{\mathcal{Y}} \\
&\quad + \rho_L \sum_{i=1}^n x^{(i)^\top} L x^{(i)} + \rho_D \|D\|_F^2 \\
&= \sum_{i=1}^n \left\| y^{(i)} - \sum_{j=1}^n k_1(x^{(i)}, x^{(j)}; G) \bar{k}_2(u_j) \right\|_{\mathcal{Y}}^2 + \lambda \sum_{i=1,j=1}^n \langle k_1(x^{(i)}, x^{(j)}; G) \bar{k}_2(u_i), u_j \rangle_{\mathcal{Y}} \\
&\quad + \rho_L \sum_{i=1}^n x^{(i)^\top} L x^{(i)} + \rho_D \|D\|_F^2 \\
&= \sum_{i=1}^n \left\| y^{(i)} - \sum_{j=1}^n e^{-\gamma(x^{(i)} - x^{(j)})^\top (L+D)(x^{(i)} - x^{(j)})} \bar{k}_2(u_j) \right\|_{\mathcal{Y}}^2 \\
&\quad + \lambda \sum_{i,j=1}^n \langle e^{-\gamma(x^{(i)} - x^{(j)})^\top (L+D)(x^{(i)} - x^{(j)})} \bar{k}_2(u_i), u_j \rangle_{\mathcal{Y}} + \rho_L \sum_{i=1}^n x^{(i)^\top} L x^{(i)} + \rho_D \|D\|_F^2.
\end{aligned}
$$

We consider the following variables for simplifying the computations:

$$
\begin{aligned}
R_{ij} &= \text{vec}((x^{(i)} - x^{(j)})(x^{(i)} - x^{(j)})^\top)^\top \mathcal{M}, \\
\bar{R}_i &= \text{vec}(x^{(i)} x^{(i)^\top})^\top \mathcal{M}.
\end{aligned} \tag{137}
$$

Therefore,

$$
\begin{aligned}
J &= \sum_{i=1}^n \left\| y^{(i)} - \sum_{j=1}^n e^{-\gamma R_{ij}(\text{vech}(L) + \text{vech}(D))} \bar{k}_2(u_j) \right\|_{\mathcal{Y}}^2 \\
&\quad + \lambda \sum_{i,j=1}^n \langle e^{-\gamma R_{ij}(\text{vech}(L) + \text{vech}(D))} \bar{k}_2(u_i), u_j \rangle_{\mathcal{Y}} + \rho_L \sum_{i=1}^n \bar{R}_i \text{vech}(L) + \rho_D \text{vech}(D)^\top \mathcal{M}^\top \mathcal{M} \text{vech}(D) \\
&= \sum_{i=1}^n \left\langle y^{(i)} - \sum_{j=1}^n e^{-\gamma R_{ij}(\text{vech}(L) + \text{vech}(D))} \bar{k}_2(u_j), y^{(i)} - \sum_{j=1}^n e^{-\gamma R_{ij}(\text{vech}(L) + \text{vech}(D))} \bar{k}_2(u_j) \right\rangle_{\mathcal{Y}} \\
&\quad + \lambda \sum_{i,j=1}^n e^{-\gamma R_{ij}(\text{vech}(L) + \text{vech}(D))} \langle \bar{k}_2(u_i), u_j \rangle_{\mathcal{Y}} + \rho_L \sum_{i=1}^n \bar{R}_i \text{vech}(L) + \rho_D \text{vech}(D)^\top \mathcal{M}^\top \mathcal{M} \text{vech}(D) \\
&= \sum_{i=1}^n \left[ \langle y^{(i)}, y^{(i)} \rangle_{\mathcal{Y}} - 2 \sum_{j=1}^n e^{-\gamma R_{ij}(\text{vech}(L) + \text{vech}(D))} \langle y^{(i)}, \bar{k}_2(u_j) \rangle_{\mathcal{Y}} \right. \\
&\quad \left. + \sum_{j,k=1}^n e^{-\gamma(R_{ij} + R_{ik})(\text{vech}(L) + \text{vech}(D))} \langle \bar{k}_2(u_j), \bar{k}_2(u_k) \rangle_{\mathcal{Y}} \right] + \lambda \sum_{i,j=1}^n e^{-\gamma R_{ij}(\text{vech}(L) + \text{vech}(D))} \langle \bar{k}_2(u_i), u_j \rangle_{\mathcal{Y}} \\
&\quad + \rho_L \sum_{i=1}^n \bar{R}_i \text{vech}(L) + \rho_D \text{vech}(D)^\top \mathcal{M}^\top \mathcal{M} \text{vech}(D).
\end{aligned}
$$

The expression above for $J$ can be used to determine the gradient with respect to $\text{vech}(L)$ and $\text{vech}(D)$. The gradient of $J$ with respect to $\text{vech}(L)$ for a fixed $\mathbf{u}$ and $D$ is given as follows:

$$
\begin{aligned}
\nabla_{\text{vech}(L)} J = {}& 2\gamma \sum_{i,j=1}^{n} R_{ij}^{\top} e^{-\gamma R_{ij}(\text{vech}(L)+\text{vech}(D))} \left\langle y^{(i)}, \bar{k}_2(u_j) \right\rangle_{\mathcal{Y}} \\
& - \gamma \sum_{i,j,k=1}^{n} (R_{ij}+R_{ik})^{\top} e^{-\gamma(R_{ij}+R_{ik})(\text{vech}(L)+\text{vech}(D))} \left\langle \bar{k}_2(u_j), \bar{k}_2(u_k) \right\rangle_{\mathcal{Y}} \\
& - \lambda\gamma \sum_{i,j=1}^{n} R_{ij}^{\top} e^{-\gamma R_{ij}(\text{vech}(L)+\text{vech}(D))} \langle \bar{k}_2(u_i), u_j \rangle_{\mathcal{Y}} + \rho_L \sum_{i=1}^{n} \bar{R}_i^{\top}.
\end{aligned}
$$

Similarly, for a fixed $\mathbf{u}$ and $\text{vech}(L)$ the gradient of $J$ with respect to $\text{vech}(D)$ can be written as

$$
\begin{aligned}
\nabla_{\text{vech}(D)} J = {}& 2\gamma \sum_{i,j=1}^{n} R_{ij}^{\top} e^{-\gamma R_{ij}(\text{vech}(L)+\text{vech}(D))} \left\langle y^{(i)}, \bar{k}_2(u_j) \right\rangle_{\mathcal{Y}} \\
& - \gamma \sum_{i,j,k=1}^{n} (R_{ij}+R_{ik})^{\top} e^{-\gamma(R_{ij}+R_{ik})(\text{vech}(L)+\text{vech}(D))} \left\langle \bar{k}_2(u_j), \bar{k}_2(u_k) \right\rangle_{\mathcal{Y}} \\
& - \lambda\gamma \sum_{i,j=1}^{n} R_{ij}^{\top} e^{-\gamma R_{ij}(\text{vech}(L)+\text{vech}(D))} \langle \bar{k}_2(u_i), u_j \rangle_{\mathcal{Y}} + 2\rho_D \mathcal{M}^{\top} \mathcal{M} \text{vech}(D).
\end{aligned}
$$

For the experiments on NBA data, we use the approach in Section 5.1.5 with $r = 2$ and $s = 2$. The inputs $\mathbf{x}^{(i)} = (x_1^{(i)}, x_2^{(i)}) \in \mathcal{X}^2$ and outputs $\mathbf{y}^{(i)} = (y_1^{(i)}, y_2^{(i)}) \in \mathcal{Y}^2$ with (25) and (26), for $i \in [n]$. We consider the following variables for simplifying the computations:

$$
\begin{aligned}
R_{ij}^1 &= \text{vec}((x_1^{(i)} - x_1^{(j)})(x_1^{(i)} - x_1^{(j)})^{\top})^{\top} \mathcal{M}, \\
R_{ij}^2 &= \text{vec}((x_2^{(i)} - x_2^{(j)})(x_2^{(i)} - x_2^{(j)})^{\top})^{\top} \mathcal{M}, \\
\bar{R}_i^1 &= \text{vec}(x_1^{(i)} x_1^{(i)\top})^{\top} \mathcal{M}, \\
\bar{R}_i^2 &= \text{vec}(x_2^{(i)} x_2^{(i)\top})^{\top} \mathcal{M}.
\end{aligned}
$$

$$J(\mathbf{u}^1,\mathbf{u}^2,L,D) = \sum_{i=1}^n \left\| y_1^{(i)} - \sum_{j=1}^n K(\mathbf{x}^{(i)},\mathbf{x}^{(j)})u_j^1 \right\|_{\mathcal{Y}}^2 + \sum_{i=1}^n \left\| y_2^{(i)} - \sum_{j=1}^n K(\mathbf{x}^{(i)},\mathbf{x}^{(j)})u_j^2 \right\|_{\mathcal{Y}}^2$$

$$+ \lambda \sum_{i,j=1}^n \langle K(\mathbf{x}^{(i)},\mathbf{x}^{(j)})u_i^1, u_j^1 \rangle_{\mathcal{Y}} + \lambda \sum_{i,j=1}^n \langle K(\mathbf{x}^{(i)},\mathbf{x}^{(j)})u_i^2, u_j^2 \rangle_{\mathcal{Y}}$$

$$+ \rho_1 \sum_{i=1}^n x_1^{(i)^\top} L x_1^{(i)} + \rho_2 \sum_{i=1}^n x_2^{(i)^\top} L x_2^{(i)} + \rho_D \|D\|_F^2$$

$$= \sum_{i=1}^n \left\| y_1^{(i)} - \sum_{j=1}^n k_1(\mathbf{x}^{(i)},\mathbf{x}^{(j)};G)\bar{k}_2^1(u_j^1) \right\|_{\mathcal{Y}}^2 + \sum_{i=1}^n \left\| y_2^{(i)} - \sum_{j=1}^n k_1(\mathbf{x}^{(i)},\mathbf{x}^{(j)};G)\bar{k}_2^2(u_j^2) \right\|_{\mathcal{Y}}^2$$

$$+ \lambda \sum_{i=1,j=1}^n \langle k_1(\mathbf{x}^{(i)},\mathbf{x}^{(j)};G)\bar{k}_2^1(u_i^1), u_j^1 \rangle_{\mathcal{Y}} + \lambda \sum_{i=1,j=1}^n \langle k_1(\mathbf{x}^{(i)},\mathbf{x}^{(j)};G)\bar{k}_2^2(u_i^2), u_j^2 \rangle_{\mathcal{Y}}$$

$$+ \rho_1 \sum_{i=1}^n x_1^{(i)^\top} L x_1^{(i)} + \rho_2 \sum_{i=1}^n x_2^{(i)^\top} L x_2^{(i)} + \rho_D \|D\|_F^2$$

$$= \sum_{i=1}^n \left\| y_1^{(i)} - \sum_{j=1}^n e^{-\gamma_1 \left(x_1^{(i)}-x_1^{(j)}\right)^\top (L+D)\left(x_1^{(i)}-x_1^{(j)}\right) - \gamma_2 \left(x_2^{(i)}-x_2^{(j)}\right)^\top (L+D)\left(x_2^{(i)}-x_2^{(j)}\right)} \bar{k}_2^1(u_j^1) \right\|_{\mathcal{Y}}^2$$

$$+ \sum_{i=1}^n \left\| y_2^{(i)} - \sum_{j=1}^n e^{-\gamma_1 \left(x_1^{(i)}-x_1^{(j)}\right)^\top (L+D)\left(x_1^{(i)}-x_2^{(j)}\right) - \gamma_2 \left(x_2^{(i)}-x_2^{(j)}\right)^\top (L+D)\left(x_2^{(i)}-x_2^{(j)}\right)} \bar{k}_2^2(u_j^2) \right\|_{\mathcal{Y}}^2$$

$$+ \lambda \sum_{i,j=1}^n \langle e^{-\gamma_1 \left(x_1^{(i)}-x_1^{(j)}\right)^\top (L+D)\left(x_1^{(i)}-x_1^{(j)}\right) - \gamma_2 \left(x_2^{(i)}-x_2^{(j)}\right)^\top (L+D)\left(x_2^{(i)}-x_2^{(j)}\right)} \bar{k}_2^1(u_j^1), u_j^1 \rangle_{\mathcal{Y}}$$

$$+ \lambda \sum_{i,j=1}^n \langle e^{-\gamma_1 \left(x_1^{(i)}-x_1^{(j)}\right)^\top (L+D)\left(x_1^{(i)}-x_1^{(j)}\right) - \gamma_2 \left(x_2^{(i)}-x_2^{(j)}\right)^\top (L+D)\left(x_2^{(i)}-x_2^{(j)}\right)} \bar{k}_2^2(u_j^2), u_j^2 \rangle_{\mathcal{Y}}$$

$$+ \rho_1 \sum_{i=1}^n x_1^{(i)^\top} L x_1^{(i)} + \rho_2 \sum_{i=1}^n x_2^{(i)^\top} L x_2^{(i)} + \rho_D \|D\|_F^2.$$

The gradient of $J$ with respect to $\mathrm{vech}(L)$ for fixed $\mathbf{u}^1,\mathbf{u}^2$ and $D$ is given by

$$\nabla_{\mathrm{vech}(L)} J = 2 \sum_{i,j=1}^n (\gamma_1 R_{ij}^1 + \gamma_2 R_{ij}^2)^\top e^{-(\gamma_1 R_{ij}^1 + \gamma_2 R_{ij}^2)(\mathrm{vech}(L)+\mathrm{vech}(D))} \left( \left\langle y_1^{(i)}, \bar{k}_2^1(u_j^1) \right\rangle_{\mathcal{Y}} + \left\langle y_2^{(i)}, \bar{k}_2^2(u_j^2) \right\rangle_{\mathcal{Y}} \right)$$

$$- \sum_{i,j,k=1}^n (\gamma_1 R_{ij}^1 + \gamma_2 R_{ij}^2 + \gamma_1 R_{ik}^1 + \gamma_2 R_{ik}^2)^\top e^{-(\gamma_1 R_{ij}^1 + \gamma_2 R_{ij}^2 + \gamma_1 R_{ik}^1 + \gamma_2 R_{ik}^2)(\mathrm{vech}(L)+\mathrm{vech}(D))} \langle \bar{k}_2^1(u_j^1), \bar{k}_2^1(u_k^1) \rangle_{\mathcal{Y}}$$

$$- \sum_{i,j,k=1}^n (\gamma_1 R_{ij}^1 + \gamma_2 R_{ij}^2 + \gamma_1 R_{ik}^1 + \gamma_2 R_{ik}^2)^\top e^{-(\gamma_1 R_{ij}^1 + \gamma_2 R_{ij}^2 + \gamma_1 R_{ik}^1 + \gamma_2 R_{ik}^2)(\mathrm{vech}(L)+\mathrm{vech}(D))} \langle \bar{k}_2^2(u_j^2), \bar{k}_2^2(u_k^2) \rangle_{\mathcal{Y}}$$

$$- \lambda \sum_{i,j=1}^n (\gamma_1 R_{ij}^1 + \gamma_2 R_{ij}^2)^\top e^{-(\gamma_1 R_{ij}^1 + \gamma_2 R_{ij}^2)(\mathrm{vech}(L)+\mathrm{vech}(D))} \langle \bar{k}_2^1(u_i^1), u_j^1 \rangle_{\mathcal{Y}}$$

$$- \lambda \sum_{i,j=1}^n (\gamma_1 R_{ij}^1 + \gamma_2 R_{ij}^2)^\top e^{-(\gamma_1 R_{ij}^1 + \gamma_2 R_{ij}^2)(\mathrm{vech}(L)+\mathrm{vech}(D))} \langle \bar{k}_2^2(u_i^2), u_j^2 \rangle_{\mathcal{Y}} + \rho_1 \sum_{i=1}^n \bar{R}_i^{1^\top} + \rho_2 \sum_{i=1}^n \bar{R}_i^{2^\top}.$$

Similarly, for fixed $\mathbf{u}^1, \mathbf{u}^2$ and $\text{vech}(L)$ the gradient of $J$ with respect to $\text{vech}(D)$ can be written as

$$
\nabla_{\text{vech}(D)} J = 2 \sum_{i,j=1}^{n} (\gamma_1 R_{ij}^1 + \gamma_2 R_{ij}^2)^\top e^{-(\gamma_1 R_{ij}^1 + \gamma_2 R_{ij}^2)(\text{vech}(L) + \text{vech}(D))} \left( \left\langle y_1^{(i)}, \bar{k}_2^1(u_j^1) \right\rangle_{\mathcal{Y}} + \left\langle y_2^{(i)}, \bar{k}_2^2(u_j^2) \right\rangle_{\mathcal{Y}} \right)
$$

$$
- \sum_{i,j,k=1}^{n} (\gamma_1 R_{ij}^1 + \gamma_2 R_{ij}^2 + \gamma_1 R_{ik}^1 + \gamma_2 R_{ik}^2)^\top e^{-(\gamma_1 R_{ij}^1 + \gamma_2 R_{ij}^2 + \gamma_1 R_{ik}^1 + \gamma_2 R_{ik}^2)(\text{vech}(L) + \text{vech}(D))} \left\langle \bar{k}_2^1(u_j^1), \bar{k}_2^1(u_k^1) \right\rangle_{\mathcal{Y}}
$$

$$
- \sum_{i,j,k=1}^{n} (\gamma_1 R_{ij}^1 + \gamma_2 R_{ij}^2 + \gamma_1 R_{ik}^1 + \gamma_2 R_{ik}^2)^\top e^{-(\gamma_1 R_{ij}^1 + \gamma_2 R_{ij}^2 + \gamma_1 R_{ik}^1 + \gamma_2 R_{ik}^2)(\text{vech}(L) + \text{vech}(D))} \left\langle \bar{k}_2^2(u_j^2), \bar{k}_2^2(u_k^2) \right\rangle_{\mathcal{Y}}
$$

$$
- \lambda \sum_{i,j=1}^{n} (\gamma_1 R_{ij}^1 + \gamma_2 R_{ij}^2)^\top e^{-(\gamma_1 R_{ij}^1 + \gamma_2 R_{ij}^2)(\text{vech}(L) + \text{vech}(D))} \langle \bar{k}_2^1(u_i^1), u_j^1 \rangle_{\mathcal{Y}}
$$

$$
- \lambda \sum_{i,j=1}^{n} (\gamma_1 R_{ij}^1 + \gamma_2 R_{ij}^2)^\top e^{-(\gamma_1 R_{ij}^1 + \gamma_2 R_{ij}^2)(\text{vech}(L) + \text{vech}(D))} \langle \bar{k}_2^2(u_i^2), u_j^2 \rangle_{\mathcal{Y}} + 2\rho_D \mathcal{M}^\top \mathcal{M} \text{vech}(D).
$$

### A.7 OpMINRES Algorithm

#### A.7.1 OpLanczos Step

OpLanczos in OpMINRES is used to trigiagonalize the operator matrix $\mathbf{P}$. The vectors obtained from OpLanczos form an orthonormal set. Using the OpLanczosStep Algorithm 5, we can obtain,

$$
\mathbf{P} Q_k = Q_k T_k, \quad \text{where } T_k = \begin{bmatrix} \alpha_1 & \beta_2 & & & & 0 \\ \beta_2 & \alpha_2 & \beta_3 & & & \\ & \beta_3 & \alpha_3 & \ddots & & \\ & & \ddots & \ddots & \beta_{k-2} & \\ & & & \beta_{k-1} & \alpha_{k-1} & \beta_k \\ 0 & & & & \beta_k & \alpha_k \end{bmatrix},
$$

and $Q_k = [q_1, q_2, \ldots, q_k]$, where $q_i$'s are obtained using OpLanczosStep Algorithm. The columns of $Q_k$ belonging to $\mathcal{Y}^n$ are orthonormal and the following equation is satisfied:

$$
\mathbf{P} Q_k = Q_{k+1} \overline{T}_k, \quad \text{where } \overline{T}_k = \begin{bmatrix} \alpha_1 & \beta_2 & & & & 0 \\ \beta_2 & \alpha_2 & \beta_3 & & & \\ & \beta_3 & \alpha_3 & \ddots & & \\ & & \ddots & \ddots & \beta_{k-2} & \\ & & & \beta_{k-1} & \alpha_{k-1} & \beta_k \\ & & & & \beta_k & \alpha_k \\ 0 & & & & & \beta_{k+1} \end{bmatrix}.
$$

We intend to solve $\mathbf{A}\mathbf{u} = \mathbf{y}$ by obtaining a solution in the Krylov space $\mathcal{K}_k(\mathbf{P}, \mathbf{y}) = \text{span}\{\mathbf{y}, \mathbf{P}\mathbf{y}, \mathbf{P}^2\mathbf{y}, \ldots, \mathbf{P}^{k-1}\mathbf{y}\}$. For each iteration $k$, we obtain the following equations using the transformation $\boldsymbol{\theta} = Q_k \vartheta$, where $\boldsymbol{\theta} \in \mathcal{Y}^n, \vartheta \in \mathbb{R}^k$.

$$
\min_{\boldsymbol{\theta} \in \mathcal{K}_k(\mathbf{P}, \mathbf{y})} \|\mathbf{y} - \mathbf{P}\boldsymbol{\theta}\|_{\mathcal{Y}^n} = \min_{\vartheta \in \mathbb{R}^k} \|\mathbf{y} - \mathbf{P}Q_k \vartheta\|_{\mathcal{Y}^n} = \min_{\vartheta \in \mathbb{R}^k} \|\mathbf{y} - Q_{k+1} \overline{T}_k \vartheta\|_{\mathcal{Y}^n}
$$

$$
= \min_{\vartheta \in \mathbb{R}^k} \|Q_{k+1}(\beta_1 e_1 - \overline{T}_k \vartheta)\|_{\mathcal{Y}^n}, \tag{138}
$$

$$
(\text{where } \beta_1 = \|\mathbf{y}\|_{\mathcal{Y}^n}, e_1 = [1\,0\,\ldots\,0]^\top \text{ and } q_1 = \mathbf{y})
$$

$$
= \min_{\vartheta \in \mathbb{R}^k} \|\beta_1 e_1 - \overline{T}_k \vartheta\|_2. \tag{139}
$$

---

**Algorithm 4 OpMINRES**$(P, b, maxiter)$

---

   **Input:** $P, b, maxiter$
   **Output:** $\vartheta, \phi, \psi, \chi$
   $\beta_1 = \|b\|_{\mathcal{Y}^n}$
   $q_0 = 0$
   $q_1 = \frac{1}{\beta_1} b$
   $\phi_0 = \tau_0 = \beta_1$
   $\chi_0 = 0$
   $\delta_1^{(1)} = 0$
   $c_0 = -1$
   $s_0 = 0$
   $d_0 = d_{-1} = \vartheta_0 = 0$
   $k = 1$
   **while** stopping criteria not satisfied **do**
      **OpLanczosStep**$(P, q_k, q_{k-1}, \beta_k) \to \alpha_k, \beta_{k+1}, q_{k+1}$
      //last left orthogonalization on middle two entries in last column of $T_{k+1,k}$
      $\delta_k^{(2)} = c_{k-1} \delta_k^{(1)} + s_{k-1} \alpha_k$
      $\gamma_k^{(1)} = s_{k-1} \delta_k^{(1)} - c_{k-1} \alpha_k$
      //last left orthogonalization to produce first two entries of $T_{k+2,k+1} e_{k+1}$
      $\epsilon_{k+1}^{(1)} = s_{k-1} \beta_{k+1}$
      $\delta_{k+1}^{(1)} = -c_{k-1} \beta_{k+1}$
      //current left orthogonalization to zero out $\beta_{k+1}$
      **SymOrtho**$(\gamma_k^{(1)}, \beta_{k+1}) \to c_k, s_k, \gamma_k^{(2)}$
      //right-hand side, residual norms
      $\tau_k = c_k \phi_{k-1}$
      $\phi_k = s_k \phi_{k-1}$
      $\psi_{k-1} = \phi_{k-1} \sqrt{(\gamma_k^{(1)})^2 + (\delta_{k+1}^{(1)})^2}$
      //update solution
      $d_k = \frac{1}{\gamma_k^{(2)}} \left( v_k - \delta_k^{(2)} d_{k-1} - \epsilon_k^{(1)} d_{k-2} \right)$
      $\vartheta_k = \vartheta_{k-1} + \tau_k d_k$
      $\chi_k = \|\vartheta_k\|$
      $k \leftarrow k + 1$
   **end while**
   $\vartheta = \vartheta_k, \phi = \phi_k, \psi = \phi_k \sqrt{(\gamma_{k+1}^{(1)})^2 + (\delta_{k+2}^{(1)})^2}, \chi = \chi_k$

---

The change in norms $\|.\|_{\mathcal{Y}^n}$ in (138) to $\|.\|_2$ is obtained based on the following arguments. Let $z = [z_1, z_2, \ldots, z_{k+1}]^\top \in \mathbb{R}^{k+1}$ and $Q_{k+1} = [q_1, q_2, \ldots, q_{k+1}]$, where $q_i \in \mathcal{Y}^n$, for $i = 1, 2, \ldots, k+1$, then we have

$$\|Q_{k+1} z_{k+1}\|_{\mathcal{Y}^n} = \|z_1 q_1 + z_2 q_2 + \cdots + z_{k+1} q_{k+1}\|_{\mathcal{Y}^n}$$

$$= \sqrt{z_1^2 \int_{\Omega_y} q_1^2(t) dt + z_2^2 \int_{\Omega_y} q_2^2(t) dt + \cdots + z_{k+1}^2 \int_{\Omega_y} q_{k+1}^2(t) dt} \tag{140}$$

$$= \sqrt{z_1^2 + z_2^2 + \cdots + z_{k+1}^2} \tag{141}$$

$$= \|z\|_2.$$

Equation (140) reduces to (141) as the $q_i$'s are orthonormal in $\mathcal{Y}^n$. Solving for $\vartheta_k = \arg\min_{\vartheta \in \mathbb{R}^k} \|\beta_1 e_1 - \bar{T}_k \vartheta\|_2$ can be done using QR decomposition (Choi, 2006) which has been discussed in the next section. Now, the

---

**Algorithm 5 OpLanczosStep**$(P, q_k, q_{k-1}, \beta_k)$

---

**Input:** $A, q_k, q_{k-1}, \beta_k$
**Output:** $\alpha_k, \beta_{k+1}, q_{k+1}$
$\bar{q}_{k+1} = Aq_k - \beta_k q_{k-1}$
$\alpha_k = \langle \bar{q}_{k+1}, q_k \rangle_{\mathcal{Y}^n}$
$\bar{q}_{k+1} \leftarrow \bar{q}_{k+1} - \alpha_k q_k$
$\beta_{k+1} = \|\bar{q}_{k+1}\|_{\mathcal{Y}^n}$
$q_{k+1} = \frac{1}{\beta_{k+1}} \bar{q}_{k+1}$

---

**Algorithm 6 SymOrtho**$(a, b)$

---

**Input:** $a, b$
**Output:** $c, s, r$
**if** $b == 0$ **then**
    $s = 0$
    $r = |a|$
    **if** $a == 0$ **then**
        $c = 1$
    **else**
        $c = \text{sign}(a)$
    **end**
**else if** $a == 0$ **then**
    $c = 0$
    $s = \text{sign}(b)$
    $r = |b|$
**else if** $|b| > |a|$ **then**
    $\tau = a/b$
    $s = \text{sign}(b)/\sqrt{1 + \tau^2}$
    $c = s\tau$
    $r = b/s$
**else if** $|a| > |b|$ **then**
    $\tau = b/a$
    $c = \text{sign}(a)/\sqrt{1 + \tau^2}$
    $s = c\tau$
    $r = a/c$
**end**

---

transformation from $\mathbb{R}^k$ back to $\mathcal{Y}^n$ to obtain $\mathbf{u}^k$ is achieved using by the following:

$$\mathbf{u}^k = Q_k \vartheta_k = Q_k \left( \arg \min_{\vartheta \in \mathbb{R}^k} \|\beta_1 e_1 - \overline{T}_k \vartheta\|_2 \right).$$

### A.7.2 QR Decomposition

In order to apply QR decomposition on symmetric $\overline{T}_k$, we use Givens rotation $S_k$ to obtain a upper-triangular system.

$$S_k \overline{T}_k = \begin{bmatrix} R_k \\ 0 \end{bmatrix} = \begin{bmatrix} \gamma_1^{(1)} & \delta_2^{(1)} & \epsilon_3^{(1)} & & & & 0 \\ & \gamma_2^{(2)} & \delta_3^{(2)} & \epsilon_4^{(1)} \\ & & \ddots & \ddots & \ddots \\ & & & \gamma_{k-2}^{(2)} & \delta_{k-1}^{(2)} & \epsilon_k^{(1)} \\ & & & & \gamma_{k-1}^{(2)} & \delta_k^{(2)} \\ & & & & & \gamma_k^{(2)} \\ 0 & & & & & 0 \end{bmatrix}, \qquad S_k(\beta_1 e_1) = \begin{bmatrix} t_k \\ \phi_k \end{bmatrix},$$

where $S_k = S_{k,k+1}\ldots S_{2,3}S_{1,2}$ and $S_{i,i+1}$ are Givens rotations created to annihilate the $\beta_i$'s in sub-diagonal of $\overline{T}_k$. The $S_{i,i+1}$'s involved in the product to obtain $S_k$ are given by,

$$S_{i,i+1} = \begin{bmatrix} I_{i-1} \\ & c_i & s_i \\ & s_i & -c_i \\ & & & I_{k-i} \end{bmatrix}.$$

The matrices $S_{i,i+1}$ are obtained using the SymOrtho Algorithm 6. The sub-problem can be rewritten with $\vartheta_k = \arg\min_{\vartheta \in \mathbb{R}^k} \|\beta_1 e_1 - \overline{T}_k \vartheta\|_2$ as

$$\vartheta_k = \arg\min_{\vartheta \in \mathbb{R}^k} \left\| \begin{bmatrix} t_k \\ \phi_k \end{bmatrix} - \begin{bmatrix} R_k \\ 0 \end{bmatrix} \vartheta \right\|_2, \text{ where } t_k = [\tau_1, \tau_2, \ldots, \tau_k]^\top \text{ and}$$

$$\begin{bmatrix} t_k \\ \phi_k \end{bmatrix} = \beta_1 S_{k,k+1}\ldots S_{2,3} \begin{bmatrix} c_1 \\ s_1 \\ 0_{k-1} \end{bmatrix} = \beta_1 S_{k,k+1}\ldots S_{3,4} \begin{bmatrix} c_1 \\ s_1 c_2 \\ s_1 s_2 \\ 0_{k-2} \end{bmatrix} = \beta_1 \begin{bmatrix} c_1 \\ s_1 c_2 \\ \vdots \\ s_1 \ldots s_{k-1} c_k \\ s_1 \ldots s_{k-1} s_k \end{bmatrix}.$$

A shorthand way to represent the action of $S_{k,k+1}$ can be described as

$$\begin{bmatrix} c_k & s_k \\ s_k & -c_k \end{bmatrix} \left[ \begin{array}{ccc|c} \gamma_k^{(1)} & \delta_{k+1}^{(1)} & 0 & \phi_{k-1} \\ \beta_{k+1} & \alpha_{k+1} & \beta_{k+2} & 0 \end{array} \right] = \left[ \begin{array}{ccc|c} \gamma_k^{(2)} & \delta_{k+1}^{(2)} & \epsilon_{k+2}^{(1)} & \tau_k \\ 0 & \gamma_{k+1}^{(1)} & \delta_{k+2}^{(1)} & \phi_k \end{array} \right].$$

OpMINRES computes $\mathbf{u}^k$ in $\mathcal{K}_k(\mathbf{P}, \mathbf{y})$ as an approximate solution to the problem $\mathbf{Pu} = \mathbf{y}$:

$$\mathbf{u}^k = Q_k \vartheta_k = Q_k R_k^{-1} t_k = D_k \begin{bmatrix} t_{k-1} \\ \tau_k \end{bmatrix} = \begin{bmatrix} D_{k-1} & d_k \end{bmatrix} \begin{bmatrix} t_{k-1} \\ \tau_k \end{bmatrix}$$
$$= \mathbf{u}^{k-1} + \tau_k d_k.$$

The relation satisfied by $d_k$ is given by,

$$d_k = \frac{1}{\gamma_k^{(2)}} \left( v_k - \delta_k^{(2)} d_{k-1} - \epsilon_k^{(1)} d_{k-2} \right).$$

These details have been incorporated in OpMINRES Algorithm A.7. The OpMINRES Algorithm A.7 is based on approximating an infinite-dimensional problem in (150) by a finite-dimensional problem in (139). As OpMINRES is based on MINRES algorithm (Choi, 2006), the convergence of OpMINRES follows from the convergence of MINRES. The case of singular systems with OpMINRES needs more investigation. In our experiments, the value of relative residual norms $\phi_k/\phi_0$ has been used as the stopping criteria for OpMINRES.

### A.8 Derivation of Linear system of operators

Derivation of $(\mathbf{K} + \lambda I)\mathbf{u} = \mathbf{y}$: We obtain a sufficient condition for stationary points for the optimization problem in Theorem A.2.

Using the representer theorem, the minimization problem can be equivalently formulated as the following problem:

$$\tilde{\mathbf{u}}_\lambda = \arg\min_{\mathbf{u} \in \mathcal{Y}^n} \sum_{i=1}^n \left\| y_i - \sum_{j=1}^n K(x^{(i)}, x^{(j)})u_j \right\|_{\mathcal{Y}}^2 + \lambda \left\langle \sum_{i=1}^n K(x^{(i)}, .)u_i, \sum_{j=1}^n K(x^{(j)}, .)u_j \right\rangle_{\mathcal{H}_{KG}}. \tag{142}$$

We have the following simplification of the term $\left\langle \sum_{i=1}^n K(x^{(i)}, .)u_i, \sum_{j=1}^n K(x^{(j)}, .)u_j \right\rangle_{\mathcal{H}_{KG}}$ in problem (142). We have

$$\left\langle \sum_{i=1}^n K(x^{(i)}, .)u_i, \sum_{j=1}^n K(x^{(j)}, .)u_j \right\rangle_{\mathcal{H}_{KG}} = \sum_{i=1}^n \left\langle K(x^{(i)}, .)u_i, \sum_{j=1}^n K(x^{(j)}, .)u_j \right\rangle_{\mathcal{H}_{KG}} \tag{143}$$

$$= \sum_{i=1}^n \sum_{j=1}^n \left\langle K(x^{(i)}, .)u_i, K(x^{(j)}, .)u_j \right\rangle_{\mathcal{H}_{KG}} \tag{144}$$

$$= \sum_{i=1}^n \sum_{j=1}^n \left\langle K(x^{(i)}, x^{(j)})u_i, u_j \right\rangle_{\mathcal{Y}}. \tag{145}$$

Note that Eq. (143) and Eq. (144) follow from the property of bilinear forms and Eq. (145) follows from the reproducing property of $K$. Thus we have the following simplified formulation:

$$\tilde{\mathbf{u}}_\lambda = \arg\min_{\mathbf{u} \in \mathcal{Y}^n} \sum_{i=1}^n \left\| y_i - \sum_{j=1}^n K(x^{(i)}, x^{(j)})u_j \right\|_{\mathcal{Y}}^2 + \lambda \sum_{i=1,j=1}^n \langle K(x^{(i)}, x^{(j)})u_i, u_j \rangle_{\mathcal{Y}}.$$

To solve this problem, we first construct the objective function $J_\lambda(\mathbf{u})$ given by

$$J_\lambda(\mathbf{u}) = \sum_{i=1}^n \left\| y_i - \sum_{j=1}^n K(x^{(i)}, x^{(j)})u_j \right\|_{\mathcal{Y}}^2 + \lambda \sum_{i=1,j=1}^n \langle K(x^{(i)}, x^{(j)})u_i, u_j \rangle_{\mathcal{Y}}, \quad \mathbf{u} \in \mathcal{Y}^n.$$

Letting $J_\lambda(\mathbf{u}) = \sum_{i=1}^n G_i(\mathbf{u}) + \lambda L(\mathbf{u})$, we can find the directional derivative of $J_\lambda(\mathbf{u})$ with respect to the direction $\mathbf{v}$ as $D_{\mathbf{v}} J_\lambda(\mathbf{u})$.

$$D_{\mathbf{v}} G_i(\mathbf{u}) = \lim_{\tau \to 0} \frac{G_i(u + \tau v) - G_i(u)}{\tau}$$

$$= -2 \left\langle y_i - \sum_{j=1}^n K(x_i, x_j)u_j, \sum_{j=1}^n K(x^{(i)}, x^{(j)})v_j \right\rangle.$$

$$D_{\mathbf{v}} L(\mathbf{u}) = \lim_{\tau \to 0} \frac{L(u + \tau v) - L(u)}{\tau}$$

$$= \lambda \sum_{i,j}^n \langle K(x^{(i)}, x^{(j)})u_i, v_j \rangle + \lambda \sum_{i,j}^n \langle K(x^{(i)}, x^{(j)})v_i, u_j \rangle.$$

As $K$ is Hermitian from the definition of operator-valued kernel, we obtain

$$\langle K(x^{(i)}, x^{(j)})u_i, v_j \rangle = \langle u_i, K(x^{(i)}, x^{(j)})v_j \rangle, \quad \forall i, j \in [n]. \tag{146}$$

Therefore,

$$D_{\mathbf{v}}L(\mathbf{u}) = \lambda \sum_{i,j}^{n} \langle K(x^{(i)}, x^{(j)})u_i, v_j \rangle + \lambda \sum_{i,j}^{n} \langle K(x^{(i)}, x^{(j)})v_i, u_j \rangle$$

$$= \lambda \sum_{i,j}^{n} \langle u_i, K(x^{(i)}, x^{(j)})v_j \rangle + \lambda \sum_{i,j}^{n} \langle K(x^{(i)}, x^{(j)})v_i, u_j \rangle \tag{147}$$

$$= \lambda \sum_{i,j}^{n} \langle u_i, K(x^{(i)}, x^{(j)})v_j \rangle + \lambda \sum_{i,j}^{n} \langle u_j, K(x^{(j)}, x^{(i)})v_i \rangle \tag{148}$$

$$= 2\lambda \sum_{i,j}^{n} \langle u_i, K(x^{(i)}, x^{(j)})v_j \rangle. \tag{149}$$

Eq. (147) follows from Eq. (146) and in Eq. (147), we use symmetry of $\langle \cdot, \cdot \rangle$ to obtain Eq. (149). In order to minimize $J_\lambda(\mathbf{u})$, its directional derivative $D_{\mathbf{v}}J_\lambda(\mathbf{u}) = 0, \ \forall v \in \mathcal{Y}^n$.

$$D_{\mathbf{v}}J_\lambda(\mathbf{u}) = 0$$

$$\implies \sum_{i=1}^{n} D_{\mathbf{v}}G_i(\mathbf{u}) + \lambda D_{\mathbf{v}}L(\mathbf{u}) = 0$$

$$\implies -2\sum_{i=1}^{n} \left\langle y_i - \sum_{j=1}^{n} K(x^{(i)}, x^{(j)})u_j, \sum_{j=1}^{n} K(x^{(i)}, x^{(j)})v_j \right\rangle + 2\lambda \sum_{i,j}^{n} \langle u_i, K(x^{(i)}, x^{(j)})v_j \rangle = 0$$

$$\implies \sum_{i=1}^{n} \left\langle \sum_{j=1}^{n} K(x^{(i)}, x^{(j)})u_j - y_i, \sum_{j=1}^{n} K(x^{(i)}, x^{(j)})v_j \right\rangle + \sum_{i,j}^{n} \langle \lambda u_i, K(x^{(i)}, x^{(j)})v_j \rangle = 0$$

$$\implies \sum_{i=1}^{n} \left\langle \sum_{j=1}^{n} K(x^{(i)}, x^{(j)})u_j - y_i, \sum_{j=1}^{n} K(x^{(i)}, x^{(j)})v_j \right\rangle + \sum_{i=1}^{n} \left\langle \lambda u_i, \sum_{j=1}^{n} K(x^{(i)}, x^{(j)})v_j \right\rangle = 0$$

$$\implies \sum_{i=1}^{n} \left\langle \sum_{j=1}^{n} K(x^{(i)}, x^{(j)})u_j - y_i + \lambda u_i, \sum_{j=1}^{n} K(x^{(i)}, x^{(j)})v_j \right\rangle = 0, \forall v \in \mathcal{Y}^n.$$

The above condition can be reduced to

$$(\mathbf{K} + \lambda I)\mathbf{u} = \mathbf{y}, \tag{150}$$

where $\mathbf{K}$ is a matrix of operators formed by using $K$.

## A.9  Results for Generalization Bounds

We recall the lemma 6.2 and provide the proof next.

**Lemma A.6.** Let $\mathcal{F}$ be a class of functions from $\mathcal{X}$ to $\mathcal{Y}$. Consider a $m$-tuple of samples from input space as $(x^{(1)}, x^{(2)}, \ldots, x^{(m)}) \in \mathcal{X}^m$. Then the following hold:

1. $\mathbb{E}\left[\sup_{F \in \mathcal{F}} \|\frac{1}{m}\sum_{i=1}^{m} F(x^{(i)}) - \mathbb{E}F\|_{\mathcal{Y}}\right] \leq 2\mathscr{R}_{m;\mathcal{Y}}(\mathcal{F})$.

2. For every $c \in \mathbb{R}$, $\mathscr{R}_{m;\mathcal{Y}}(c\mathcal{F}) = |c|\mathscr{R}_{m;\mathcal{Y}}(\mathcal{F})$.

3. For $\phi : \mathcal{Y} \to \mathbb{R}$, if $\phi$ is a Lipschitz function with Lipschitz constant $L$, then $\mathscr{R}_{m;\mathbb{R}}(\phi \circ \mathcal{F}) \leq L\mathscr{R}_{m;\mathcal{Y}}(\mathcal{F})$.

*Proof.* For part 1, we start with denoting $S = (x^{(1)}, x^{(2)}, \ldots, x^{(m)}) \in \mathcal{X}^m$ and another independent sample as $\bar{S} = (\bar{x}^{(1)}, \bar{x}^{(2)}, \ldots, \bar{x}^{(m)}) \in \mathcal{X}^m$, we have

$$\mathbb{E}_{S \sim \mu_{\mathcal{X}}^m}[F(x)] = \mathbb{E}_{\bar{S} \sim \mu_{\mathcal{X}}^m}\left[\frac{1}{m}\sum_{i=1}^{m} F(\bar{x}^{(i)})\right]. \tag{151}$$

We note that the Rademacher random variables $(\varepsilon_1, \varepsilon_2, \ldots, \varepsilon_m)$ are uniformly distributed over $\{+1, -1\}$ and every possible value they take has an equal probability of $1/2^m$. Without loss of generality, we can always permute $(\varepsilon_1, \varepsilon_2, \ldots, \varepsilon_m)$ to obtain $\varepsilon_{P_1} = 1, \ldots, \varepsilon_{P_k} = 1, \varepsilon_{P_{k+1}} = -1, \ldots, \varepsilon_{P_m} = -1$, where $0 \le k \le m$ and $\{P_1, \ldots, P_m\}$ is a permutation of $[m]$. Therefore,

$$
\begin{aligned}
&\mathbb{E}_{S \sim \mu_{\mathcal{X}}^m} \left[ \mathbb{E}_{\bar{S} \sim \mu_{\mathcal{X}}^m} \left[ \sup_{F \in \mathcal{F}} \frac{1}{m} \left\| \sum_{i=1}^m \varepsilon_i \left( F(x^{(i)}) - F(\bar{x}^{(i)}) \right) \right\|_{\mathcal{Y}} \right] \right] \\
&= \mathbb{E}_{S \sim \mu_{\mathcal{X}}^m} \left[ \mathbb{E}_{\bar{S} \sim \mu_{\mathcal{X}}^m} \left[ \sup_{F \in \mathcal{F}} \frac{1}{m} \left\| \sum_{i=1}^k \left( F(x^{(i)}) - F(\bar{x}^{(i)}) \right) + \sum_{i=k+1}^m \left( F(x^{(i)}) - F(\bar{x}^{(i)}) \right) \right\|_{\mathcal{Y}} \right] \right] \\
&= \mathbb{E}_{S \sim \mu_{\mathcal{X}}^m} \left[ \mathbb{E}_{\bar{S} \sim \mu_{\mathcal{X}}^m} \left[ \sup_{F \in \mathcal{F}} \frac{1}{m} \left\| \sum_{i=1}^m \left( F(x^{(i)}) - F(\bar{x}^{(i)}) \right) \right\|_{\mathcal{Y}} \right] \right].
\end{aligned}
\tag{152}
$$

The expressions above hold as $x^{(i)}$ and $\bar{x}^{(i)}$ are independent and symmetric. We obtain the following based on the arguments made above.

$$
\mathbb{E}_{S \sim \mu_{\mathcal{X}}^m} \left[ \sup_{F \in \mathcal{F}} \left\| \frac{1}{m} \sum_{i=1}^m F(x^{(i)}) - \mathbb{E}_{x^{(i)} \sim \mu_{\mathcal{X}}} F(x^{(i)}) \right\|_{\mathcal{Y}} \right]
$$

$$
= \mathbb{E}_{S \sim \mu_{\mathcal{X}}^m} \left[ \sup_{F \in \mathcal{F}} \left\| \frac{1}{m} \sum_{i=1}^m F(x^{(i)}) - \mathbb{E}_{\bar{S} \sim \mu_{\mathcal{X}}^m} \left[ \frac{1}{m} F(\bar{x}^{(i)}) \right] \right\|_{\mathcal{Y}} \right]
\tag{153}
$$

$$
= \mathbb{E}_{S \sim \mu_{\mathcal{X}}^m} \left[ \sup_{F \in \mathcal{F}} \left\| \mathbb{E}_{\bar{S} \sim \mu_{\mathcal{X}}^m} \frac{1}{m} \left( \sum_{i=1}^m F(x^{(i)}) - F(\bar{x}^{(i)}) \right) \right\|_{\mathcal{Y}} \right]
$$

$$
\le \mathbb{E}_{S \sim \mu_{\mathcal{X}}^m} \left[ \mathbb{E}_{\bar{S} \sim \mu_{\mathcal{X}}^m} \left[ \sup_{F \in \mathcal{F}} \left\| \frac{1}{m} \left( \sum_{i=1}^m F(x^{(i)}) - F(\bar{x}^{(i)}) \right) \right\|_{\mathcal{Y}} \right] \right]
\tag{154}
$$

$$
= \mathbb{E}_{S \sim \mu_{\mathcal{X}}^m} \left[ \mathbb{E}_{\bar{S} \sim \mu_{\mathcal{X}}^m} \left[ \mathbb{E} \left[ \sup_{F \in \mathcal{F}} \frac{1}{m} \left\| \sum_{i=1}^m \varepsilon_i \left( F(x^{(i)}) - F(\bar{x}^{(i)}) \right) \right\|_{\mathcal{Y}} \right] \right] \right]
\tag{155}
$$

$$
\le \mathbb{E}_{S \sim \mu_{\mathcal{X}}^m} \left[ \mathbb{E} \left[ \sup_{F \in \mathcal{F}} \frac{1}{m} \left\| \sum_{i=1}^m \varepsilon_i F(x^{(i)}) \right\|_{\mathcal{Y}} \right] \right]
$$

$$
+ \mathbb{E}_{\bar{S} \sim \mu_{\mathcal{X}}^m} \left[ \mathbb{E} \left[ \sup_{F \in \mathcal{F}} \frac{1}{m} \left\| \sum_{i=1}^m \varepsilon_i F(\bar{x}^{(i)}) \right\|_{\mathcal{Y}} \right] \right]
\tag{156}
$$

$$
= 2 \mathscr{R}_{m;\mathcal{Y}}(\mathcal{F}).
$$

Equation (153) follows from (151). Jensen's inequality is used to obtain (154) and (155) follows from (152) by the fact that using Rademacher variables does not change the value of the expression in (223) (see proof of Theorem 4.1 in (Liao, 2020)). The inequality (156) uses the fact that $\varepsilon_i$ and $-\varepsilon_i$ follow the same Rademacher distribution and triangle inequality for norm $\|\cdot\|_{\mathcal{Y}}$ with supremum.

For part 2,

$$
\begin{aligned}
\mathscr{R}_{m;\mathcal{Y}}(c\mathcal{F}) &= \mathbb{E} \left[ \sup_{F \in \mathcal{F}} \frac{1}{m} \left\| \sum_{i=1}^m \varepsilon_i c F(x^{(i)}) \right\|_{\mathcal{Y}} \right] \\
&= |c| \mathbb{E} \left[ \sup_{F \in \mathcal{F}} \frac{1}{m} \left\| \sum_{i=1}^m \varepsilon_i F(x^{(i)}) \right\|_{\mathcal{Y}} \right] \\
&= |c| \mathscr{R}_{m;\mathcal{Y}}(\mathcal{F}).
\end{aligned}
$$

For part 3, consider a $m$-tuple of samples from input space $\mathcal{X}$ as $(x^{(1)}, x^{(2)}, \ldots, x^{(m)}) \in \mathcal{X}^m$ and Rademacher random variables $\varepsilon_i$ for $i \in [m]$. We assume $\phi : \mathcal{Y} \to \mathbb{R}$ and $\phi$ is a Lipschitz function with Lipschitz constant $L$. Then

$$
\begin{aligned}
\mathscr{R}_{m;\mathbb{R}}(\phi \circ \mathcal{F}) &= \mathbb{E}\left[\sup_{F \in \mathcal{F}} \frac{1}{m} \sum_{i=1}^{m} \varepsilon_i (\phi \circ F)(x^{(i)})\right] \\
&= \mathbb{E}_{x^{(i)} \sim \mu_{\mathcal{X}}^m} \frac{1}{m} \left[\mathbb{E}_{\varepsilon \backslash \varepsilon_m} \left[\mathbb{E}_{\varepsilon_m} \left[\sup_{F \in \mathcal{F}} \mathfrak{u}_m(F) + \varepsilon_m (\phi \circ F)(x^{(m)})\right]\right]\right],
\end{aligned} \tag{157}
$$

where $\mathfrak{u}_m(F) = \sum_{i=1}^{m-1} \varepsilon_i (\phi \circ F)(x^{(i)})$.

From the definition of the supremum, for any $\epsilon^m > 0$ (note that $m$ is not an exponent in this notation), there exists $F_1^m, F_2^m \in \mathcal{F}$ such that

$$
\mathfrak{u}_m(F_1^m) + \varepsilon_m (\phi \circ F_1^m)(x^{(m)}) \geq (1 - \epsilon^m) \left[\sup_{F \in \mathcal{F}} \mathfrak{u}_m(F) + \varepsilon_m (\phi \circ F)(x^{(m)})\right] \tag{158}
$$

$$
\mathfrak{u}_m(F_2^m) - \varepsilon_m (\phi \circ F_2^m)(x^{(m)}) \geq (1 - \epsilon^m) \left[\sup_{F \in \mathcal{F}} \mathfrak{u}_m(F) - \varepsilon_m (\phi \circ F)(x^{(m)})\right], \tag{159}
$$

otherwise it leads to a contradiction to the supremum assumption.

Therefore,

$$
\begin{aligned}
(1 - \epsilon^m) \mathbb{E}_{\varepsilon_m} &\left[\sup_{F \in \mathcal{F}} \mathfrak{u}_m(F) + \varepsilon_m (\phi \circ F)(x^{(m)})\right] \\
&= (1 - \epsilon^m) \left[\frac{1}{2} \sup_{F \in \mathcal{F}} \mathfrak{u}_m(F) + (\phi \circ F)(x^{(m)}) + \frac{1}{2} \sup_{F \in \mathcal{F}} \mathfrak{u}_m(F) - (\phi \circ F)(x^{(m)})\right] \tag{160} \\
&\leq \frac{1}{2} \left[\mathfrak{u}_m(F_1^m) + \varepsilon_m (\phi \circ F_1^m)(x^{(m)}) + \mathfrak{u}_m(F_2^m) - \varepsilon_m (\phi \circ F_2^m)(x^{(m)})\right] \tag{161} \\
&= \frac{1}{2} \left[\mathfrak{u}_m(F_1^m) + \mathfrak{u}_m(F_2^m) + \varepsilon_m \left((\phi \circ F_1^m)(x^{(m)}) - (\phi \circ F_2^m)(x^{(m)})\right)\right]. \tag{162} \\
&\leq \sup_{F \in \mathcal{F}} \mathfrak{u}_m(F) + \frac{1}{2} \left[\varepsilon_m \left((\phi \circ F_1^m)(x^{(m)}) - (\phi \circ F_2^m)(x^{(m)})\right)\right]. \tag{163}
\end{aligned}
$$

Equation (160) is obtained by using the definition of Rademacher random variable $\varepsilon_m$. The inequality (161) is obtained using inequalities for $F_1^m$ and $F_2^m$ in (158) and (159), respectively. Inequality (163) is obtained by introducing supremum in the terms $\mathfrak{u}_m(F_1^m) + \mathfrak{u}_m(F_2^m)$. As the inequality (163) holds for any $\epsilon^m > 0$, we claim

$$
\mathbb{E}_{\varepsilon_m} \left[\sup_{F \in \mathcal{F}} \mathfrak{u}_m(F) + \varepsilon_m (\phi \circ F)(x^{(m)})\right] \leq \sup_{F \in \mathcal{F}} \mathfrak{u}_m(F) + \frac{1}{2} \left[\varepsilon_m \left((\phi \circ F_1^m)(x^{(m)}) - (\phi \circ F_2^m)(x^{(m)})\right)\right]. \tag{164}
$$

Using (164) in (157) we obtain

$$
\begin{aligned}
\mathscr{R}_{m;\mathbb{R}}(\phi \circ \mathcal{F}) &= \mathbb{E}_{x^{(i)} \sim \mu_{\mathcal{X}}^m} \frac{1}{m} \left[\mathbb{E}_{\varepsilon \backslash \varepsilon_m} \left[\mathbb{E}_{\varepsilon_m} \left[\sup_{F \in \mathcal{F}} \mathfrak{u}_m(F) + \varepsilon_m (\phi \circ F)(x^{(m)})\right]\right]\right] \\
&\leq \mathbb{E}_{x^{(i)} \sim \mu_{\mathcal{X}}^m} \frac{1}{m} \left[\mathbb{E}_{\varepsilon \backslash \varepsilon_m} \left[\sup_{F \in \mathcal{F}} \mathfrak{u}_m(F) + \frac{1}{2} \left[\varepsilon_m \left((\phi \circ F_1^m)(x^{(m)}) - (\phi \circ F_2^m)(x^{(m)})\right)\right]\right]\right] \tag{165} \\
&\leq \mathbb{E}_{x^{(i)} \sim \mu_{\mathcal{X}}^m} \frac{1}{m} \left[\mathbb{E}_{\varepsilon \backslash \{\varepsilon_{m-1}, \varepsilon_m\}} \left[\mathbb{E}_{\varepsilon_{m-1}} \left[\sup_{F \in \mathcal{F}} \mathfrak{u}_{m-1} + \varepsilon_{m-1} (\phi \circ F)(x^{(m-1)})\right]\right.\right. \\
&\quad \left.\left. + \frac{1}{2} \left[\varepsilon_m \left((\phi \circ F_1^m)(x^{(m)}) - (\phi \circ F_2^m)(x^{(m)})\right)\right]\right]\right], \tag{166}
\end{aligned}
$$

where $\mathfrak{u}_{m-1} = \sup_{F \in \mathcal{F}} \sum_{i=1}^{m-2} \varepsilon_i (\phi \circ F)(x^{(i)})$. Now, for any $\epsilon^{m-1} > 0$ a similar approach is followed to obtain $F_1^{m-1}, F_2^{m-1} \in \mathcal{F}$ such that

$$\mathfrak{u}_{m-1}(F_1^{m-1}) + \varepsilon_{m-1}(\phi \circ F_1^{m-1})(x^{(m-1)}) \geq (1 - \epsilon^{m-1}) \left[ \sup_{F \in \mathcal{F}} \mathfrak{u}_{m-1}(F) + \varepsilon_{m-1}(\phi \circ F)(x^{(m-1)}) \right] \quad (167)$$

$$\mathfrak{u}_{m-1}(F_2^{m-1}) - \varepsilon_{m-1}(\phi \circ F_2^{m-1})(x^{(m-1)}) \geq (1 - \epsilon^{m-1}) \left[ \sup_{F \in \mathcal{F}} \mathfrak{u}_{m-1}(F) - \varepsilon_{m-1}(\phi \circ F)(x^{(m-1)}) \right]. \quad (168)$$

Using (167) and (168), similar arguments as made earlier help us to claim

$$\mathbb{E}_{\varepsilon_{m-1}} \left[ \sup_{F \in \mathcal{F}} \mathfrak{u}_{m-1}(F) + \varepsilon_{m-1}(\phi \circ F)(x^{(m-1)}) \right]$$
$$\leq \sup_{F \in \mathcal{F}} \mathfrak{u}_{m-1}(F) + \frac{1}{2} \left[ \varepsilon_{m-1} \left( (\phi \circ F_1^{m-1})(x^{(m-1)}) - (\phi \circ F_2^{m-1})(x^{(m-1)}) \right) \right]. \quad (169)$$

Inequality (169) with (166) provides the following

$$\mathscr{R}_{m;\mathbb{R}}(\phi \circ \mathcal{F}) \leq \mathbb{E}_{x^{(i)} \sim \mu_{\mathcal{X}}^m} \frac{1}{m} \left[ \mathbb{E}_{\varepsilon \backslash \{\varepsilon_{m-2}, \varepsilon_{m-1}, \varepsilon_m\}} \left[ \mathbb{E}_{\varepsilon_{m-2}} \left[ \sup_{F \in \mathcal{F}} \mathfrak{u}_{m-2} + \varepsilon_{m-2}(\phi \circ F)(x^{(m-2)}) \right] \right. \right.$$
$$\left. \left. + \frac{1}{2} \sum_{i=m-1}^m \left[ \varepsilon_i \left( (\phi \circ F_1^i)(x^{(i)}) - (\phi \circ F_2^i)(x^{(i)}) \right) \right] \right] \right], \quad (170)$$

where $\mathfrak{u}_{m-2} = \sup_{F \in \mathcal{F}} \sum_{i=1}^{m-3} \varepsilon_i (\phi \circ F)(x^{(i)})$. We iterate till the last step where

$$\mathscr{R}_{m;\mathbb{R}}(\phi \circ \mathcal{F}) \leq \mathbb{E}_{x^{(i)} \sim \mu_{\mathcal{X}}^m} \frac{1}{m} \left[ \mathbb{E}_{\varepsilon_1} \left[ \sup_{F \in \mathcal{F}} \varepsilon_1 (\phi \circ F)(x^{(1)}) \right] \right.$$
$$\left. + \frac{1}{2} \sum_{i=2}^m \left[ \varepsilon_i \left( (\phi \circ F_1^i)(x^{(i)}) - (\phi \circ F_2^i)(x^{(i)}) \right) \right] \right]. \quad (171)$$

For any $\epsilon^1 > 0$, there exists $F_1^1, F_2^1 \in \mathcal{F}$ such that

$$\varepsilon_1 (\phi \circ F_1^1)(x^{(1)}) \geq (1 - \epsilon^1) \left[ \sup_{F \in \mathcal{F}} \varepsilon_1 (\phi \circ F)(x^{(1)}) \right] \quad (172)$$

$$-\varepsilon_1 (\phi \circ F_2^1)(x^{(1)}) \geq (1 - \epsilon^1) \left[ \sup_{F \in \mathcal{F}} -\varepsilon_1 (\phi \circ F)(x^{(1)}) \right]. \quad (173)$$

Using (172) and (173), we obtain

$$\mathbb{E}_{\varepsilon_1} \left[ \sup_{F \in \mathcal{F}} \varepsilon_1 (\phi \circ F)(x^{(1)}) \right] \leq \frac{1}{2} \left[ \varepsilon_1 \left( (\phi \circ F_1^1)(x^{(1)}) - (\phi \circ F_2^1)(x^{(1)}) \right) \right]. \quad (174)$$

Next, we simplify (171) using (174),

$$\frac{1}{2} \sum_{i=1}^m \left[ \varepsilon_i \left( (\phi \circ F_1^i)(x^{(i)}) - (\phi \circ F_2^i)(x^{(i)}) \right) \right]$$

$$\leq \frac{1}{2} \left[ \sum_{i=1}^m \varepsilon_i \left( (\phi \circ F_1^i)(x^{(i)}) - (\phi \circ F_2^i)(x^{(i)}) \right) \right] \quad (175)$$

$$= \frac{1}{2} \left[ \left( (\phi \circ \sum_{i=1}^m \varepsilon_i F_1^i)(x^{(i)}) - (\phi \circ \sum_{i=1}^m \varepsilon_i F_2^i)(x^{(i)}) \right) \right] \quad (176)$$

$$\leq L \left\| \sum_{i=1}^m \left( \frac{\varepsilon_i}{2} F_1^i(x^{(i)}) - \frac{\varepsilon_i}{2} F_2^i(x^{(i)}) \right) \right\|_{\mathcal{Y}} \quad (177)$$

$$\leq L \mathbb{E} \sup_{F \in \mathcal{F}} \left\| \sum_{i=1}^m \varepsilon_i F(x^{(i)}) \right\|_{\mathcal{Y}}. \quad (178)$$

Inequality (175) is obtained by using $F_1^i, F_2^i, \forall i \in [m]$ which are obtained based on the procedure for $F_1^m$ and $F_2^m$. The definition of $\phi$ is utilized to establish (176). Inequality (177) is obtained by using Lipschitz continuity of $\Phi$ and (178) follows from the definition of supremum and expectation with respect to Rademacher random variables. Hence using (178) with (174) and (171), we obtain

$$\mathscr{R}_{m;\mathbb{R}}(\phi \circ \mathcal{F}) \leq L\mathscr{R}_{m;\mathcal{Y}}(\mathcal{F}).$$

$\square$

Next, we recall Lemma 6.6 and provide its proof.

**Lemma A.7.** $\mathbb{E}[\Theta(z)]$ is bounded above as follows:

$$\mathbb{E}[\Theta(z)] \leq 2\rho L(\tau)\beta^{1/4}(\mathscr{R}_{m;\mathcal{Y}}(\mathcal{K}_0))^{1/4}.$$

*Proof.*

$$\mathbb{E}[\Theta(z)] = \mathbb{E}\sup_{F \in \rho\mathcal{B}_{\mathcal{K}}}[\mathscr{E}(F) - \mathscr{E}_z(F)] \tag{179}$$

$$\leq 2\mathbb{E}\sup_{F \in \rho\mathcal{B}_{\mathcal{K}}}\left[\frac{1}{m}\sum_{i=1}^{m}\varepsilon_i\mathscr{L}(y^{(i)}, F(x^{(i)}))\right] \tag{180}$$

$$\leq 2L(\tau)\mathscr{R}_{m;\mathcal{Y}}(\rho\mathcal{B}_{\mathcal{K}}) \tag{181}$$

$$= 2\rho L(\tau)\mathscr{R}_{m;\mathcal{Y}}(\mathcal{B}_{\mathcal{K}}). \tag{182}$$

Equation (179) involves a supremum of $F \in \rho\mathcal{B}_{\mathcal{K}}$ as $\|F\|_\infty$ is bounded by $\rho$ and inequality (180) is obtained by using Lemma A.9 in Appendix A.9. Part 3 of Lemma 6.2 is used with $\phi_i(.) = \mathscr{L}(y^{(i)}, .)$ which has a Lipschitz constant $L(\tau)$ in order to obtain (181). (182) follows from Part 2 of Lemma 6.2. Now,

$$\mathscr{R}_{m;\mathcal{Y}}(\mathcal{B}_{\mathcal{K}}) = \mathbb{E}\left[\sup_{F \in \mathcal{B}_{\mathcal{K}}}\frac{1}{m}\left\|\sum_{i=1}^{m}\varepsilon_i F(x^{(i)})\right\|_{\mathcal{Y}}\right]$$

$$= \mathbb{E}\left[\sup_{F \in \mathcal{B}_{\mathcal{K}}}\frac{1}{m}\left(\sum_{i,j=1}^{m}\varepsilon_i\varepsilon_j\left\langle F(x^{(i)}), F(x^{(j)})\right\rangle_{\mathcal{Y}}\right)^{1/2}\right] \tag{183}$$

$$= \mathbb{E}\left[\sup_{F \in \mathcal{B}_{\mathcal{K}}}\frac{1}{m}\left(\sum_{i,j=1}^{m}\varepsilon_i\varepsilon_j\left\langle F, K(x^{(i)}, .)F(x^{(j)})\right\rangle_{\mathcal{H}_K}\right)^{1/2}\right] \tag{184}$$

$$\leq \mathbb{E}\left[\sup_{K \in \mathcal{K}}\sup_{F \in \mathcal{B}_{\mathcal{K}}}\sup_{y \in \mathcal{Y}}\frac{1}{m}\left(\left\langle F, \sum_{i,j=1}^{m}\varepsilon_i\varepsilon_j K(x^{(i)}, .)y\right\rangle_{\mathcal{H}_K}\right)^{1/2}\right] \tag{185}$$

$$\leq \mathbb{E}\left[\sup_{K \in \mathcal{K}}\sup_{\substack{F \in \mathcal{B}_{\mathcal{K}}: \\ \|F\|_{\mathcal{H}_K} \leq 1}}\sup_{y \in \mathcal{Y}}\frac{1}{m}\left(\|F\|_{\mathcal{H}_K}\left\|\sum_{i,j=1}^{m}\varepsilon_i\varepsilon_j K(x^{(i)}, .)y\right\|_{\mathcal{H}_K}\right)^{1/2}\right] \tag{186}$$

$$\leq \mathbb{E}\left[\sup_{K \in \mathcal{K}}\sup_{y \in \mathcal{Y}}\frac{1}{m}\left(\left\|\sum_{i,j=1}^{m}\varepsilon_i\varepsilon_j K(x^{(i)}, .)y\right\|_{\mathcal{H}_K}\right)^{1/2}\right] \tag{187}$$

$$\leq \mathbb{E}\left[\sup_{K \in \mathcal{K}}\sup_{y \in \mathcal{Y}}\frac{1}{\sqrt{m}}\left(\left\|\sum_{i=1}^{m}\varepsilon_i K(x^{(i)}, .)y\right\|_{\mathcal{H}_K}\right)^{1/2}\right] \tag{188}$$

$$= \mathbb{E}\left[\sup_{K \in \mathcal{K}} \sup_{y \in \mathcal{Y}} \frac{1}{\sqrt{m}} \left(\left\langle \sum_{i=1}^{m} \varepsilon_i K(x^{(i)}, .)y, \sum_{j=1}^{m} \varepsilon_j K(x^{(j)}, .)y \right\rangle_{\mathcal{H}_K}\right)^{1/4}\right] \tag{189}$$

$$\leq \beta^{1/4} \mathbb{E}\left[\sup_{K \in \mathcal{K}} \sup_{y \in \mathcal{Y}} \sup_{x \in \mathcal{X}} \frac{1}{\sqrt{m}} \left(\sum_{j=1}^{m} \left\|\sum_{i=1}^{m} \varepsilon_i K(x^{(i)}, x)y\right\|_{\mathcal{Y}}\right)^{1/4}\right] \tag{190}$$

$$= \beta^{1/4} \mathbb{E}\left[\left(\sup_{K \in \mathcal{K}} \sup_{y \in \mathcal{Y}} \sup_{x \in \mathcal{X}} \frac{1}{m} \left\|\sum_{i=1}^{m} \varepsilon_i K(x^{(i)}, x)y\right\|_{\mathcal{Y}}\right)^{1/4}\right] \tag{191}$$

$$= \beta^{1/4} \left(\mathscr{R}_{m;\mathcal{Y}}(\mathcal{K}_0)\right)^{1/4}. \tag{192}$$

The steps in deriving $\mathscr{R}_{m;\mathcal{Y}}(\mathcal{B}_{\mathcal{K}}) \leq \beta^{1/4}(\mathscr{R}_{m;\mathcal{Y}}(\mathcal{K}_0))^{1/4}$ use properties of norm, inner product and reproducing property of OVKs which have been discussed in Lemma A.8 (Appendix A.9). Therefore,

$$\mathbb{E}[\Theta(z)] \leq 2\rho L(\tau)\beta^{1/4}(\mathscr{R}_{m;\mathcal{Y}}(\mathcal{K}_0))^{1/4}.$$

$\square$

We discuss the steps involved in deriving $\mathscr{R}_{m;\mathcal{Y}}(\mathcal{B}_{\mathcal{K}}) \leq \beta^{1/4}(\mathscr{R}_{m;\mathcal{Y}}(\mathcal{K}_0))^{1/4}$ next.

**Lemma A.8.**

$$\mathscr{R}_{m;\mathcal{Y}}(\mathcal{B}_{\mathcal{K}}) \leq \beta^{1/4} \left(\mathscr{R}_{m;\mathcal{Y}}(\mathcal{K}_0)\right)^{1/4}. \tag{193}$$

*Proof.*

$$\mathscr{R}_{m;\mathcal{Y}}(\mathcal{B}_{\mathcal{K}}) = \mathbb{E}\left[\sup_{F \in \mathcal{B}_{\mathcal{K}}} \frac{1}{m} \left\|\sum_{i=1}^{m} \varepsilon_i F(x^{(i)})\right\|_{\mathcal{Y}}\right]$$

$$= \mathbb{E}\left[\sup_{F \in \mathcal{B}_{\mathcal{K}}} \frac{1}{m} \left(\left\langle \sum_{i=1}^{m} \varepsilon_i F(x^{(i)}), \sum_{j=1}^{m} \varepsilon_m F(x^{(j)}) \right\rangle_{\mathcal{Y}}\right)^{1/2}\right] \tag{194}$$

$$= \mathbb{E}\left[\sup_{F \in \mathcal{B}_{\mathcal{K}}} \frac{1}{m} \left(\sum_{i,j=1}^{m} \varepsilon_i \varepsilon_m \left\langle F(x^{(i)}), F(x^{(j)}) \right\rangle_{\mathcal{Y}}\right)^{1/2}\right] \tag{195}$$

$$= \mathbb{E}\left[\sup_{F \in \mathcal{B}_{\mathcal{K}}} \frac{1}{m} \left(\sum_{i,j=1}^{m} \varepsilon_i \varepsilon_m \left\langle F, K(x^{(i)}, .)F(x^{(j)}) \right\rangle_{\mathcal{H}_K}\right)^{1/2}\right] \tag{196}$$

$$= \mathbb{E}\left[\sup_{F \in \mathcal{B}_{\mathcal{K}}} \frac{1}{m} \left(\left\langle F, \sum_{i,j=1}^{m} \varepsilon_i \varepsilon_m K(x^{(i)}, .)F(x^{(j)}) \right\rangle_{\mathcal{H}_K}\right)^{1/2}\right] \tag{197}$$

$$\leq \mathbb{E}\left[\sup_{K \in \mathcal{K}} \sup_{F \in \mathcal{B}_{\mathcal{K}}} \frac{1}{m} \left(\left\langle F, \sum_{i,j=1}^{m} \varepsilon_i \varepsilon_m K(x^{(i)}, .)F(x^{(j)}) \right\rangle_{\mathcal{H}_K}\right)^{1/2}\right] \tag{198}$$

$$\leq \mathbb{E}\left[\sup_{K \in \mathcal{K}} \sup_{F \in \mathcal{B}_{\mathcal{K}}} \sup_{y \in \mathcal{Y}} \frac{1}{m} \left(\left\langle F, \sum_{i,j=1}^{m} \varepsilon_i \varepsilon_m K(x^{(i)}, .)y \right\rangle_{\mathcal{H}_K}\right)^{1/2}\right] \tag{199}$$

$$\leq \mathbb{E}\left[\sup_{K\in\mathcal{K}}\sup_{F\in\mathcal{B}_{\mathcal{K}}}\sup_{y\in\mathcal{Y}}\frac{1}{m}\left(\|F\|_{\mathcal{H}_K}\left\|\sum_{i,j=1}^{m}\varepsilon_i\varepsilon_m K(x^{(i)},.)y\right\|_{\mathcal{H}_K}\right)^{1/2}\right] \tag{200}$$

$$=\mathbb{E}\left[\sup_{K\in\mathcal{K}}\sup_{\substack{F\in\mathcal{B}_{\mathcal{K}}:\\\|F\|_{\mathcal{H}_K}\leq 1}}\sup_{y\in\mathcal{Y}}\frac{1}{m}\left(\|F\|_{\mathcal{H}_K}\left\|\sum_{i,j=1}^{m}\varepsilon_i\varepsilon_m K(x^{(i)},.)y\right\|_{\mathcal{H}_K}\right)^{1/2}\right] \tag{201}$$

$$\leq \mathbb{E}\left[\sup_{K\in\mathcal{K}}\sup_{y\in\mathcal{Y}}\frac{1}{m}\left(\left\|\sum_{i,j=1}^{m}\varepsilon_i\varepsilon_m K(x^{(i)},.)y\right\|_{\mathcal{H}_K}\right)^{1/2}\right] \tag{202}$$

$$=\mathbb{E}\left[\sup_{K\in\mathcal{K}}\sup_{y\in\mathcal{Y}}\frac{1}{m}\left(\left\|\sum_{j=1}^{m}\varepsilon_m\sum_{i=1}^{m}\varepsilon_i K(x^{(i)},.)y\right\|_{\mathcal{H}_K}\right)^{1/2}\right] \tag{203}$$

$$\leq \mathbb{E}\left[\sup_{K\in\mathcal{K}}\sup_{y\in\mathcal{Y}}\frac{1}{m}\left(\sum_{j=1}^{m}|\varepsilon_m|\left\|\sum_{i=1}^{m}\varepsilon_i K(x^{(i)},.)y\right\|_{\mathcal{H}_K}\right)^{1/2}\right] \tag{204}$$

$$=\mathbb{E}\left[\sup_{K\in\mathcal{K}}\sup_{y\in\mathcal{Y}}\frac{1}{\sqrt{m}}\left(\left\|\sum_{i=1}^{m}\varepsilon_i K(x^{(i)},.)y\right\|_{\mathcal{H}_K}\right)^{1/2}\right] \tag{205}$$

$$=\mathbb{E}\left[\sup_{K\in\mathcal{K}}\sup_{y\in\mathcal{Y}}\frac{1}{\sqrt{m}}\left(\left\langle\sum_{i=1}^{m}\varepsilon_i K(x^{(i)},.)y,\sum_{j=1}^{m}\varepsilon_m K(x^{(j)},.)y\right\rangle_{\mathcal{H}_K}\right)^{1/4}\right] \tag{206}$$

$$=\mathbb{E}\left[\sup_{K\in\mathcal{K}}\sup_{y\in\mathcal{Y}}\frac{1}{\sqrt{m}}\left(\sum_{i,j=1}^{m}\varepsilon_i\varepsilon_m\langle K(x^{(i)},.)y,K(x^{(j)},.)y\rangle_{\mathcal{H}_K}\right)^{1/4}\right] \tag{207}$$

$$=\mathbb{E}\left[\sup_{K\in\mathcal{K}}\sup_{y\in\mathcal{Y}}\frac{1}{\sqrt{m}}\left(\sum_{i,j=1}^{m}\varepsilon_i\varepsilon_m\langle K(x^{(i)},x^{(j)})y,y\rangle_{\mathcal{Y}}\right)^{1/4}\right] \tag{208}$$

$$=\mathbb{E}\left[\sup_{K\in\mathcal{K}}\sup_{y\in\mathcal{Y}}\frac{1}{\sqrt{m}}\left(\left\langle\sum_{i,j=1}^{m}\varepsilon_i\varepsilon_m K(x^{(i)},x^{(j)})y,y\right\rangle_{\mathcal{Y}}\right)^{1/4}\right] \tag{209}$$

$$\leq \mathbb{E}\left[\sup_{K\in\mathcal{K}}\sup_{y\in\mathcal{Y}}\frac{1}{\sqrt{m}}\left(\left\|\sum_{i,j=1}^{m}\varepsilon_i\varepsilon_m K(x^{(i)},x^{(j)})y\right\|_{\mathcal{Y}}\|y\|_{\mathcal{Y}}\right)^{1/4}\right] \tag{210}$$

$$\leq \beta^{1/4}\mathbb{E}\left[\sup_{K\in\mathcal{K}}\sup_{y\in\mathcal{Y}}\frac{1}{\sqrt{m}}\left(\left\|\sum_{i,j=1}^{m}\varepsilon_i\varepsilon_m K(x^{(i)},x^{(j)})y\right\|_{\mathcal{Y}}\right)^{1/4}\right] \tag{211}$$

$$=\beta^{1/4}\mathbb{E}\left[\sup_{K\in\mathcal{K}}\sup_{y\in\mathcal{Y}}\frac{1}{\sqrt{m}}\left(\left\|\sum_{j=1}^{m}\varepsilon_m\sum_{i=1}^{m}\varepsilon_i K(x^{(i)},x^{(j)})y\right\|_{\mathcal{Y}}\right)^{1/4}\right] \tag{212}$$

$$\leq \beta^{1/4}\mathbb{E}\left[\sup_{K\in\mathcal{K}}\sup_{y\in\mathcal{Y}}\frac{1}{\sqrt{m}}\left(\sum_{j=1}^{m}|\varepsilon_m|\left\|\sum_{i=1}^{m}\varepsilon_i K(x^{(i)},x^{(j)})y\right\|_{\mathcal{Y}}\right)^{1/4}\right] \tag{213}$$

$$= \beta^{1/4} \mathbb{E} \left[ \sup_{K \in \mathcal{K}} \sup_{y \in \mathcal{Y}} \frac{1}{\sqrt{m}} \left( \sum_{j=1}^{m} \left\| \sum_{i=1}^{m} \varepsilon_i K(x^{(i)}, x^{(j)}) y \right\|_{\mathcal{Y}} \right)^{1/4} \right] \tag{214}$$

$$\leq \beta^{1/4} \mathbb{E} \left[ \sup_{K \in \mathcal{K}} \sup_{y \in \mathcal{Y}} \sup_{x \in \mathcal{X}} \frac{1}{\sqrt{m}} \left( \sum_{j=1}^{m} \left\| \sum_{i=1}^{m} \varepsilon_i K(x^{(i)}, x) y \right\|_{\mathcal{Y}} \right)^{1/4} \right] \tag{215}$$

$$= \beta^{1/4} \mathbb{E} \left[ \left( \sup_{K \in \mathcal{K}} \sup_{y \in \mathcal{Y}} \sup_{x \in \mathcal{X}} \frac{1}{m} \left\| \sum_{i=1}^{m} \varepsilon_i K(x^{(i)}, x) y \right\|_{\mathcal{Y}} \right)^{1/4} \right] \tag{216}$$

$$= \beta^{1/4} \left( \mathscr{R}_{m;\mathcal{Y}}(\mathcal{K}_0) \right)^{1/4} . \tag{217}$$

Reproducing property of OVK $K$ is used to obtain (196) and (208). Inequalities (200) and 210 are obtained by using the Cauchy-Schwarz inequality. (202) is obtained by using the definition of $\mathcal{B}_\mathcal{K}$ with $\|F\|_{\mathcal{H}_K} \leq 1$. The rest of the steps follow from the properties of inner-product and norm in $\mathcal{H}_K$ and $\mathcal{Y}$. $\qquad \square$

**Lemma A.9.**

$$\mathbb{E} \sup_{F \in \rho \mathcal{B}_\mathcal{K}} [\mathscr{E}(F) - \mathscr{E}_z(F)] \leq 2\mathbb{E} \left[ \sup_{F \in \rho \mathcal{B}_\mathcal{K}} \frac{1}{m} \sum_{i=1}^{m} \varepsilon_i \mathscr{L}(y^{(i)}, F(x^{(i)})) \right] .$$

*Proof.*

$$\mathbb{E} \sup_{F \in \rho \mathcal{B}_\mathcal{K}} [\mathscr{E}(F) - \mathscr{E}_z(F)] = \mathbb{E}_{z \sim \mathfrak{z}^m} \sup_{F \in \rho \mathcal{B}_\mathcal{K}} \left[ \frac{1}{m} \sum_{i=1}^{m} \mathbb{E}_{(x', y') \sim \mathfrak{z}} \mathscr{L}(y'_i, F(x'_i)) - \mathscr{E}_z(F) \right] \tag{218}$$

$$= \mathbb{E}_{z \sim \mathfrak{z}^m} \sup_{F \in \rho \mathcal{B}_\mathcal{K}} \left[ \mathbb{E}_{z' \sim \mathfrak{z}^m} \frac{1}{m} \sum_{i=1}^{m} \mathscr{L}(y'_i, F(x'_i)) - \frac{1}{m} \sum_{i=1}^{m} \mathscr{L}(y^{(i)}, F(x^{(i)})) \right] \tag{219}$$

$$= \mathbb{E}_{z \sim \mathfrak{z}^m} \left[ \sup_{F \in \rho \mathcal{B}_\mathcal{K}} \mathbb{E}_{z' \sim \mathfrak{z}^m} \left[ \frac{1}{m} \sum_{i=1}^{m} \left( \mathscr{L}(y'_i, F(x'_i)) - \mathscr{L}(y^{(i)}, F(x^{(i)})) \right) \right] \right] . \tag{220}$$

Consider a function $f$ dependent on two random variables $X, Y$ in a class of functions $\mathcal{F}$. Then

$$\implies f(X, Y) \leq \sup_{f \in \mathcal{F}} f(X, Y)$$
$$\implies \mathbb{E}_Y[f(X, Y)] \leq \mathbb{E}_Y[\sup_{f \in \mathcal{F}} f(X, Y)]$$
$$\implies \sup_{f \in \mathcal{F}} \mathbb{E}_Y[f(X, Y)] \leq \mathbb{E}_Y[\sup_{f \in \mathcal{F}} f(X, Y)]$$
$$\implies \mathbb{E}_X \sup_{f \in \mathcal{F}} \mathbb{E}_Y[f(X, Y)] \leq \mathbb{E}_X \mathbb{E}_Y[\sup_{f \in \mathcal{F}} f(X, Y)] . \tag{221}$$

Therefore, using the property established in (221), we obtain

$$\mathbb{E} \sup_{F \in \rho \mathcal{B}_\mathcal{K}} [\mathscr{E}(F) - \mathscr{E}_z(F)] = \mathbb{E}_{z \sim \mathfrak{z}^m} \left[ \sup_{F \in \rho \mathcal{B}_\mathcal{K}} \mathbb{E}_{z' \sim \mathfrak{z}^m} \left[ \frac{1}{m} \sum_{i=1}^{m} \left( \mathscr{L}(y'_i, F(x'_i)) - \mathscr{L}(y^{(i)}, F(x^{(i)})) \right) \right] \right] \tag{222}$$

$$\leq \mathbb{E}_{z \sim \mathfrak{z}^m} \mathbb{E}_{z' \sim \mathfrak{z}^m} \left[ \sup_{F \in \rho \mathcal{B}_\mathcal{K}} \frac{1}{m} \sum_{i=1}^{m} \left( \mathscr{L}(y'_i, F(x'_i)) - \mathscr{L}(y^{(i)}, F(x^{(i)})) \right) \right] \tag{223}$$

$$= \mathbb{E}_{z, z' \sim \mathfrak{z}^m, \varepsilon} \left[ \sup_{F \in \rho \mathcal{B}_\mathcal{K}} \frac{1}{m} \sum_{i=1}^{m} \varepsilon_i \left( \mathscr{L}(y'_i, F(x'_i)) - \mathscr{L}(y^{(i)}, F(x^{(i)})) \right) \right] \tag{224}$$

$$=\mathbb{E}_{z,z'\sim\mathfrak{Z}^m,\varepsilon}\left[\sup_{F\in\rho\mathcal{B}_\mathcal{K}}\left(\frac{1}{m}\sum_{i=1}^m\varepsilon_i\mathscr{L}(y_i',F(x_i'))\right)\right.$$
$$\left.+\sup_{F\in\rho\mathcal{B}_\mathcal{K}}\left(\frac{1}{m}\sum_{i=1}^m(-\varepsilon_i)\mathscr{L}(y^{(i)},F(x^{(i)}))\right)\right] \quad (225)$$

$$=\mathbb{E}_{z'\sim\mathfrak{Z}^m,\varepsilon}\left[\sup_{F\in\rho\mathcal{B}_\mathcal{K}}\left(\frac{1}{m}\sum_{i=1}^m\varepsilon_i\mathscr{L}(y_i',F(x_i'))\right)\right]$$
$$+\mathbb{E}_{z\sim\mathfrak{Z}^m,\varepsilon}\left[\sup_{F\in\rho\mathcal{B}_\mathcal{K}}\left(\frac{1}{m}\sum_{i=1}^m\varepsilon_i\mathscr{L}(y^{(i)},F(x^{(i)}))\right)\right] \quad (226)$$

$$=2\mathbb{E}\left[\sup_{F\in\rho\mathcal{B}_\mathcal{K}}\frac{1}{m}\sum_{i=1}^m\varepsilon_i\mathscr{L}(y^{(i)},F(x^{(i)}))\right]. \quad (227)$$

(224) follows from (223) by the fact that using Rademacher variables does not change the value of the expression in (223) (see proof of Theorem 4.1 in (Liao, 2020)). (225) follows from (224) as $-\varepsilon_i$ has the same distribution as $\varepsilon_i$. (227) is obtained from (226) as $z$ and $z'$ follow identical distribution. □

## A.10 Details of Experiments

In order to illustrate the effectiveness of the developed framework, we have used functional regression problem with an unknown graph structure in the input data for both synthetic and real datasets. The task of predicting output functions with the help of a Laplacian matrix denoting the relationship between the set of $p$ input functions has been illustrated in the experiments. As practical data is always available as discrete observations corresponding to functions, standard FDA techniques can be used for the conversion of functional data into vector representation using basis functions, e.g. Fourier basis, B-spline basis, etc. Let $\mathcal{X}=(L^2([a,b]))^p$ and $\mathcal{Y}=L^2([c,d])$ be the input and output spaces, respectively. For our experiments, the error metric used is residual sum of squares error (RSSE) (Kadri et al., 2016) defined as $RSSE=\int_c^d\sum_i\{y^{(i)}(t)-\hat{y}^{(i)}(t)\}^2dt$, where $y^{(i)}$ is the actual output function and $\hat{y}^{(i)}$ is the predicted output function. RSSE is better suited to compare functional outputs. The integrals involved have been approximated by using numerical integration in our implementation. The quadratic programs involved in (23) and (15) are solved by using CVXOPT (Andersen et al., 2023).

**Experimental Setting**: All methods were coded in Python 3.7. All experiments were run on a Linux box with 182 Gigabytes main memory and 28 CPU cores. As methods to solve the problem of functional regression problem simultaneously with learning $L$ and/or $D$ are not available, we use popular algorithms to first determine $L$. Then for the learned $L$, we use our alternating minimization framework to learn $D$ using projected gradient descent and $\mathbf{u}$ using OpMINRES. For the MCP-based $L$ learning and $D$ learning in the proposed alternating minimization framework, we use a decaying step-size in the projected gradient descent. The decaying step-size regime involves starting with an initial step-size (e.g. $10^{-4}$) and reducing it by a fixed factor (e.g. 2) after a set of iterations (e.g. 5) continuously till a final step-size (e.g. $10^{-9}$). Section 7 includes the details of the methods fglasso-OpMINRES-D, KGL-OpMINRES-D, Sparse Non-Pos-OpMINRES-L-D and Sparse OpMINRES-L-D which we use in this section.

### A.10.1 Experiments with synthetic data

**Data Generation**: For synthetic experiments, three sets of experiments have been considered with input functions for graph structures having 3-nodes, 12-nodes and 25-nodes, respectively. Here, we discuss the data generation for all three settings and results for 3-nodes and 25-nodes setting. For all the methods, a truncated trigonometric basis of $L^2([0,2\pi])$ with 30 basis functions has been considered for encoding the functional data. The experiments were run for three settings where the data has been divided randomly into a training set, a validation set and a test set. The following data splits have been considered: (80/20/20), (160/40/40) and (320/80/80), representing the number of training samples/validation samples/test samples. The data generation is discussed below.

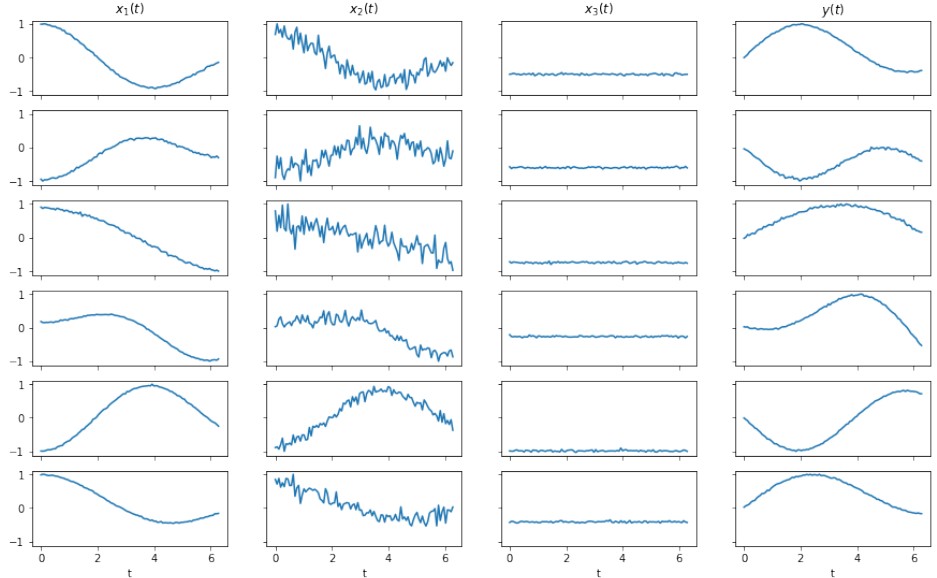

Figure 3: Samples from the 3-node based synthetic data.

For 3-node setting,

$$x_1(t) = \sum_{i=1}^{P} w_i \cos(\alpha_i t) \qquad x_2(t) = \sum_{i=1}^{P} w_i \cos(\alpha_i t) + \epsilon_i \qquad x_3(t) = b,$$

$$y(t) = \sum_{i=1}^{P} w_i \sin(\alpha_i t),$$

where $t \in [0, 2\pi], w_i, b \in U([-1, 1]), \alpha_i \in U([0, 1]), \epsilon_i \in N(0, \sigma^2)$, for $i \in [P], \sigma \in U([0, 0.25])$. The functions are sampled at 100 points and normalization has been done after introducing Gaussian noise with 0.02 standard deviation for both input and output functions. Figure 3 includes some samples generated from the dataset. For 12-node setting,

$$x_1(t) = \sum_{i=1}^{P} w_i^1 \cos(\alpha_i^1 t) \qquad\qquad x_7(t) = b_1 + noise_1$$

$$x_2(t) = \sum_{i=1}^{P} w_i^2 \cos(\alpha_i^1 t) \qquad\qquad x_8(t) = b_2 + noise_2$$

$$x_3(t) = \sum_{i=1}^{P} w_i^3 \cos(\alpha_i^1 t) \qquad\qquad x_9(t) = b_3 + noise_3$$

$$x_4(t) = \sum_{i=1}^{P} w_i^1 \cos(\alpha_i^2 t) \qquad\qquad x_{10}(t) = b_4$$

$$x_5(t) = \sum_{i=1}^{P} w_i^2 \cos(\alpha_i^2 t) \qquad\qquad x_{11}(t) = b_5$$

$$x_6(t) = \sum_{i=1}^{P} w_i^3 \cos(\alpha_i^2 t) \qquad\qquad x_{12}(t) = b_6$$

$$y(t) = \sum_{i=1}^{P} (w_i^1 + w_i^2 + w_i^3) \sin((\alpha_i^1 + \alpha_i^2)t),$$

where $t \in [0, 2\pi]$, $w_i^j, b_1, b_2, b_3 \in U([-1, 1])$, $b_4, b_5, b_6, \alpha_i^k \in U([0, 1])$, $\epsilon_i \in N(0, \sigma^2)$, for $i \in [P], j = 1, 2, 3, k = 1, 2, \sigma \in U([0, 0.25])$ and $noise_l \in N(0, 0.25^2), l = 1, 2, 3$ for $l$-th partition of $[0, 2\pi]$. The functions are sampled at 100 points and normalized after Gaussian noise with 0.02 standard deviation being introduced for both. Figure 4 includes some samples generated from the dataset. Note that results for 12 node case have been discussed in the main paper.

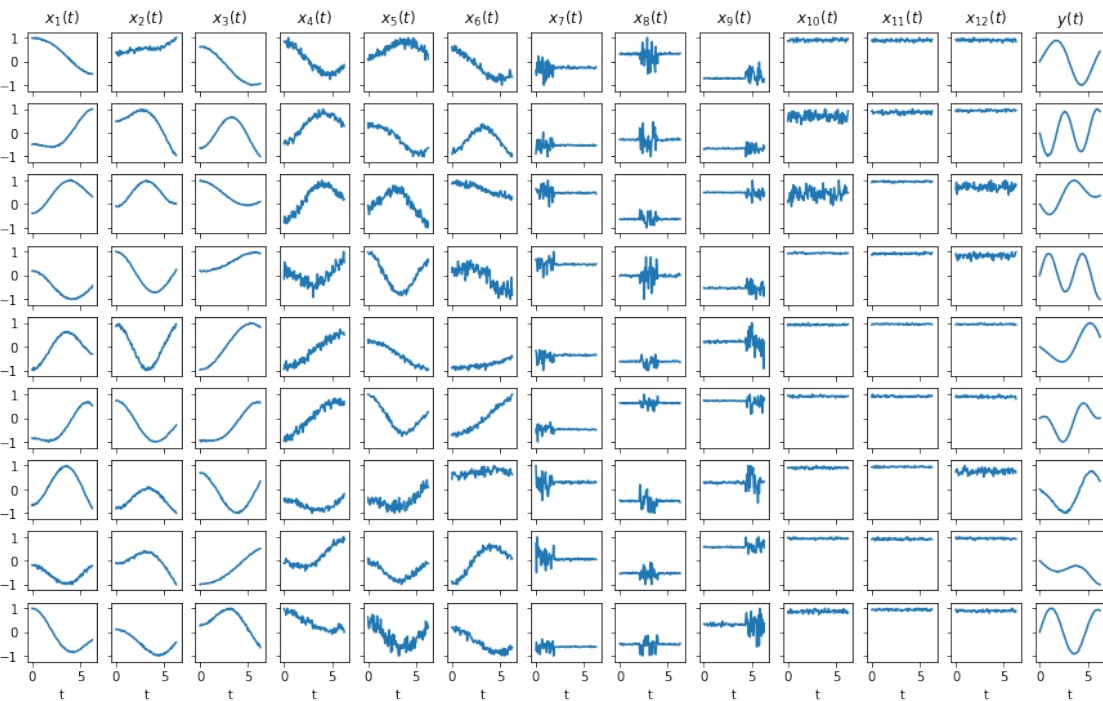

Figure 4: Samples from the 12-node based synthetic data.

For 25-node setting,

$$x_1(t) = \sum_{i=1}^{P} w_i^1 \cos(\alpha_i^1 t) \qquad\qquad x_2(t) = \sum_{i=1}^{P} w_i^2 \cos(\alpha_i^1 t)$$

$$x_3(t) = \sum_{i=1}^{P} w_i^3 \cos(\alpha_i^1 t) \qquad\qquad x_4(t) = \sum_{i=1}^{P} w_i^4 \cos(\alpha_i^1 t)$$

$$x_5(t) = \sum_{i=1}^{P} w_i^5 \cos(\alpha_i^1 t) \qquad\qquad x_6(t) = \sum_{i=1}^{P} w_i^1 \cos(\alpha_i^2 t)$$

$$x_7(t) = \sum_{i=1}^{P} w_i^2 \cos(\alpha_i^2 t) \qquad\qquad x_8(t) = \sum_{i=1}^{P} w_i^3 \cos(\alpha_i^2 t)$$

$$x_9(t) = \sum_{i=1}^{P} w_i^4 \cos(\alpha_i^2 t) \qquad\qquad x_{10}(t) = \sum_{i=1}^{P} w_i^5 \cos(\alpha_i^2 t)$$

$$x_{11}(t) = b_1 + noise_1 \qquad\qquad x_{12}(t) = b_2 + noise_2$$
$$x_{13}(t) = b_3 + noise_3 \qquad\qquad x_{14}(t) = b_4 + noise_4$$
$$x_{15}(t) = b_5 + noise_5 \qquad\qquad x_{16}(t) = c_1$$
$$x_{17}(t) = c_2 \qquad\qquad x_{18}(t) = c_3$$
$$x_{19}(t) = c_4 \qquad\qquad x_{20}(t) = c_5$$

$$x_{21}(t) = d_1 \qquad x_{22}(t) = d_2 \qquad x_{23}(t) = d_3 \qquad x_{24}(t) = d_4 \qquad x_{25}(t) = d_5$$

$$y(t) = \sum_{i=1}^{P}(w_i^1 + w_i^2 + w_i^3 + w_i^4)\sin((\alpha_i^1 + \alpha_i^2)t),$$

where $t \in [0, 2\pi], w_i^j, b_j \in U([-1, 1]), \alpha_i^k, c_j \in U([0, 1]), d_j \in U([0, -1])\epsilon_i \in N(0, \sigma^2)$, for $i \in [P], j \in [5], k = 1, 2, \sigma \in U([0, 0.25])$ and $noise_l \in N(0, 0.25^2), l \in [5]$ for $l$-th partition of $[0, 2\pi]$. The functions are sampled at 100 points and normalization has been done after introducing Gaussian noise with 0.02 standard deviation for both input and output functions. Figure 5 includes some samples generated from the dataset.

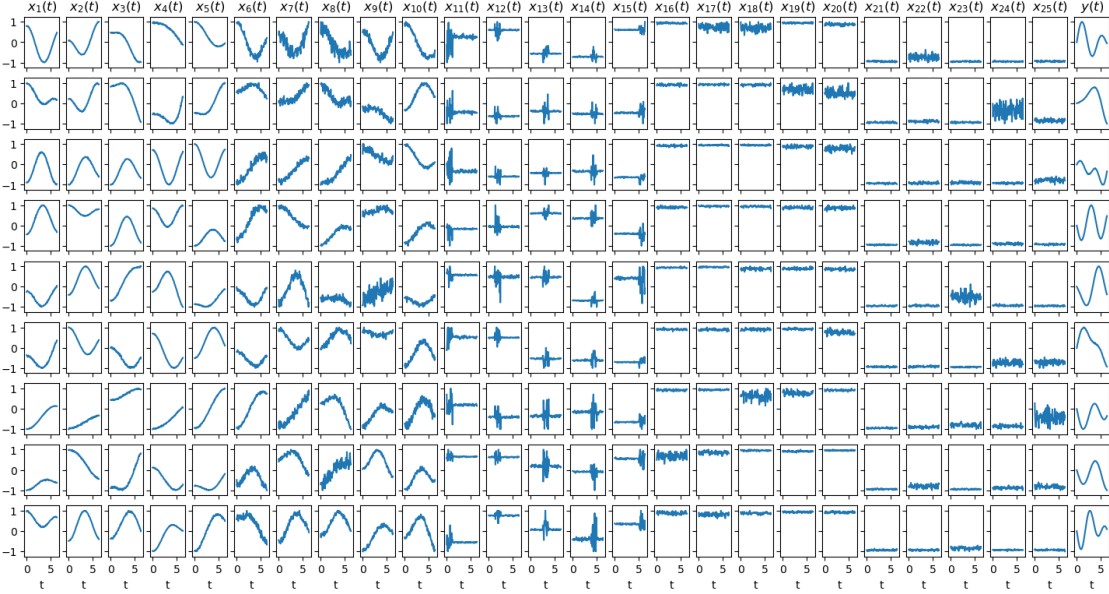

Figure 5: Samples from the 25-node based synthetic data.

The results for synthetic data is summarized in Tables 9-13 where Sparse OpMINRES-L-D attains comparable performance with learned sparse graphs illustrating important relationships driving the functional regression. Table 9 shows that Sparse OpMINRES-L-D provides comparable results to other methods with respect to the mean RSSE on test data for 3-nodes setting. Table 10 contains the $D$ values learned for experiments with 3 nodes which improves the performance in functional regression task in a regularized manner. For 3-nodes setting, the data generation process involves similar information corresponding to node 1 and 2, whereas node 3 involves random constants. Sparse OpMINRES-L-D captures relationship which includes sparse relation between nodes 1, 2 and 3. fglasso-OpMINRES-L-D and KGL-OpMINRES-L-D learn fully connected graphs in Table 12.

Table 11 showcases the mean RSSE results for the functional regression problem for 25-nodes experiment where Sparse OpMINRES-L-D produces comparable results on the test data. In 25-nodes setting, the data generation process involves varied information in nodes 1-10, whereas nodes 11-25 contain information which does not impact the generation of the output function $y$. The graphs obtained for Sparse OpMINRES-L-D in

Table 9: Mean RSSE results for 3-node synthetic data.

| Train/Val/Test samples | Methods | Mean RSSE | | |
|---|---|---|---|---|
| | | Train | Val | Test |
| 80/20/20 | Sparse OpMINRES-L-D | 0.188184 | 0.117119 | 0.104051 |
| | fglasso-OpMINRES-D | 0.109485 | 0.124759 | 0.124163 |
| | KGL-OpMINRES-D | 0.086952 | 0.124781 | 0.183851 |
| | Sparse Non-Pos-OpMINRES-L-D | 0.064445 | 0.158646 | 0.149415 |
| 160/40/40 | Sparse OpMINRES-L-D | 0.040655 | 0.211139 | 0.198398 |
| | fglasso-OpMINRES-D | 0.062094 | 0.183526 | 0.193644 |
| | KGL-OpMINRES-D | 0.07809 | 0.181575 | 0.201469 |
| | Sparse Non-Pos-OpMINRES-L-D | 0.089997 | 0.19503 | 0.218885 |
| 320/80/80 | Sparse OpMINRES-L-D | 0.046926 | 0.153285 | 0.274924 |
| | fglasso-OpMINRES-D | 0.095652 | 0.145256 | 0.281665 |
| | KGL-OpMINRES-D | 0.057804 | 0.145889 | 0.271459 |
| | Sparse Non-Pos-OpMINRES-L-D | 0.087422 | 0.150039 | 0.274598 |

Table 10: $D$ for 3-node synthetic data.

| Train/ Val/ Test samples | Sparse OpMINRES-L-D | | | fglasso-OpMINRES-D | | | KGL-OpMINRES-D | | |
|---|---|---|---|---|---|---|---|---|---|
| 80/20/20 | 0.011319 | 0.036068 | 0.119428 | 1.001583 | 1.002949 | 1.000191 | 1.000672 | 1.002477 | 0.999940 |
| 160/40/40 | 1.104571 | 0.747521 | 1.229991 | 1.008645 | 1.007241 | 1.001036 | 1.010710 | 1.009007 | 1.000914 |
| 320/80/80 | 1.227095 | 0.761956 | 0.956403 | 1.019795 | 1.012624 | 1.001624 | 1.026910 | 1.019523 | 1.004683 |

Table 11: Mean RSSE results for 25-node synthetic data.

| Train/Val/Test samples | Methods | Mean RSSE | | |
|---|---|---|---|---|
| | | Train | Val | Test |
| 80/20/20 | Sparse OpMINRES-L-D | 0.754677 | 1.567458 | 1.822983 |
| | fglasso-OpMINRES-D | 0.905085 | 1.527465 | 1.605906 |
| | KGL-OpMINRES-D | 0.934007 | 1.478522 | 1.594960 |
| | Sparse Non-Pos-OpMINRES-L-D | 0.922789 | 1.53816 | 1.564855 |
| 160/40/40 | Sparse OpMINRES-L-D | 0.662029 | 1.549598 | 1.215493 |
| | fglasso-OpMINRES-D | 0.678837 | 1.602842 | 1.231550 |
| | KGL-OpMINRES-D | 0.742796 | 1.571745 | 1.212629 |
| | Sparse Non-Pos-OpMINRES-L-D | 0.600849 | 1.573893 | 1.215729 |
| 320/80/80 | Sparse OpMINRES-L-D | 0.767516 | 1.436166 | 1.436166 |
| | fglasso-OpMINRES-D | 1.063051 | 1.385366 | 1.429937 |
| | KGL-OpMINRES-D | 1.069962 | 1.356429 | 1.425034 |
| | Sparse Non-Pos-OpMINRES-L-D | 0.668318 | 1.418141 | 1.437544 |

Table 13 show connections majorly between nodes 1-15. Though the input functions for nodes 11-15 contain noisy random constant values, this information seems to be associated with input functions for nodes 5-10. Sparse Non-Pos-OpMINRES-L-D also discovers relations in the clusters of nodes 1-10 and 11-25 majorly.

Table 12: Graphs corresponding to learned $L$ for 3-node synthetic data.

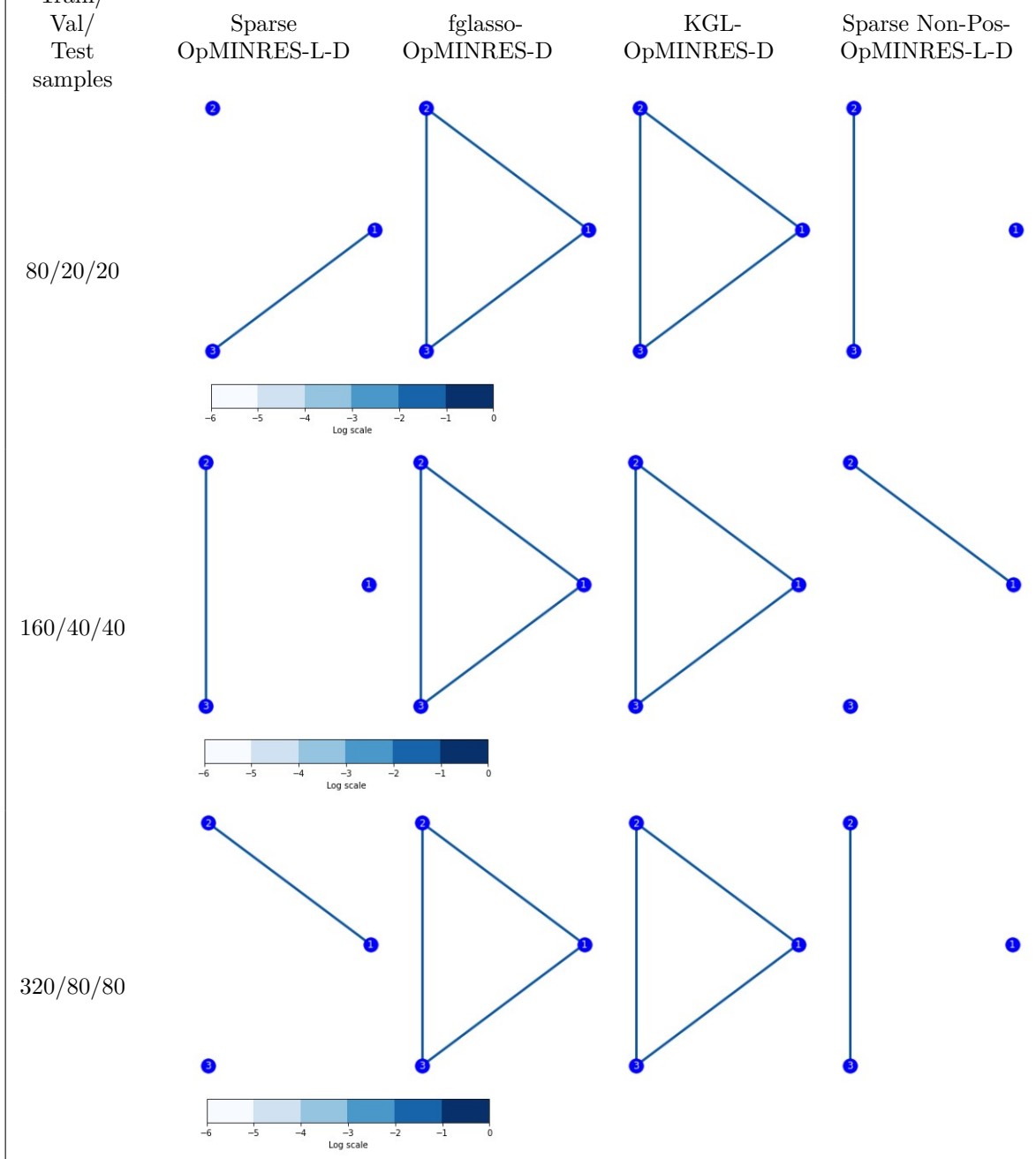

### A.10.2 Experiments on weather data

Weather data is dynamic and inter-relationships between different parameters can be hard to predict. As our problem solves a functional regression problem based on a relationship between a set of input functions, we intend to showcase the effectiveness of the proposed algorithm by predicting average dew-point temperature (F) across 12 weather stations based on their respective air temperatures (F). We consider 1 minute data of Wyoming ASOS data collected from IEM ASOS One Minute Data (Iowa Environmental Mesonet, 2022). The data has been collected for an interval of 2 hours for both input functions and output function from

Table 13: Graphs corresponding to learned $L$ for 25-node synthetic data. [Best viewed in color]

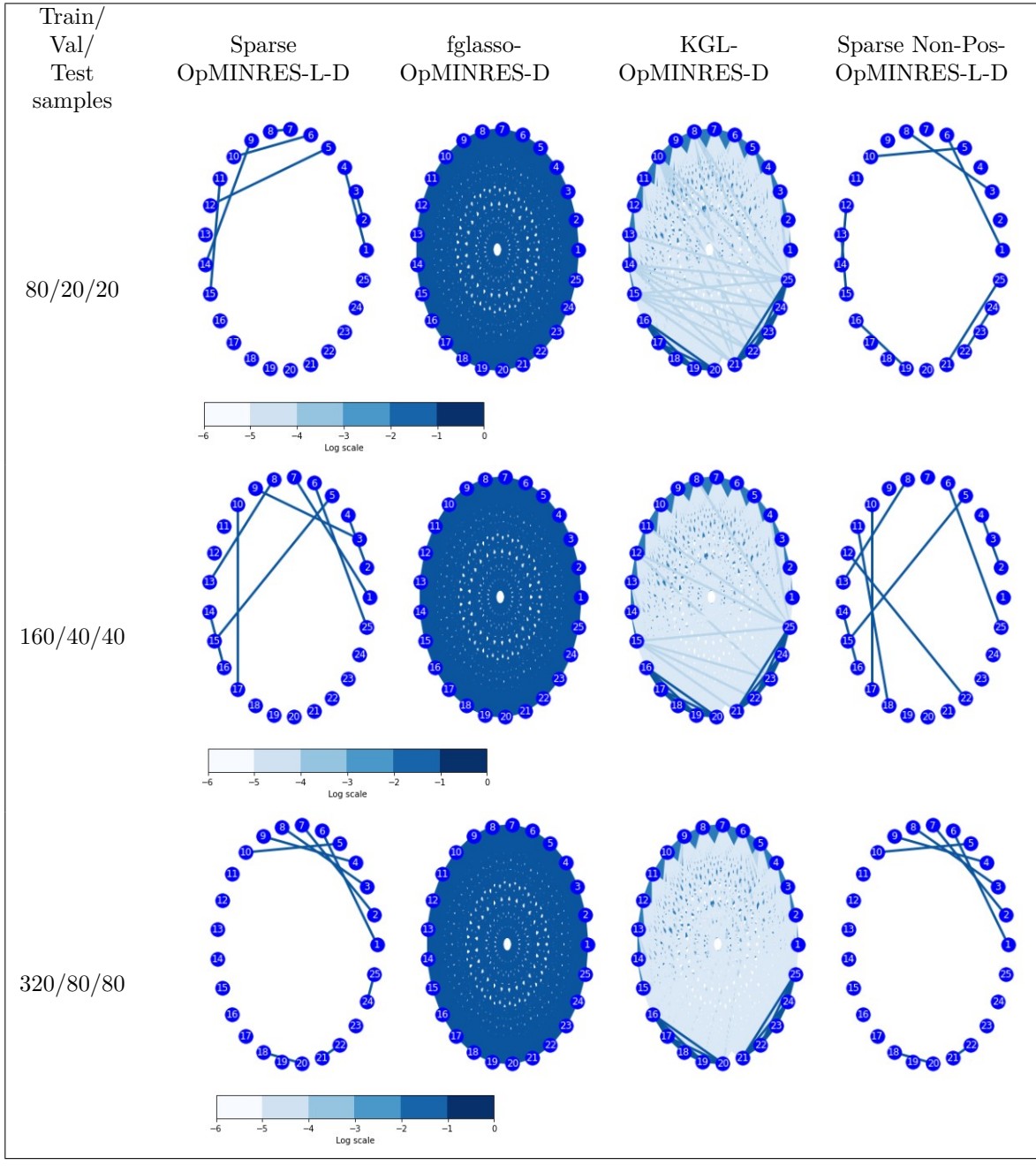

January, 2022 to August, 2022. Data collected at one minute interval for different 12 weather stations in Wyoming was pre-processed to create 2 hour interval data by disregarding intervals where data was missing in any of the 12 stations. A total of 718 samples have been collected after removing missing data. The following 12 weather stations in Wyoming have been considered: Big Piney (1), Casper/Natrona Intl (2), Cheyenne/Warren AFB (3), Gillette (4), Laramie/Gen. Brees (5), Lander/Hunt Field (6), Yellowstone (7), Riverton (8), Rawlins Municipal (9), Sheridan Co. Airport (10), Torrington Municipal Airport (11), Worland Municipal (12).

Table 14: Mean RSSE results for small weather data.

| Train/Val/Test samples | Methods | Mean RSSE | | |
|---|---|---|---|---|
| | | Train | Val | Test |
| 80/20/20 | Sparse OpMINRES-L-D | 0.004302 | 0.041949 | 0.092553 |
| | fglasso-OpMINRES-D | 0.001716 | 0.059419 | 0.082662 |
| | KGL-OpMINRES-D | 0.002357 | 0.049899 | 0.097951 |

For all the methods, a truncated trigonometric basis of $L^2([0, 1])$ with 80 basis functions has been considered for encoding the functional data. We segregate the weather data experiments into small weather data experiments by considering 120 samples and full weather data experiments. The following random data splits have been considered: (80/20/20) and (472/123/123), representing the number of training samples/validation samples/test samples in small weather data and full weather data settings, respectively. We discuss the results for the small weather data here. Note that results for full weather data related experimments are already presented in the main paper.

Initially, we use a small dataset with 120 samples drawn at random from the considered 8 months. Tables 14-15 showcase the performance of the algorithms for small weather data. Sparse OpMINRES-L-D performs the best in terms of mean RSSE on the test data compared to fglasso-OpMINRES-L-D and KGL-OpMINRES-L-D (Table 14). In Table 15, fglasso-OpMINRES-L-D and KGL-OpMINRES-L-D learn dense fully connected graphs which do not provide much information regarding the impact of different weather stations on the relationship of respective air temperature to the average dew point temperature. The plots illustrate location based relation between the 12 weather stations considered in Wyoming. Sparse OpMINRES-L-D learns a sparse $L$ where stations CYS(3) and TOR(11), GCC(4) and WRL(12), LND(6) and RIW(8) along with P60(7) and RWL(9) are connected. CYS(3) $(41.15564, -104.81047)$ and TOR(11) $(42.06472, -104.15278)$ are 114.89 km apart with an elevation of 1871 m and 1282 m, respectively. GCC(4) $(44.34892, -105.53936)$ and WRL(12) $(43.96571, -107.95083)$ are 197.54 km apart with an elevation of 1230 m and 1294 m, respectively. LND(6) $(42.81524, -108.72984)$ and RIW(8) $(43.06423, -108.45984)$ are 35.37 km apart with an elevation of 1694 m and 1688 m. P60(7) $(44.54444, -110.42111)$ and RWL(9) $(41.8056, -107.19994)$ are 401.40 km apart with an elevation of 2368 m and 2077 m. It can be observed that the connections in the learned graph structure have been established between stations with varying distances lying in close proximity elevation-wise (in 3 out of 4 cases) and latitude-wise.

### A.10.3 Experiments on NBA data

The movement of basketball and 21 players involved on the court (x-y coordinates) in the Atlanta Hawks (ATL) vs Utah Jazz (UTA) match on November 15, 2015 has been considered in this experiment. This data is available in the Github repo NBA Movement Data (Seward, 2018). The data has been collected for different plays for both input functions of 21 players and output function denoting the position of the ball, which includes missing data corresponding to some players in different plays. The data corresponding to the following players were used: Kyle Korver [ATL, G] (1), Thabo Sefolosha [ATL, G-F] (2), Paul Millsap [ATL, F] (3), Al Horford [ATL, C-F] (4), Tiago Splitter [ATL, F-C] (5), Derrick Favors [UTA, F-C] (6), Gordon Hayward [UTA, F] (7), Trevor Booker [UTA, F] (8), Alec Burks [UTA, G] (9), Shelvin Mack [ATL, G] (10), Kent Bazemore [ATL, F-G] (11), Chris Johnson [UTA, F] (12), Justin Holiday [ATL, G] (13), Dennis Schroder [ATL, G] (14), Jeff Withey [UTA, C] (15), Mike Muscala [ATL, F-C] (16), Rudy Gobert [UTA, C] (17), Trey Burke [UTA, G] (18), Raul Neto [UTA, G] (19), Rodney Hood [UTA, G] (20), Joe Ingles [UTA, F] (21), where the team, position and number assigned for the experiments has been provided. As plays in a basketball game are of different time duration, we use a truncated trigonometric basis of $L^2([0, 1])$ with 80 basis functions to sample the functions at fixed 100 points on $[0, 1]$. A total of 351 samples have been collected based on removing missing data. A random data split of (233/59/59) representing the number of training samples/validation samples/test samples has been considered. The problem requires solving a multi-dimensional functional regression problem which is incompatible with fglasso and KGL algorithms, as both fglasso & KGL are based on single dimensional input functions. Hence, we compare our method with the algorithm OpMINRES-D where a fixed $L$ is incorporated in our alternating minimization framework.

Table 15: Graphs corresponding to learned $L$ for small weather data. [Best viewed in color]

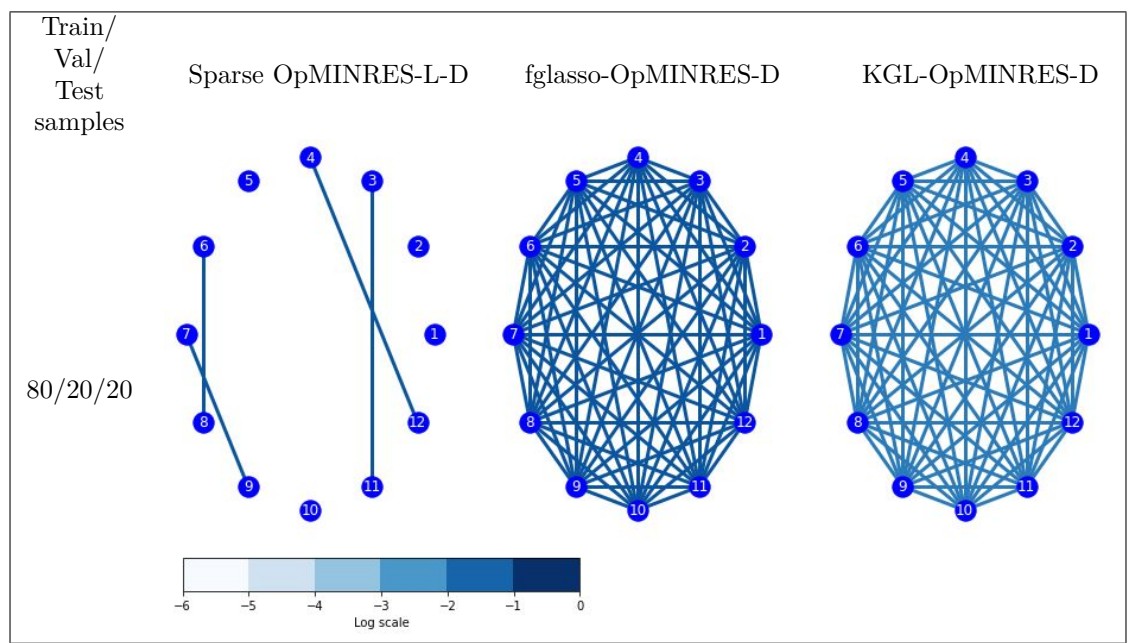

**OpMINRES-D**: A fixed $L$ is considered corresponding to a fully connected network of 21 nodes. This decision was made as fglasso mostly learns a fully connected graph in earlier experiments. Thus, a fixed $L$ (with no sparsity-inducing MCP) is used in the proposed alternating minimization regime for optimizing $\mathbf{u}$ and $D$. OpMINRES is used with $k_1(x, x'; G) = e^{-\gamma_x(x-x')^\top(L+D)(x-x')-\gamma_y(x-x')^\top(L+D)(x-x')}$ and $k_2^1(s, t) = e^{-\gamma_{op}^1|s-t|}, k_2^1(s, t) = e^{-\gamma_{op}^2|s-t|}$, where $\gamma_x, \gamma_y \in \{10^{-6}, 10^{-5}, 10^{-4}, 10^{-3}, 10^{-2}, 10^{-1}, 1, 10, 100\}$ and $\gamma_{op}^1, \gamma_{op}^2 \in \{10^{-6}, 10^{-5}, 10^{-4}, 10^{-3}, 10^{-2}, 10^{-1}, 1, 10, 100\}$.

**Sparse OpMINRES-L-D**: We consider the graph-induced operator-valued kernels using $k_1(x, x'; G) = e^{-\gamma_x(x-x')^\top(L+D)(x-x')-\gamma_y(x-x')^\top(L+D)(x-x')}$ and $k_2^1(s, t) = e^{-\gamma_{op}^1|s-t|}, k_2^2(s, t) = e^{-\gamma_{op}^2|s-t|}$, where $\gamma_x, \gamma_y \in \{10^{-6}, 10^{-5}, 10^{-4}, 10^{-3}, 10^{-2}, 10^{-1}, 1, 10, 100\}$ and $\gamma_{op}^1, \gamma_{op}^2 \in \{10^{-6}, 10^{-5}, 10^{-4}, 10^{-3}, 10^{-2}, 10^{-1}, 1, 10, 100\}$. Projected gradient descent is used in minimization with respect to $L$ and $D$ based on a decaying step-size. The sparsity is aided by the MCP regularization considered in learning of $L$.

Table 17: Mean RSSE results for NBA data.

| Train/Val/Test samples | Methods | Mean RSSE | | |
|---|---|---|---|---|
| | | Train | Val | Test |
| 233/59/59 | Sparse OpMINRES-L-D | 0.025200 | 0.087748 | 0.106344 |
| | OpMINRES-D | 0.023261 | 0.191459 | 0.265513 |

The results are illustrated in Tables 16-17. where comparison method OpMINRES-D uses a fully connected graph, however Sparse OpMINRES-L-D performs better with a sparse learned graph in terms of mean RSSE on the test data. The following major relations are obtained for the game based on graph structure corresponding to the learned $L$ in Table 16:

- Derrick Favors [UTA, F-C]—Trevor Booker [UTA, F] (6—8)

- Al Horford [ATL, C-F]—Gordon Hayward [UTA, F] (4—7)

- Alec Burks [UTA, G]—Trey Burke [UTA, G] (9—18)

Table 16: Graph corresponding to learned $L$ for NBA data. [Best viewed in color]

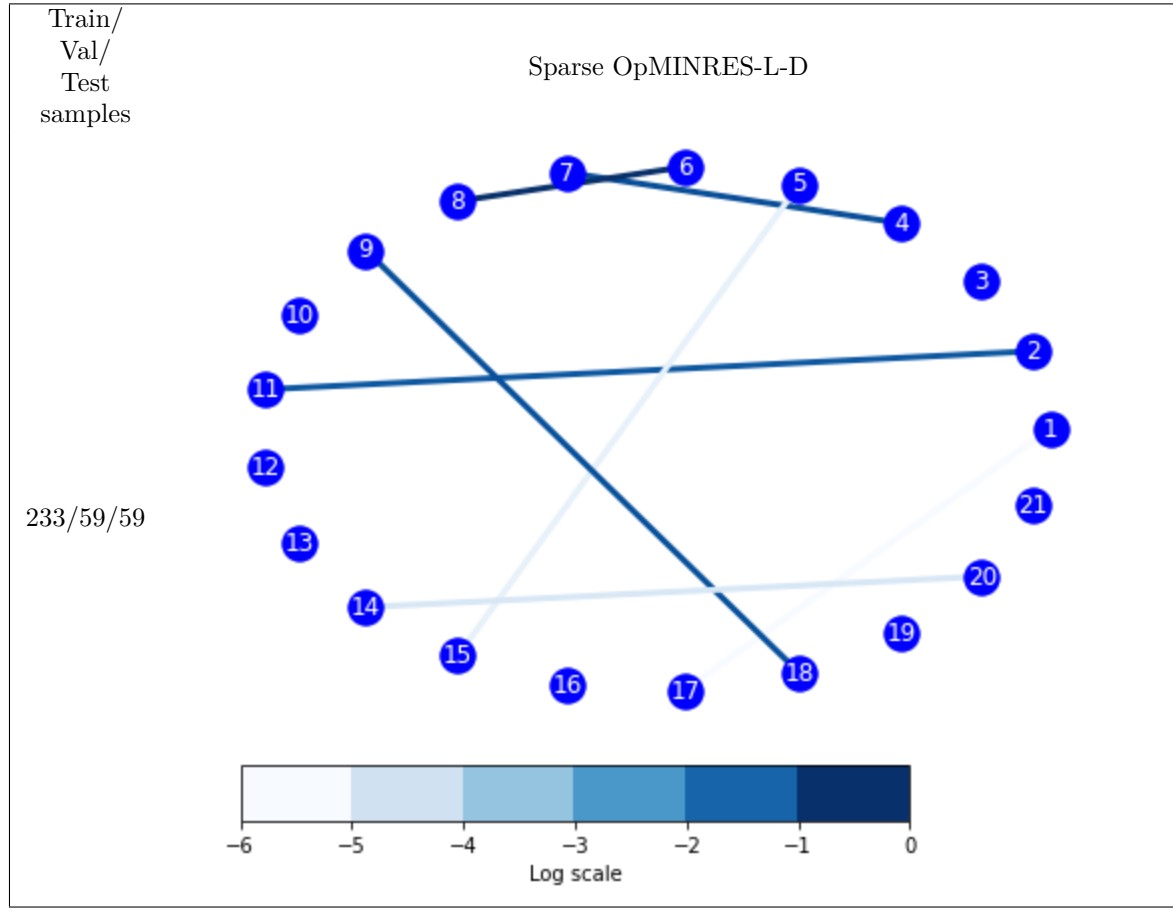

- Thabo Sefolosha [ATL, G-F]—Kent Bazemore [ATL, F-G] (2—11)

- Dennis Schroder [ATL, G]—Rodney Hood [UTA, G] (14—20)

- Tiago Splitter [ATL, F-C]—Jeff Withey [UTA, F-C] (5—15).

From the match report published in ESPN match recap and ESPN match scoreboard (ESPN, 2015a;b), it is clear that the relation Derrick Favors—Trevor Booker (6—8) had been pivotal in the win of Utah Jazz. The performance of Al Horford and Kent Bazemore for Atlanta Hawks was mentioned and captured by the relations (4—7) and (2—11). Though the partnership of Alec Burks—Trey Burke (9—18) for Utah Jazz is not evident in the match reports, their ball carrying interactions may be the reason for being learned in $L$.

### A.10.4 Hyperparameters for Experiments

In this section, we list the hyperparameters used for Sparse OpMINRES-L-D for different experiments illustrated in this work.

**Common hyperparameters:** $\lambda_{reg} = 0.5$, $\gamma_{reg} = 1$ (MCP), maxiter $= 1000$, tol $= 10^{-3}$ (OpMINRES). The decaying step-size regime for projected gradient descent in $L$ and $D$-based minimization involves starting with an initial step-size $10^{-4}$ and reducing it by a fixed factor 2 after a set of 5 iterations continuously till a final step-size (or learning rate) $10^{-9}$.

As our proposed approach Sparse OpMINRES-L-D aggregates many components, we provide some ablation studies to illustrate effectiveness in the next section.

Table 18: Hyperparameters used for experiments with Sparse OpMINRES-L-D on different data sets

| Experiment | No. of Training Samples | $\gamma_{op}$ | $\lambda$ | $\gamma$ | $\rho_L$ | $\rho_D$ | $m_{trace}$ |
|---|---|---|---|---|---|---|---|
| 3-node | 80 | 10 | $10^{-5}$ | $10^{-4}$ | $10^{-3}$ | 100 | 2 |
| | 160 | 10 | $10^{-4}$ | $10^{-2}$ | $10^{-3}$ | 10 | 2 |
| | 320 | 10 | $10^{-4}$ | $10^{-2}$ | $10^{-3}$ | 10 | 2 |
| 12-node | 80 | 1 | $10^{-5}$ | $10^{-6}$ | $10^{-2}$ | 1000 | 6 |
| | 160 | 10 | $10^{-6}$ | $10^{-6}$ | $10^{-2}$ | 10 | 6 |
| | 320 | 10 | $10^{-4}$ | $10^{-5}$ | $10^{-1}$ | $10^{-5}$ | 6 |
| 25-node | 80 | $10^{-1}$ | $10^{-3}$ | $10^{-4}$ | $10^{-1}$ | 10 | 13 |
| | 160 | 10 | $10^{-3}$ | $10^{-3}$ | $10^{-2}$ | 100 | 13 |
| | 320 | 100 | $10^{-2}$ | $10^{-2}$ | $10^{-1}$ | 100 | 13 |
| Weather Data | 80 | $10^{-3}$ | $10^{-2}$ | $10^{-1}$ | $10^{-4}$ | 1 | 6 |
| | 472 | 1 | $10^{-2}$ | $10^{-1}$ | $10^{-4}$ | $10^{-1}$ | 6 |
| NBA Data | 233 | $\gamma_{op}^1 = 100$ $\gamma_{op}^2 = 100$ | $10^{-2}$ | $\gamma_x = 0.5$ $\gamma_y = 0.5$ | $10^2$ | $10^2$ | 11 |

### A.10.5 Experiments for Ablation Studies

In order to understand the impact of different components of Sparse OpMINRES-L-D, we have run experiments for 12-node synthetic data by varying different hyperparameters and switching off different components in our approach. In order to enforce sparsity of learned graphs, we introduce $m_{trace}$ based constraint in Section 5.1.4. In Table 19, we tabulate the graphs corresponding to learned $L$ with Sparse OpMINRES-L-D for 12-node experiments using $m_{trace} \in \{0.1p, 0.25p, 0.5p, 0.75p, 0.9p\}$ with $p = 12$. From Table 19, we observe that most of the edges are being retained as $m_{trace}$ value is increased. The connections learned are illustrative of the generation process of synthetic data in Section A.10.1 as $m_{trace}$ is increased. Choice of $m_{trace}$ can be based on the error corresponding to the validation set as well as the desired number of connections to be learned since the number of edges increases with increase in $m_{trace}$. Table 20 illustrates that the performance with different $m_{trace}$ is comparable and a trade-off is expected when varying $m_{trace}$.

To illustrate the impact of choosing different kernels in our framework, we utilize different kernels as $k_2$ in (2) by utilizing the following:

- **ABS:** $k_2(s, t) = e^{-\gamma_{op}|s-t|}$

- **DIFFABS:** $k_2(s, t) = e^{-\gamma_{op_1}|s-t|} - e^{-\gamma_{op_2}|s-t|}$

- **RBF:** $k_2(s, t) = e^{-\gamma_{op_1}|s-t|^2}$

- **EPAN:** $k_2(s, t) = \max(0, 1 - \gamma_{op}|s-t|^2)$,

with $\gamma_{op}, \gamma_{op_1}, \gamma_{op_2} \in \{10^{-6}, 10^{-5}, 10^{-4}, 10^{-3}, 10^{-2}, 10^{-1}, 1, 10, 100\}$. Note that we use $k_1$ as defined in (108) which incorporates $L$ and $D$ of the learned graph. Table 21 showcases comparable performance of different kernels as $k_2$, but Table 22 illustrates the graphs learned different connections for ABS, DIFFABS. We see that RBF and EPAN kernel choices learn different graphs which contain some useful connections which can be related to the generation of the synthetic data. It can be interpreted that most of the connections learned by ABS, DIFFABS and RBF kernels better represent the generation of 12-node synthetic data.

Next, we study the impact of the $m_{trace}$ constraint in the sparsity regularization and the complete sparsity regularization framework as proposed in Section 5.1.4. Table 23 illustrates comparable performance based on mean RSSE error where $m_{trace}$ constraint is removed and $L$ and $D$-based regularization is removed by setting $\rho_L$ and $\rho_D$ as 0. Similarly, Table 24 illustrates comparable performance when MCP regularization is

Table 19: Graph corresponding to learned $L$ by considering $m_{trace}$ in $\{0.1p, 0.25p, 0.5p, 0.75p, 0.9p\}$ in Sparse OpMINRES-L-D for 12-node experiments ($p = 12$). [Best viewed in color]

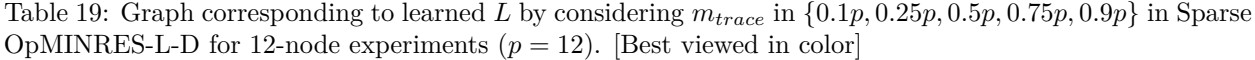

completely removed and $L$ and $D$-based regularization is removed by setting $\rho_L$ and $\rho_D$ as 0. Although the performance is comparable in terms of mean RSSE error, however we note that Tables 25 and 26 illustrate the failure to learn meaningful graphs since most of the graphs have connections with equal weights providing no relevant information. Without the $m_{trace}$ constraint, Table 25 showcases negligible weights being assigned to each connection, while without the complete MCP regularization framework Table 26 showcases uniformly distributed weights across fully connected graphs.

Table 20: Mean RSSE results for $m_{trace}$ in $\{0.1p, 0.25p, 0.5p, 0.75p, 0.9p\}$ using Sparse OpMINRES-L-D in 12-node synthetic data ($p = 12$).

| Train/Val/Test samples | $m_{trace}$ in Sparse OpMINRES-L-D | Mean RSSE | | |
|---|---|---|---|---|
| | | Train | Val | Test |
| 80/20/20 | $m_{trace} = 0.1p$ | 1.250593 | 1.727192 | 1.532670 |
| | $m_{trace} = 0.25p$ | 1.234657 | 1.735397 | 1.587149 |
| | $m_{trace} = 0.5p$ | 1.140691 | 1.735397 | 1.583640 |
| | $m_{trace} = 0.75p$ | 1.092453 | 1.904231 | 1.636385 |
| | $m_{trace} = 0.9p$ | 1.075372 | 1.916426 | 1.640379 |
| 160/40/40 | $m_{trace} = 0.1p$ | 0.873584 | 1.291735 | 1.380035 |
| | $m_{trace} = 0.25p$ | 0.886945 | 1.263236 | 1.368678 |
| | $m_{trace} = 0.5p$ | 0.888574 | 1.229568 | 1.385952 |
| | $m_{trace} = 0.75p$ | 0.849892 | 1.265577 | 1.412620 |
| | $m_{trace} = 0.9p$ | 0.831497 | 1.284316 | 1.425376 |
| 320/80/80 | $m_{trace} = 0.1p$ | 1.073370 | 1.291374 | 1.237471 |
| | $m_{trace} = 0.25p$ | 1.071354 | 1.295419 | 1.238216 |
| | $m_{trace} = 0.5p$ | 1.062102 | 1.294110 | 1.239181 |
| | $m_{trace} = 0.75p$ | 1.056678 | 1.295300 | 1.242216 |
| | $m_{trace} = 0.9p$ | 1.053001 | 1.296809 | 1.241735 |

Table 21: Mean RSSE results for different kernels as $k_2$ with 12-node synthetic data.

| Train/Val/Test samples | $k_2$ in Sparse OpMINRES-L-D | Mean RSSE | | |
|---|---|---|---|---|
| | | Train | Val | Test |
| 80/20/20 | ABS | 1.140691 | 1.780445 | 1.583640 |
| | DIFFABS | 1.167264 | 1.806093 | 1.618175 |
| | RBF | 1.111369 | 2.092855 | 1.754655 |
| | EPAN | 0.759214 | 2.035527 | 1.838925 |
| 160/40/40 | ABS | 0.888574 | 1.229568 | 1.385952 |
| | DIFFABS | 1.154356 | 1.362239 | 1.417921 |
| | RBF | 0.846649 | 1.236121 | 1.428687 |
| | EPAN | 0.906656 | 1.260602 | 3.626625 |
| 320/80/80 | ABS | 1.062102 | 1.294110 | 1.239181 |
| | DIFFABS | 1.073292 | 1.295140 | 1.243346 |
| | RBF | 0.931760 | 1.330628 | 1.283005 |
| | EPAN | 1.026490 | 1.304147 | 1.247868 |

Further, we perform experiments where we utilize a lasso based regularization to induce sparsity instead of depending upon the proposed sparsity inducing framework. We introduce $\rho_D \sum_{i=1}^{p} |D_{ii}|, \rho_D > 0$ in (3) instead of $\rho_L \sum_{i=1}^{n} x^{(i)^\top} L x^{(i)} + \rho_D \|D\|_F^2$ and ignore the proposed MCP regularization framework. Table 28 illustrates comparable performance in terms of mean RSSE error but Table 27 showcases the failure to distinguish meaningful interactions in the learned graphs.

Table 22: Graph corresponding to learned $L$ by considering different kernels for $k_2$ in Sparse OpMINRES-L-D for 12-node experiments. [Best viewed in color]

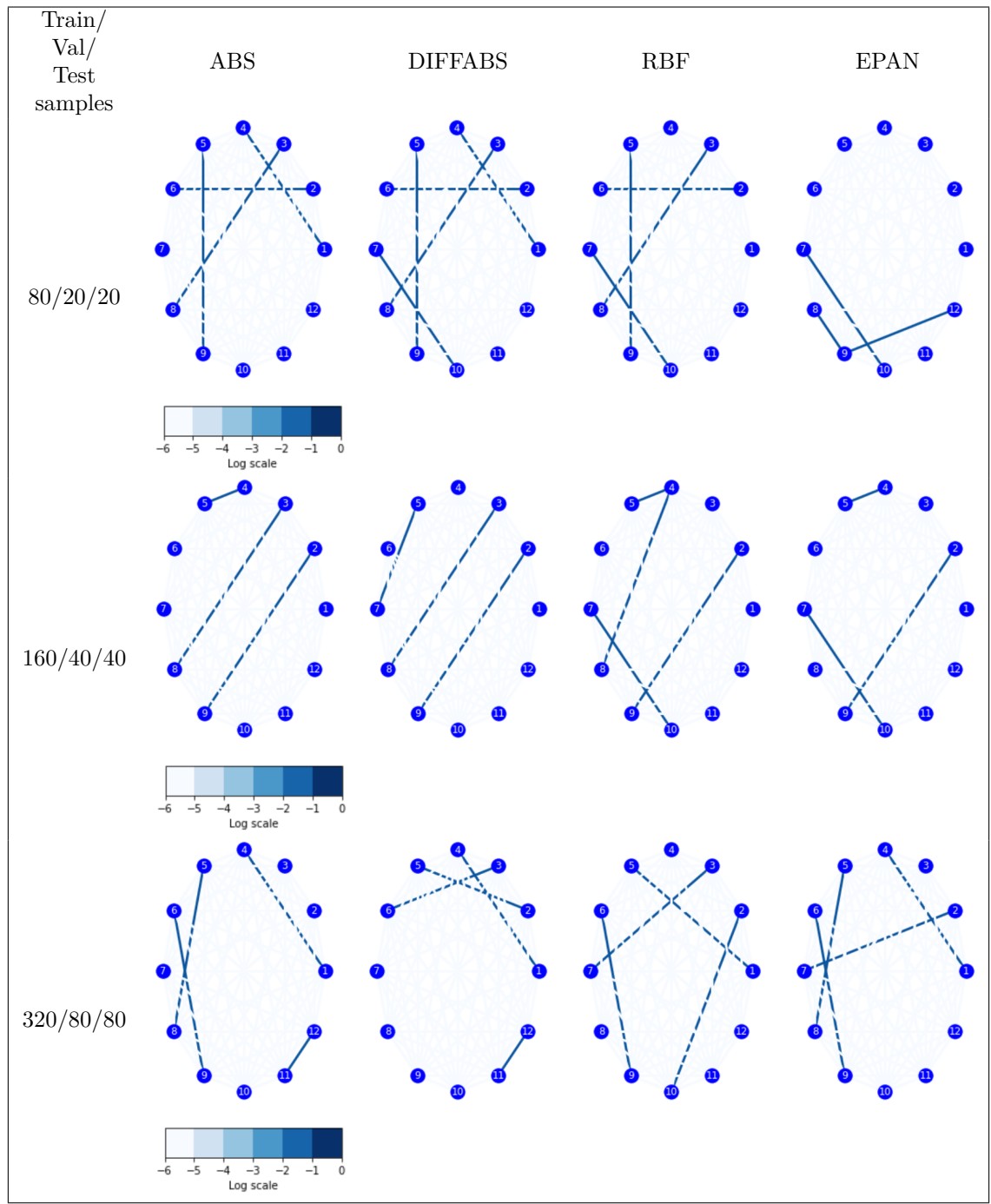

Table 23: Mean RSSE results for 12-node synthetic data using Sparse OpMINRES-L-D with no $m_{trace}$ constraint and ablation of regularization with respect to $L$ and $D$.

| Train/Val/ Test samples | Methods | Mean RSSE | | |
|---|---|---|---|---|
| | | Train | Val | Test |
| 80/20/20 | Sparse OpMINRES-L-D | 1.140691 | 1.780445 | 1.583640 |
| | Sparse OpMINRES-L-D with no $m_{trace}$ | 1.274110 | 1.729524 | 1.530376 |
| | Sparse OpMINRES-L-D with no $m_{trace}$ and $\rho_L = 0$ | 1.273360 | 1.729269 | 1.530041 |
| | Sparse OpMINRES-L-D with no $m_{trace}$ and $\rho_D = 0$ | 0.790852 | 1.891594 | 1.407203 |
| 160/40/40 | Sparse OpMINRES-L-D | 0.888574 | 1.229568 | 1.385952 |
| | Sparse OpMINRES-L-D with no $m_{trace}$ | 0.896321 | 1.229568 | 1.371430 |
| | Sparse OpMINRES-L-D with no $m_{trace}$ and $\rho_L = 0$ | 0.896311 | 1.280351 | 1.371432 |
| | Sparse OpMINRES-L-D with no $m_{trace}$ and $\rho_D = 0$ | 0.833068 | 1.291457 | 1.410686 |
| 320/80/80 | Sparse OpMINRES-L-D | 1.062102 | 1.294110 | 1.239181 |
| | Sparse OpMINRES-L-D with no $m_{trace}$ | 1.078058 | 1.291549 | 1.237178 |
| | Sparse OpMINRES-L-D with no $m_{trace}$ and $\rho_L = 0$ | 1.078026 | 1.291574 | 1.237129 |
| | Sparse OpMINRES-L-D with no $m_{trace}$ and $\rho_D = 0$ | 1.078058 | 1.291550 | 1.237179 |

Table 24: Mean RSSE results for 12-node synthetic data using Sparse OpMINRES-L-D without MCP regularization and ablation of regularization with respect to $L$ and $D$.

| Train/Val/ Test samples | Methods | Mean RSSE | | |
|---|---|---|---|---|
| | | Train | Val | Test |
| 80/20/20 | Sparse OpMINRES-L-D | 1.140691 | 1.780445 | 1.583640 |
| | Sparse OpMINRES-L-D with no MCP | 0.911711 | 1.893550 | 1.481332 |
| | Sparse OpMINRES-L-D with no MCP and $\rho_L = 0$ | 0.911711 | 1.893550 | 1.481332 |
| | Sparse OpMINRES-L-D with no MCP and $\rho_D = 0$ | 0.755296 | 1.892366 | 1.399945 |
| 160/40/40 | Sparse OpMINRES-L-D | 0.888574 | 1.229568 | 1.385952 |
| | Sparse OpMINRES-L-D with no MCP | 0.838759 | 1.282838 | 1.397084 |
| | Sparse OpMINRES-L-D with no MCP and $\rho_L = 0$ | 0.838759 | 1.282838 | 1.397084 |
| | Sparse OpMINRES-L-D with no MCP and $\rho_D = 0$ | 0.793097 | 1.289822 | 1.429700 |
| 320/80/80 | Sparse OpMINRES-L-D | 1.062102 | 1.294110 | 1.239181 |
| | Sparse OpMINRES-L-D with no MCP | 1.062727 | 1.298913 | 1.240176 |
| | Sparse OpMINRES-L-D with no MCP and $\rho_L = 0$ | 1.062727 | 1.298913 | 1.240176 |
| | Sparse OpMINRES-L-D with no MCP and $\rho_D = 0$ | 1.062727 | 1.298926 | 1.240208 |

Table 25: Graph corresponding to learned $L$ by no $m_{trace}$ constraint and controlling $\rho_L$ in $L$-based regularization and $\rho_D$ in $D$-based regularization for 12-node experiments. [Best viewed in color]

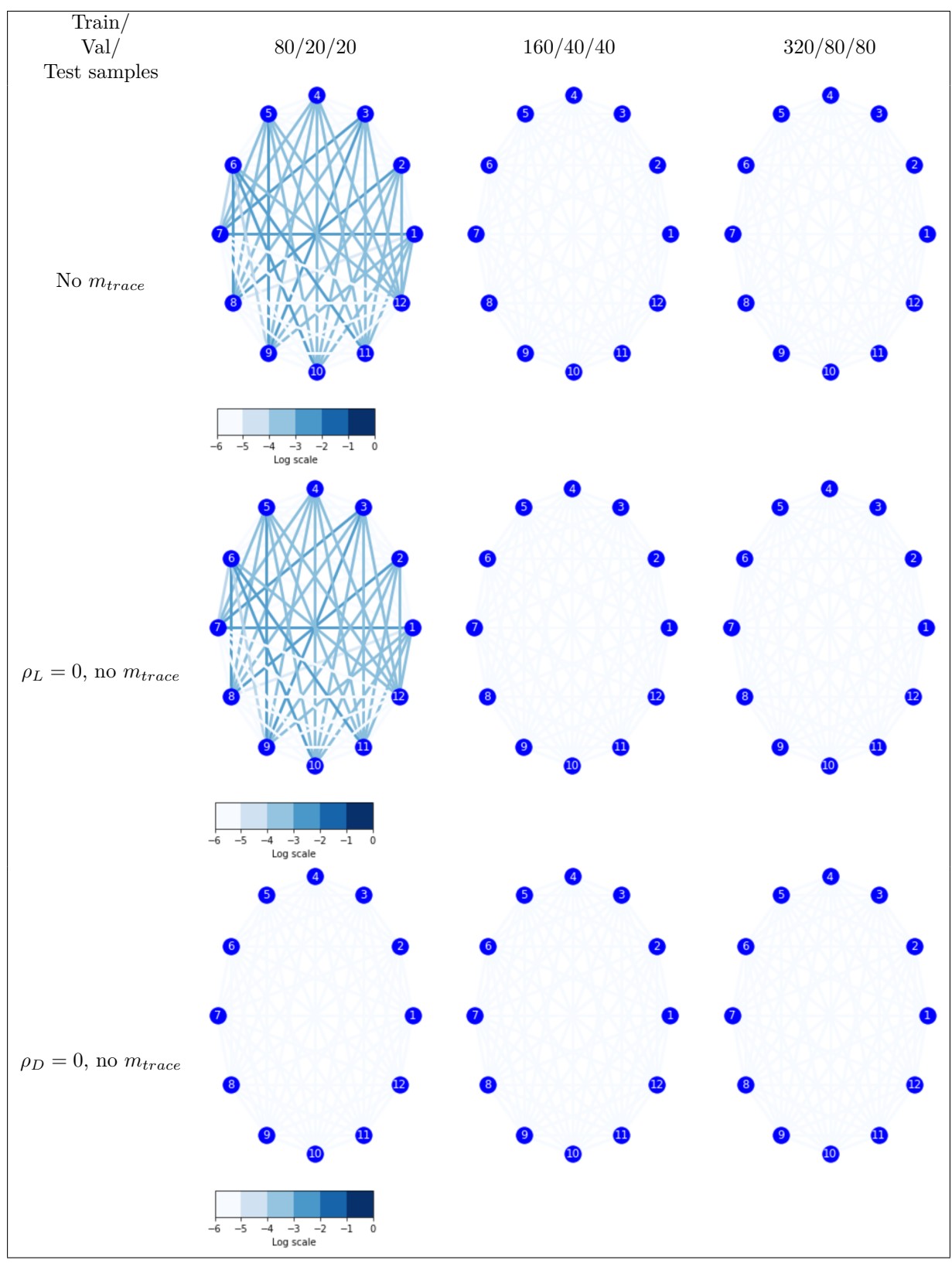

Table 26: Graph corresponding to learned $L$ by switching off the MCP-based regularization and controlling $\rho_L$ in $L$-based regularization and $\rho_D$ in $D$-based regularization for 12-node experiments. [Best viewed in color]

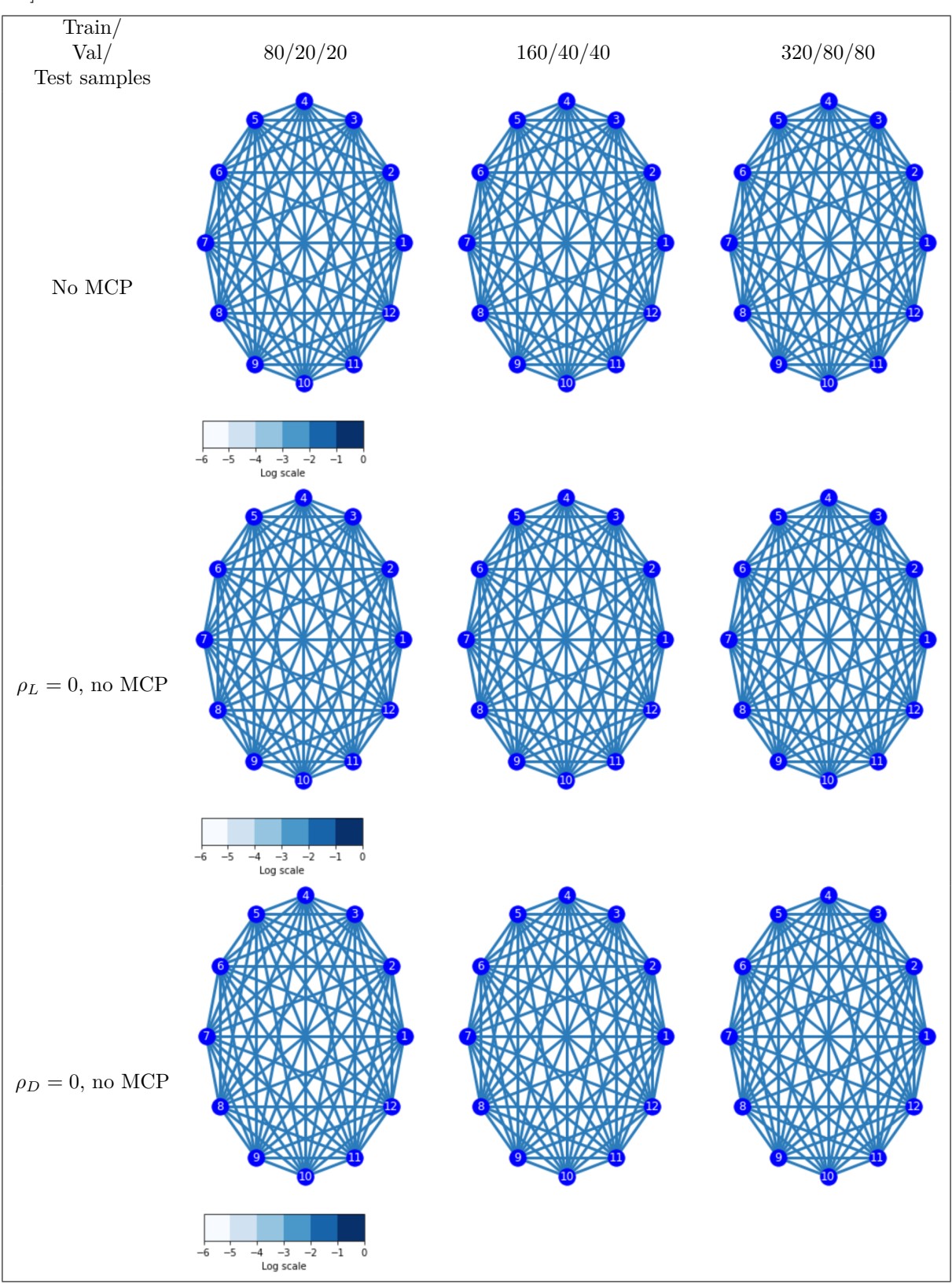

Table 27: Graph corresponding to learned $L$ by using only D-lasso based regularization without MCP-based sparsity inducing regularization instead of Sparse OpMINRES-L-D for 12-node experiments. [Best viewed in color]

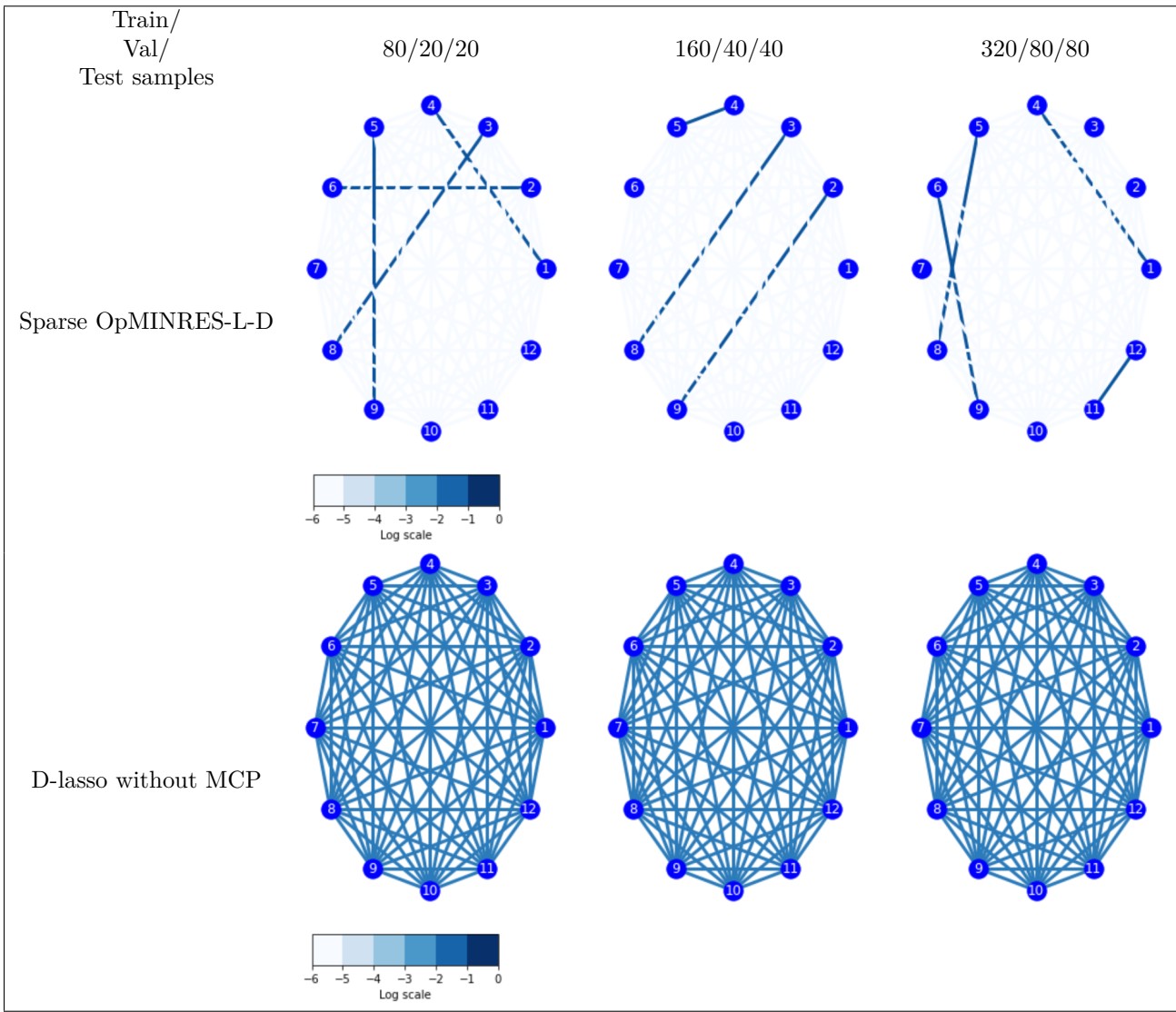

Table 28: Mean RSSE results for 12-node synthetic data using Sparse OpMINRES-L-D without MCP regularization and lasso regularization for $D$.

| Train/Val/ Test samples | Methods | Mean RSSE | | |
|---|---|---|---|---|
| | | Train | Val | Test |
| 80/20/20 | Sparse OpMINRES-L-D | 1.140691 | 1.780445 | 1.583640 |
| | Sparse OpMINRES-L-D with no MCP and D lasso | 0.745433 | 1.891283 | 1.402357 |
| 160/40/40 | Sparse OpMINRES-L-D | 0.888574 | 1.229568 | 1.385952 |
| | Sparse OpMINRES-L-D with no MCP and D lasso | 0.821067 | 1.284889 | 1.409910 |
| 320/80/80 | Sparse OpMINRES-L-D | 1.062102 | 1.294110 | 1.239181 |
| | Sparse OpMINRES-L-D with no MCP and D lasso | 1.073490 | 1.297428 | 1.241109 |

