# OpenReview forum: "Learning Sparse Graphs for Functional Regression using Graph-induced Operator-valued Kernels"
_TMLR — Accepted by TMLR_

### Review · Reviewer_kJos · 2023-12-06

**Summary Of Contributions:**

The paper outlines a novel approach to functional regression problems, where the objective is to map multiple input functions to a single output function. The key contributions and methodologies presented in the paper are as follows:

The authors introduced a graph-induced Operator-Valued Kernel (OVK) to solve functional regression problems, leveraging a graphical structure to represent interactions among input functions, especially a sparsity-inducing regularization on the Laplacian matrix. This framework assists in learning the regression map along with the graphical structure itself. The authors raised an alternating minimization framework for learning the map and the matrices representing the graphical structure through the Operator-based Minimum Residual (OpMINRES) algorithm and projected gradient descent and they extended the framework to address multi-dimensional functional regression problems. They also studied the generalization error bounds for the learned functional regression map, which also takes into account the learning of the graph-induced OVK.

**Audience:**

Yes

**Broader Impact Concerns:**

Not very applicable here.

**Claims And Evidence:**

Yes

**Requested Changes:**

1. The notation is (4) and below are confusing. Does this inner product stand for $x^\top Lx'=\sum_{i,j} \int_0^1x_i(t) L_{ij} x_j'(t)dt$?
2. Proposition 5.1: typo $L([0,1])$.
3. Typo in (9).
4. If we consider a graph $G$ without self-loops, then $L= D-A$ will determine $D$. In this case, we do not need regularization on $D$ in (13), right? This should simplify the algorithm. In the results of the experiments, I did not see self-loops for all vertices. So why do we need regularization on $D$?
5. At the top of page 10, the authors mentioned a different regularization from Humbert et al., 2021. Is this smoothness term better than (13)? There should be more explanations here.
6. In  (22), using upper indices to indicate the number of iterations is confusing with the powers of the quantities.
7. In Section 5.2.4, you used the trace of Laplacian for sparsity-induced regularization, but why not use $\ell_1$ minimization of $D$ for regularization? Besides, any insight into why you chose $m_{trace}=p/2$. The choice of $m_{trace}$ indicates the graph is extremely sparse but this hyperparameter should depend on the problem and datasets, which can be estimated in other ways.
8. Please specify the hyperparameters in Algorithms 1 and 2.
9. Typo above (73): $X^m$ should be $\mathcal{X}^m$.
10. In Section 7, you used KPL-OpMINRES-D in many places which should be KGL-OpMINRES-D.
11. How do you explain the results in Section 7.2 on weather data? Sparse OpMINRES-L-D provided a sparse graph but should it be better if the graph has more edges for places close to each other?
12. In A.9, you repeated the details of the experiments stated in Section 7, which is a redundancy. Maybe you should provide more details such as the learning rates, regularization parameters, and $m_{trace}$.

**Strengths And Weaknesses:**

## Upsides:
1. In general, this paper is well-written. Background information and relevant literature are adequately covered. Technical developments of the work are rigorous, intuitive, and well-founded from existing works. The mathematical quality of the work is good.
2. The authors provided extensive examples and experiments for functional regression models with graph structures. This empirically shows the improvement of the authors' methods in some spatial-temporal functional regression problems.

## Downsides:
1. There is a lack of proof for the convergence of alternating minimization of a non-convex function $J$ proposed in this paper. The choice of hyperparameters, e.g. learning rates, in OpMINRES is not explicitly discussed in the paper to ensure convergence.
2. Although the authors provided a generalization analysis, there is a lack of discussion on how the graph-induced kernel regression and sparsity of the graph change the generalization error bound for the regression problem. The interpretation of Theorem 6.3 and (125) should be expanded in the paper.
3. In the kernel regression part, the authors chose specific kernels $k_1$ and $k_2$ in (2) without enough explanations on the choice of the kernels. There should be either empirical or theoretical comparisons with different kernels on this problem.

---

> ### Author Response · Authors · 2024-01-17
> **Response to Reviewer kJos**
>
> Downsides:
> 1. As mentioned in Section 5.1.5, the objective function $J$ is not just noncovex, but is also dependent on a collection of heterogeneous variables where $u_i$'s$\in\mathcal{Y}$ and $\text{vech}(L),\text{vech}(D)\in\mathbb{R}^{\frac{p(p+1)}{2}}$. It is not clear to us how to prove theoretical guarantees for alternating minimization of such objective functions. We leave this for a future work.
> $\hspace{4cm}$ The hyperparameters for OpMINRES used in the experiments are the same as mentioned in [2] and we have included these details in Appendix A.10.4 of the revised version. Other hyperparameters used in our approach have also been mentioned in Appendix A.10.4.
>
> 2. Theorem 6.3 provides a probabilistic upper bound for the sample error in terms of Rademacher average for the class of graph-induced operator-valued kernels. Equation (101) in the revised version provides an upper bound on the generalization error with the help of a bound on the Rademacher average. We have included these explanations in the revised version. Though we have considered a weaker bound on the generalization error without considering sparsity regularization for the graph-induced OVKs, a tighter bound may be obtained by constraining the class of graph-induced OVKs based on MCP regularization with additional assumptions, which we will be considering in a future work.
>
> 3. In OVK based approaches ([1], [2]) for functional regression, different kernels used as $k_1$ and $k_2$ produced similar/comparable results. This encouraged us to proceed with a specific choice for just $k_2$ since we have already motivated the use of graph-induced kernel as $k_1$ in Definition 5.1 in the revised version. To investigate the impact of choosing different $k_2$, we considered multiple choices of $k_2$ and give results in Tables 21-22 in Appendix A.10.5.
>
> Requested changes:
> 1. Yes, it is equivalent. In the revised version, we have explicitly included this notation in the first paragraph of Page 7 as well as in equations (105) and (106) in Appendix A.1 in the revised version.
> 2. Corrected typos.
> 3. Corrected typos.
> 4. Considering a graph $G$ without self-loops, the Laplacian matrix $L$ is given by $L=\mathbb{D}-W$, where $\mathbb{D}$ is the degree matrix and $W$ is the weight matrix of $G$. The main goal of our work is to learn the graph $G$ for the input functional variables determining the output functional variable. Please note that the $D$ considered for regularization is a diagonal matrix with non-negative entries. In the unknown graph structure setting, $D$ is used as a perturbation matrix of $L$ to aid the performance in functional regression task and is different from the degree matrix $\mathbb{D}$. This has been explained in the paragraph following Definition 5.1 in Section 5.1 in the revised version.
> 5. The regularizer for $L$ inspired from the regularizer considered in (Humbert et al., 2021) is different from the traditional Frobenius norm based regularizer, since $\sum_{i=1}^{n}{x^{(i)}}^\top Lx^{(i)}$ incorporates the variability corresponding to functional data $x^{(i)}$, for $i=1,2,\dots,n$.
> 6. Due to notational scarcity, we have clearly defined the notation used in relevant sections and believe that the context of the notation makes it clear. In that spirit, we have used the  notation of using upper indices to denote the number of iterations.
> 7. We have included experiments for the 12-node setting in Appendix A.10.5, which showcase the use of $\ell_1$ minimization of $D$ for regularization instead of MCP-based regularization of $L$. This produces comparable results in terms of validation error but the plots learned result in a fully connected graph which fails to distinguish any meaningful interactions.
> $\hspace{2.5cm}$ We acknowledge the fact that choice for $m_{trace}$ is highly dependent on the problem and the dataset. In order to illustrate the impact of different $m_{trace}$, we have included an ablation study in Appendix A.10.5.
> 8. Specified in Appendix A.10.4 of the revised version.
> 9. Please note that $X^m$ above (63) in the revised version denotes the standard definition of Rademacher average with the input space being a space of scalars/vectors, which we have further extended to a functional setting in Definition 6.1.
> 10. Typos corrected.
> 11. From the plots it can be observed that the learned graph establishes edges between places with varying elevations and ones lying in close proximity latitude-wise.
> 12. We have removed the redundant sections from A.10 (in the revised version) and have included all the hyperparameter details in Appendix A.10.4 of the revised version.
>
> References:
>
> [1] Hachem Kadri, Emmanuel Duflos, Philippe Preux,  Stéphane Canu, Alain Rakotomamonjy, and Julien Audiffren. Operator-valued kernels for learning from functional response data. JMLR, 17(20):1–54, 2016.
>
>
> [2] Akash Saha and Balamurugan Palaniappan. Learning with operator-valued kernels in reproducing kernel krein spaces. NeurIPS, 2020.

---

### Review · Reviewer_mn8m · 2023-12-27

**Summary Of Contributions:**

This research focuses on the task of function-to-function regression, which involves learning a mapping from a set of input functions to an output function. The authors propose an innovative approach using a graph-induced operator-valued kernel to capture the relationships among the input functions. Even in cases where the underlying graphical structure is unknown, they present a method to learn an appropriate Laplacian matrix, enabling a comprehensive understanding of the connections. To tackle this learning problem, the authors develop an alternating minimization framework. This framework incorporates a min-max concave penalty that encourages meaningful interactions within the graphical structure, leading to more reliable results. The framework is also designed to handle complex scenarios involving multi-dimensional input and output functions. In addition, they introduce an efficient sample-based approximation algorithm, allowing for efficient processing of large datasets. The research also provides valuable insights into the generalization performance of the learned mapping. The authors establish bounds on the generalization error of the proposed method. Furthermore, through extensive empirical evaluations on both synthetic and real datasets, they demonstrate the effectiveness of their learning framework.

**Audience:**

Yes

**Claims And Evidence:**

Yes

**Requested Changes:**

1.The numerical experiments involve only several hundred training examples, which leaves uncertainty regarding the assessment of scalability.

2.The numerical experiments do not provide clear evidence on whether the proposed nonlinear model outperforms the linear model presented in (Gómez et al., 2021).

3. It is a little bit strange that the convergence rate $m^{-\eta/8(1+\eta)}$ in (125) is independent of the covering number index $q$ in (118). Moreover, it is strong to assume that $F\in\mathcal{H}_{\mathcal{K}}$ in Theorem 6.3.

4. Any special reason to use MCP instead of LASSO?

**Strengths And Weaknesses:**

Strength:
1. A strength of this research lies in its ability to incorporate the learning of an appropriate graphical structure within the context of functional regression.
2. The incorporation of a sparsity-inducing regularizer in the Laplacian matrix enhances interpretability by highlighting the most relevant interactions among input functions. Additionally, the proposed sample-based approximation algorithm improves scalability, enabling efficient processing of the data.
3. The research establishes a generalization error bound for the proposed approach.

Weakness:
1. The research does not extensively compare the proposed method with existing approaches in functional regression, e.g. functional PCA based approach or neural networks based approach.
2. The theoretical results presented in the paper focus on the simplest case, without considering the incorporation of a sparsity-inducing regularizer or the use of a sample-based approximation algorithm.

---

> ### Author Response · Authors · 2024-01-17
> **Response to Reviewer mn8m**
>
> Weaknesses:
> 1. FPCA useful for encoding functional data was not utilized in our work since we opted for OpMINRES algorithm which is suited for a pre-specified set of basis functions. However, we can take up FPCA in OpMINRES as a future work. Please note that a linear functional model has been proposed in ([6]) and it uses FPCA to learn response functions from a predictor function. However the model in ([6]) does not incorporate multiple input functions and their graphical structure in order to learn the output function. Hence we did not do a direct comparison with the method in ([6]).
> $\hspace{5cm}$ Neural network based approaches for functional regression task in ([1]) are effective in a larger data setting than what we have considered in our experiments. Even then the approach in [1] results in solving the functional regression task without any graph structure being learned, which forced us to not go for a direct comparison with our proposed approach.
>
> 2. Please note that by focusing on the simplest case, we have obtained a weaker bound on the generalization error without considering sparsity regularization for the graph-induced OVKs, a tighter bound may be obtained by constraining the class of graph-induced OVKs based on MCP regularization with additional assumptions which we will consider in a future work.
> $\hspace{2cm}$The proposed sample-based approximation algorithm is a heurestic for incorporating larger datasets and it is not clear to us how it can be incorporated in deriving generalization bounds.
>
> Requested Changes:
> 1. Though the numerical experiments involve only several hundred training examples, each example involves around a hundred discretizations. Recent approaches in scalability of scalar-valued kernel approaches ([2]) bank on Nyström approximations and powerful computation hardware, eg. GPUs. Developing an analogous extension for functional data in the case of OVKs is not very clear to us. Our focus in this work is concerning practical applications where large functional datasets may not be readily available. The proposed sample-based approximation algorithm to aid in scalability showcases that even fewer than 100 samples of such datasets produces comparable results.
>
> 2. The linear model in (Gómez et al., 2021) performs root-cause analysis and diagnosis for functional directed graphical models. A major difference between Gómez et al., and our approach is that Gómez et al.,  considers a directed graph where output functional variable is also a part of the graphical structure along with other input functional variables, whereas in our approach  the output functional variable is not considered to be part of the graphical structure, rather the output function is predicted with the help of a graphical structure imposed only on the input functional variables. This restricts us from having a direct comparison of Gomez et al.'s approach with our setting. We have updated the related work section to differentiate our approach from that of Gómez et al., 2021.
>
> 3. (a) Though, the covering number index $q$ is absent in $m^{-\eta/8(1+\eta)}$, $q$ is present in the multiplicative term of $m^{-\eta/8(1+\eta)}$. A similar approach in [3] showcases results for convergence rate being independent of covering number index which we have extended to a functional regression problem.
>  (b) We agree that the assumption of $F\in\mathcal{H}_K$ in Theorem 6.3 may be strong in general, but this assumption plays a pivotal role in deriving the results required for establishing Theorem 6.3. A similar assumption can be found in ([3]).
>
> 4. LASSO is a popular candidate for inducing sparsity in graph learning but recent works ([4], [5]) prove that a large regularization parameter in LASSO will lead to a solution representing a complete graph, i.e., every pair of vertices is connected by an edge which is not desired. Hence, we used MCP as a nonconvex sparsity-inducing regularizer.
>
> Ref:
>
> [1] Aniruddha Rajendra Rao and Matthew Reimherr. Modern non-linear function-on-function regression. Statistics and Computing, 33(6):1–12, 2023.
>
> [2] Giacomo Meanti, Luigi Carratino, Lorenzo Rosasco, and Alessandro Rudi. Kernel methods through the roof: handling billions of points efficiently. NeurIPS, 2020.
>
> [3] Charles A. Micchelli, Massimiliano Pontil, Qiang Wu, and Ding-Xuan Zhou. Error bounds for learning the kernel. Analysis and Applications, 14(06):849–868, 2016.
>
> [4] Jiaxi Ying, José  Vinìcius de Miranda Cardoso, and Daniel Palomar. Nonconvex sparse graph learning under laplacian constrained graphical model. NeurIPS, 2020.
>
> [5] Jiaxi Ying, José  Vinìcius de Miranda Cardoso, and Daniel Palomar. Minimax Estimation of Laplacian Constrained Precision Matrices. AISTATS 2021.
>
> [6] Harjit Hullait, David S. Leslie, Nicos G. Pavlidis, and Steve King. Robust function-on-function regression. Technometrics, 63(3):396–409, 2021.

---

### Review · Reviewer_VaBc · 2024-01-03

**Summary Of Contributions:**

In this paper, the authors propose a approach to learn relational structures given multiple input variables. These variables may be features, or functions of input-output maps, etc. There is a label associated with reach realization of the inputs. The goal is to learn the mapping from the input variables to the output label.

The approach here is based on kernel methods. This kernel operates on the multiple input variables, with possible extensions such as adding regularization to the weights of the kernel map.

One of the main idea of this paper is to add a Laplacian matrix on top of this kernel. This Laplacian matrix takes an input graph, and then uses the Laplacian matrix product to associate the input variables.

- In the setting where the graph is given, learning the kernel is straightforward, similar to standard kernel methods.

- When the graph is not known, the paper designs an alternating minimization method, which learns the kernel and the Laplacian in an alternative fashion.

Some theoretical analysis and experimental results are used to illustrate the above.
- In particular, the theoretical analysis concerns the generalization error of learning the kernel.
- The experiments apply the alternating minimization method on a weather forecasting dataset and an NBA player movement dataset. The results illustrate the presence of associations within the input features.

**Audience:**

Yes

**Claims And Evidence:**

Yes

**Requested Changes:**

I think this paper introduces a useful framework for learning relational maps given multiple variables. The algorithm design is sound. The experiment findings would be interesting to the community. At the same time, I think the paper requires some revision, for instance reducing supporting information from the main text that are irrelevant from the main claims. Besides, experiment codes should be released.

**Strengths And Weaknesses:**

S1) A useful framework for learning structural associations between input features. I could imagine use cases of this framework for learning relations among the input features.

S2) Overall, paper is clearly written and easy to follow.

S3) Experiments are helpful for understanding the results.

=========================

W1) The paper is quite long and I find that lots of the content are unrelated to the main claims of the paper.

- In Sec 5.1, this section focuses learning with known graph structure; is this setting used elsewhere in the paper? If I understand correctly, both settings in the experiments are in the case where the graph structure is not known.

- In Sec 6, things like Definition 6.1 and Lemma 6.2 should probably be in the appendix (which look like textbook materials).

- The proof of Lemma 6.6 should also be in the appendix (steps look standard).

W2) Experiment codes are not available.

W3) There are no ablation studies detailing each component of the algorithm. There is no ablation on the effect of the regularization penalty as well.

---

> ### Author Response · Authors · 2024-01-17
> **Response to Reviewer VaBc**
>
> Weaknesses:
>
> W1. We have now moved section 5.1 (on known graph structure) in the earlier version to the appendix. Thus in the revised version, Section 5.1 now corresponds to the unknown graph case. We have also moved the proof of Lemma 6.6 (in the earlier and revised versions) to the appendix A.9.
>
> Please note that Definition 6.1 and Lemma 6.2 are extensions of a scalar/vector setting to a functional one, where the Rademacher average has been appropriately defined to incorporate the functional nature of the output space $\mathcal{Y}$. Hence we wish to keep them in the main paper.
>
> W2. We have shared our codes for 12-node setting with data generation in supplementary material of the revised version. We assure the reviewers that the codes for our experiments will be released publicly with a link in the paper on acceptance.
>
> W3. We have performed ablation studies for the components involved in our algorithm the details of which can be found in Appendix A.10.5 of the revised version. As part of the ablation study, we have considered removal of $m_{trace}$ constraint as well as the removal of MCP-based sparsity regularization where we ignore the $L$-based and $D$-based regularizations as well. We have also included the details of experiments illustrating the impact of choosing $m_{trace}$.
>
> Requested Changes:
>
> 1. We have considered the suggestions for improving the readability of the paper in our revised version. We have also shared the experiment codes which we plan to release publicly.

---

### Decision · Action_Editor_rcVE · 2024-02-12

**Recommendation:** Accept as is

**Comment:**

The paper formulates a new problem (many-to-one functional regression) and properly motivates it.  The paper provides a method to address this challenge, and demonstrates empirically that it is general and effective.  In some cases, convergence bounds are provided and proven.  The scope is clearly within machine learning.

The reviewers felt the authors were appropriately responsive to their comments.  Ultimately, the improvements made were minor, and the paper is ready for publication.

**Audience:**

Yes.

**Claims And Evidence:**

Yes.

---

> ### Author Response · Authors · 2024-03-13
>
> The camera-ready version of this paper has now been submitted for publication. We extend our sincere appreciation to the action editor and reviewers for their perceptive comments and invaluable inputs through the revision process. Your expertise has played a crucial role in refining our work to its ultimate rendition.